

# The cranial morphology of *Tanystropheus hydroides* (Tanystropheidae, Archosauromorpha) as revealed by synchrotron microtomography

Stephan N.F. Spiekman[1], James M. Neenan[2], Nicholas C. Fraser[3], Vincent Fernandez[4,5], Olivier Rieppel[6], Stefania Nosotti[7] and Torsten M. Scheyer[1]

[1] University of Zurich, Palaeontological Institute and Museum, Zurich, Switzerland
[2] Oxford University Museum of Natural History, Oxford, UK
[3] National Museums Scotland, Edinburgh, UK
[4] European Synchrotron Radiation Facility, Grenoble, France
[5] The Natural History Museum, London, UK
[6] Field Museum of Natural History, Chicago, IL, USA
[7] Museo Civico di Storia Naturale di Milano, Milan, Italy

Corresponding author
Stephan N.F. Spiekman,
stephanspiekman@gmail.com

## ABSTRACT

The postcranial morphology of the extremely long-necked *Tanystropheus hydroides* is well-known, but observations of skull morphology were previously limited due to compression of the known specimens. Here we provide a detailed description of the skull of PIMUZ T 2790, including a partial endocast and endosseous labyrinth, based on synchrotron microtomographic data, and compare its morphology to that of other early Archosauromorpha. In many features, such as the wide and flattened snout and the configuration of the temporal and palatal regions, *Tanystropheus hydroides* differs strongly from other early archosauromorphs. The braincase possesses a combination of derived archosaur traits, such as the presence of a laterosphenoid and the ossification of the lateral wall of the braincase, but also differs from archosauriforms in the morphology of the ventral ramus of the opisthotic, the horizontal orientation of the parabasisphenoid, and the absence of a clearly defined crista prootica. *Tanystropheus hydroides* was a ram-feeder that likely caught its prey through a laterally directed snapping bite. Although the cranial morphology of other archosauromorph lineages is relatively well-represented, the skulls of most tanystropheid taxa remain poorly understood due to compressed and often fragmentary specimens. The recent descriptions of the skulls of *Macrocnemus bassanii* and now *Tanystropheus hydroides* reveal a large cranial disparity in the clade, reflecting wide ecological diversity, and highlighting the importance of non-archosauriform Archosauromorpha to both terrestrial and aquatic ecosystems during the Triassic.

## INTRODUCTION

Archosauromorpha, the lineage that includes modern crocodylians and birds, first appeared in the Permian and subsequently radiated during the Triassic into one of the dominant vertebrate groups of the terrestrial realm (*Ezcurra, Scheyer & Butler, 2014*; *Foth et al., 2016*). Among the earliest members of the lineage are the non-archosauriform archosauromorphs, which consist of *Protorosaurus speneri*, *Prolacerta broomi*, the herbivorous Rhynchosauria and Allokotosauria, and the long-necked Tanystropheidae (*Ezcurra, 2016*). Tanystropheidae represents a particularly ecomorphologically diverse group that includes terrestrial (e.g. *Macrocnemus bassanii* and *Langobardisaurus pandolfii*), largely aquatic (*Tanytrachelos ahynis* and *Tanystropheus hydroides*), and possibly even fully marine (*Dinocephalosaurus orientalis*) taxa (*Liu et al., 2017*; *Miedema et al., 2020*; *Olsen, 1979*; *Renesto & Dalla Vecchia, 2000*; *Rieppel, Li & Fraser, 2008*; *Spiekman et al., 2020*). The clade had a likely worldwide distribution and occurred between the Early and Late Triassic (*De Oliveira et al., 2018*, *2020*; *Formoso et al., 2019*; *Pritchard et al., 2015*; *Sennikov, 2011*; *Spiekman & Scheyer, 2019*). Due to their unique morphology, diversity, distribution, and phylogenetic position, Tanystropheidae are important both in reconstructing early archosauromorph evolution and in understanding the complexity and composition of Triassic faunas.

Tanystropheids are characterized by their elongate cervical vertebrae and accompanying cervical ribs, and individual taxa are often diagnosed based on characters of these and other postcranial elements. However, due to the generally poor and fragmentary preservation of specimens, our understanding of tanystropheids is limited, and information on skull morphology in particular is sparse. Nevertheless, tanystropheids likely exhibited widely diverse cranial morphologies, as can be deduced from their ecological disparity and the diversity of their dentitions, which range from small conical teeth in for instance *Macrocnemus bassanii* and *Amotosaurus rotfeldensis*, to the 'fish-trap' type dentition of *Tanystropheus hydroides*, to the partially tricuspid dentition of *Tanystropheus longobardicus* and *Langobardisaurus pandolfii* (*Fraser & Rieppel, 2006*; *Li et al., 2017*; *Miedema et al., 2020*; *Rieppel, Li & Fraser, 2008*; *Spiekman et al., 2020*). The dental morphology of *Langobardisaurus pandolfii* in particular is peculiar, as the premaxilla was likely edentulous and the posteriormost teeth of both the upper and lower jaw were modified into large and flat tooth plates used for crushing, thus representing a unique dental system among tetrapods (*Renesto & Dalla Vecchia, 2000*; *Saller, Renesto & Dalla Vecchia, 2013*).

In contrast to the poorly known skull morphology of tanystropheids, largely complete and generally three-dimensionally preserved skulls are known from other early archosauromorphs. Their morphology has revealed valuable insights into archosauromorph palaeobiology and phylogeny, and has shed light on the acquisition of typical archosaur characters such as the presence of recurved teeth, an antorbital and mandibular fenestra, and the loss of the pineal foramen (*Flynn et al., 2010*; *Pinheiro, Simão-Oliveira & Butler, 2019*; *Spiekman, 2018*).

Synchrotron radiation X-ray micro computed tomography (SRμCT) has recently revealed the cranial morphology of the tanystropheid taxa *Macrocnemus bassanii* and *Tanystropheus hydroides* in previously unachievable detail, providing much improved cranial reconstructions (*Miedema et al., 2020*; *Spiekman et al., 2020*). This has shown that the cranial morphology of the terrestrial *Macrocnemus bassanii* is remarkably similar to that of *Prolacerta broomi* and that *Macrocnemus bassanii* possessed many characters that are likely plesiomorphic to Archosauromorpha and Tanystropheidae (*Miedema et al., 2020*). In contrast, *Tanystropheus hydroides* exhibits a highly derived cranial morphology that bears several adaptations indicating that it was an aquatic ambush predator (*Spiekman et al., 2020*). Furthermore, its morphology, together with osteohistological data, revealed that *Tanystropheus hydroides* represents a separate species from the smaller specimens known from the same localities that are referred to *Tanystropheus longobardicus*.

The aim of this study is to describe the skull and preserved cervical vertebrae of PIMUZ T 2790 in high detail based on the SRμCT data. This represents the most complete and detailed cranial description of any tanystropheid to date and expands our understanding of early archosauromorph cranial diversity and evolution.

## MATERIALS AND METHODS

PIMUZ T 2790 consists of eight cervical vertebrae, including the atlas and axis, their associated cervical ribs, and a nearly complete, dorsoventrally compacted skull (Figs. 1 and 2). The specimen was figured in *Wild (1973)* but not described as it was considered too poorly preserved. The length of the cervical vertebrae of the specimen were also used for comparison in *Nosotti (2007*: figure 54*)*. The specimen was discovered in 1952 at the Punkt 902 locality of the Besano Formation (formerly Grenzbitumenzone), which is of latest Anisian to earliest Ladinian age (*Stockar, 2010*).

The specimen was SRμCT scanned at BM05 beamline of the European Synchrotron Radiation Facility (ESRF, Grenoble, France). The resulting data were segmented using Mimics Research v19.0 (https://www.materialise.com/en/medical/mimics-innovation-suite/mimics; Materialise NV, Leuven, Belgium). The skull of PIMUZ T 2790 is dorsoventrally compressed and most elements have become disarticulated and overlap each other, hampering observation of their morphology and the overall configuration of the skull (Fig. 3). Using Blender 2.7 (https://blender.org; Stitching Blender Foundation, Amsterdam, the Netherlands), the elements were digitally positioned in their perceived in-vivo positions, thus 're-assembling' the skull (Figs. 4–6). Blender 2.7 and Mimics Research v19.0 were also used to render images for publication, some of which are also presented in *Spiekman et al. (2020)*. A more detailed overview of the data acquisition and processing can be found in the "Synchrotron micro Computer Tomography acquisition and image processing" section of the "Material and Methods" in *Spiekman et al. (2020)*.

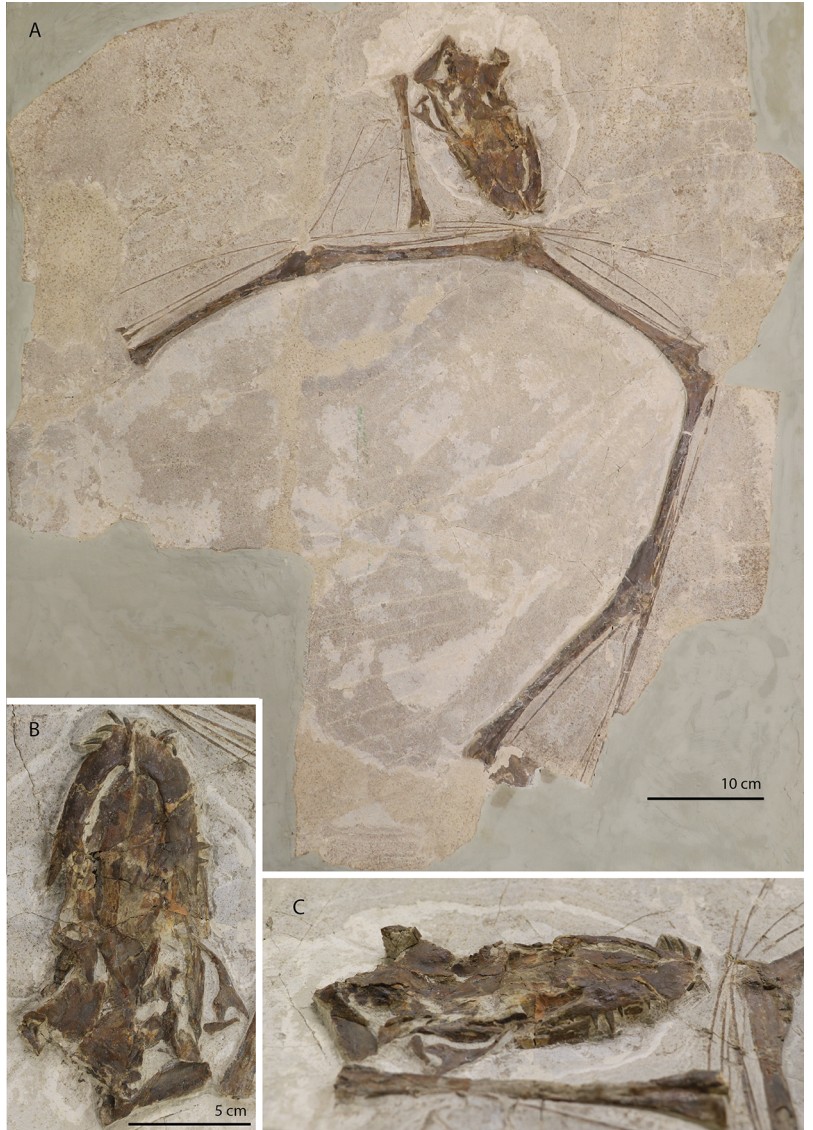

**Figure 1** **The holotype of *Tanystropheus hydroides* PIMUZ T 2790.** (A) The complete specimen. (B) Close-up of the skull in dorsal view. (C) Close-up of the skull in oblique right lateral view.

## RESULTS

### Systematic palaeontology

Diapsida *Osborn, 1903*

Archosauromorpha *Huene, 1946*

Tanystropheidae *Camp, 1945*

*Tanystropheus Meyer, 1855*

*Tanystropheus hydroides* Spiekman, Neenan, Fraser, Fernandez, Rieppel, Nosotti, Scheyer, 2020

#### Holotype

PIMUZ T 2790, a virtually complete, strongly dorsoventrally compressed skull and the first eight cervical vertebrae.

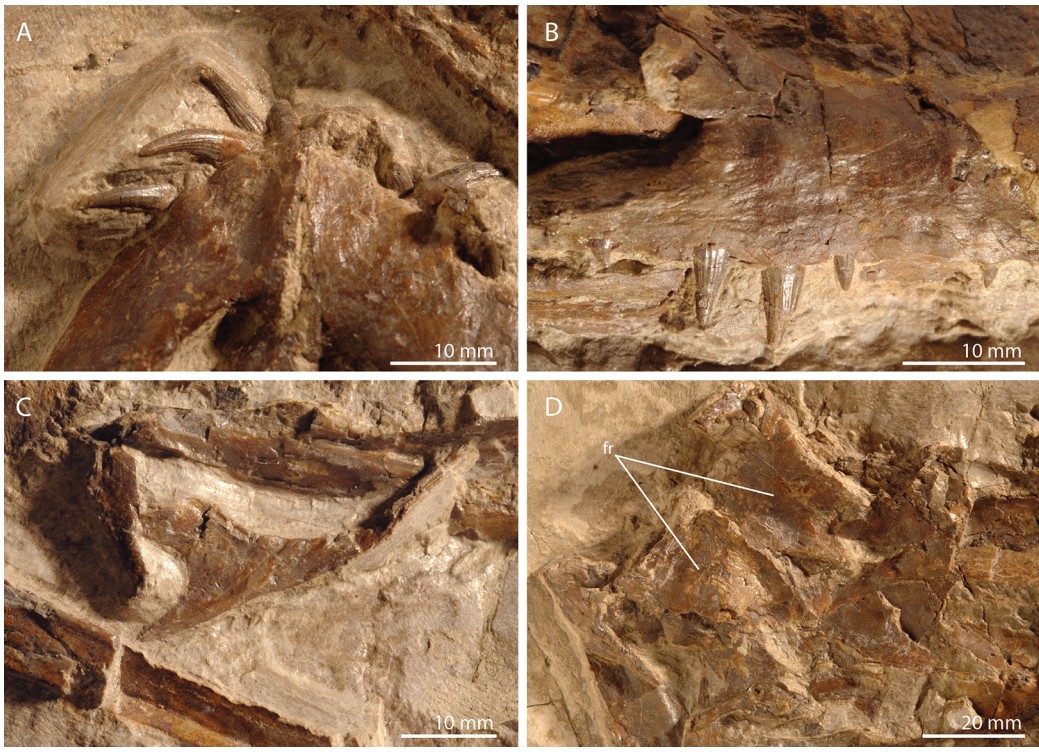

**Figure 2 Detailed images of the skull of PIMUZ T 2790.** (A) Premaxillae. (B) Right maxilla. (C) Right jugal. (D) Posterior skull region including the large and plate-like frontals. Abbreviation: fr, frontal.

### Referred material

MSNM BES 351*, MSNM V 3663, PIMUZ T 1270*, PIMUZ T 1307*, PIMUZ T 2480*, PIMUZ T 2483*, PIMUZ T 2497*, PIMUZ T 2787, PIMUZ T 2788*, PIMUZ T 2793, PIMUZ T 2818, PIMUZ T 2819, PIMUZ T 183, PIMUZ T 2817*, SNSB-BSPG 1953 XV 2.

(Specimens from the Besano Formation referred to *Tanystropheus hydroides* based on relative body size but lacking diagnostic cranial material are indicated by an asterisk).

### Locality

Monte San Giorgio on the border of Switzerland (canton Ticino) and Italy (Lombardy).

### Horizon

Besano Formation, Anisian-Ladinian boundary, Middle Triassic.

### Diagnosis

The following diagnosis has also been presented in *Spiekman et al. (2020)*. *Tanystropheus hydroides* can be distinguished from other *Tanystropheus* species based on the following combination of characters (autapomorphies among Triassic archosauromorphs indicated by an asterisk): premaxilla lacking a postnarial process; single cusped marginal dentition; dentary tooth piercing through a foramen in the maxilla*; depression on the dorsal surface of the nasals; straight suture between frontals; fused parietals; conspicuously hooked dorsal quadrate head; wide and anteriorly rounded vomers with a single row of

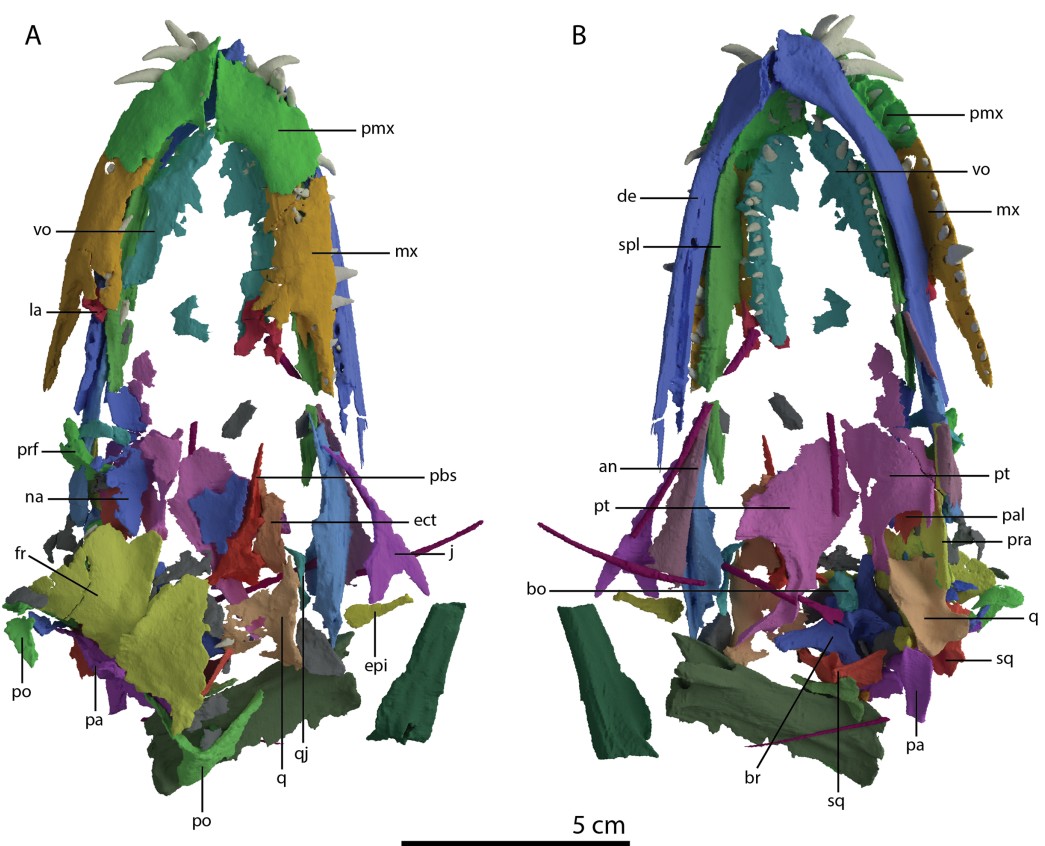

**Figure 3 Digital reconstruction of the skull and proximal cervical vertebrae of PIMUZ T 2790.** (A) Dorsal view. (B) Ventral view. Abbreviations: an, angular; bo, basioccipital; br, braincase; de, dentary; ect, ectopterygoid; epi, epipterygoid; fr, frontal; j, jugal; la, lacrimal; mx, maxilla; na, nasal; pa, parietal; pal, palatine; pbs, parabasisphenoid; pmx, premaxilla; po, postorbital; pra, prearticular; prf, prefrontal; pt, pterygoid; q, quadrate; qj, quadratojugal; sa, surangular; spl, splenial; sq, squamosal; vo, vomer. Parts of this figure have also been presented in *Spiekman et al. (2020)*.

large recurved teeth along its outer margin*; edentulous palatine and pterygoid; dentary bearing a distinct ventral keel at its anterior end*; a maximum total length of over 5 m.

### Remarks

Recently only specimens preserving diagnostic cranial material (e.g. the presence of exclusively single cusped marginal dentition) were referred to *Tanystropheus hydroides* (*Spiekman et al., 2020*). Specimens from the Besano Formation of Monte San Giorgio exceeding the known size range for *Tanystropheus longobardicus* of approximately 2 m that did not preserve these diagnostic features were referred to *Tanystropheus* cf. *T. hydroides* (identified as the large morphotype of *Tanystropheus* cf. *T. longobardicus* in *Spiekman & Scheyer, 2019*). GMPKU-P-1527, a specimen from the Zhuganpo Member of the Falang Formation in China, which in both size and postcranial morphology is indistinguishable from *Tanystropheus hydroides* (*Rieppel et al., 2010*), was also identified as *Tanystropheus* cf. *T. hydroides*. Furthermore, isolated remains, mostly comprising cervical vertebrae, from the Upper Muschelkalk of Europe that have been referred to the

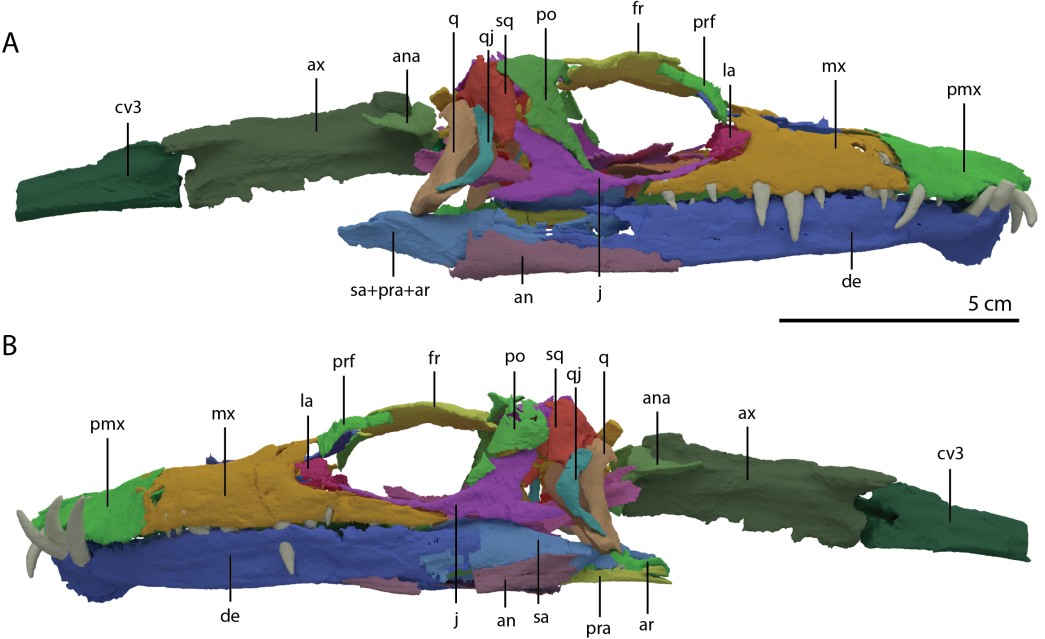

**Figure 4 'Re-assembled' digital reconstruction of the skull and proximal cervical vertebrae of PIMUZ T 2790.** (A) Right lateral view. (B) Left lateral view. Abbreviations: an, angular; ana, atlas neural arch; ar, articular; ax, axis; cv3, cervical vertebra 3; de, dentary; fr, frontal; j, jugal; la, lacrimal; mx, maxilla; pmx, premaxilla; po, postorbital; pra, prearticular; prf, prefrontal; q, quadrate; qj, quadratojugal; sa, surangular; sq, squamosal. Images of this figure have also been presented in *Spiekman et al. (2020)*.

nomen dubium *Tanystropheus* 'conspicuus' are also morphologically indistinguishable from *Tanystropheus hydroides* (*Spiekman & Scheyer, 2019*). Due to the high interspecific variability in the morphology of the skull compared to the postcranium in *Tanystropheus* (*Spiekman et al., 2020*; *Spiekman & Scheyer, 2019*), we maintain that both GMPKU-P-1527 and specimens currently referred to *Tanystropheus* 'conspicuus' cannot be assigned to *Tanystropheus hydroides* as long as insufficient diagnostic cranial material is known from the localities from which these specimens originate. However, the Besano Formation represents a relatively restricted habitat, both spatially and temporally (*Stockar, 2010*). Since there is currently no evidence for the presence of a third *Tanystropheus* species from this formation, we consider the probability very low that the large sized specimens lacking diagnostic cranial features represent a separate species from *Tanystropheus hydroides*. Therefore, we refer these specimens to *Tanystropheus hydroides* here (contra *Spiekman et al., 2020*).

## Comparative morphological description
### *Skull*

Even though the skull of PIMUZ T 2790 is dorsoventrally compacted, most of the bones still preserve a three-dimensional morphology with only certain bones being somewhat deformed (Fig. 3). This is in stark contrast to the other known skulls of *Tanystropheus hydroides*, which are all largely or completely flattened. Most of the bones are preserved underneath the two large plate-like frontals, which have been displaced somewhat

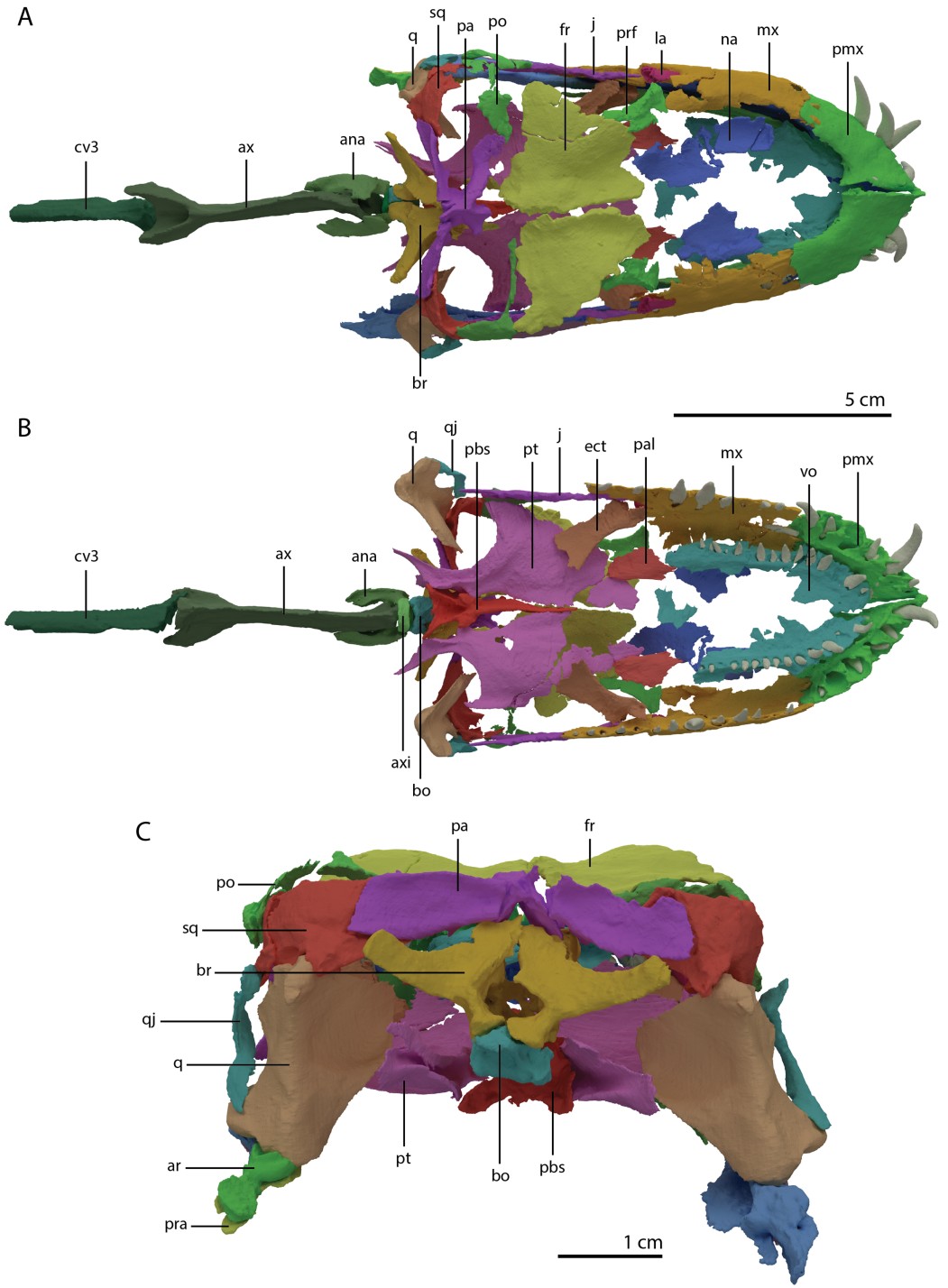

**Figure 5** 'Re-assembled' digital reconstruction of the skull and proximal cervical vertebrae of PIMUZ T 2790. (A) Dorsal view. (B) Ventral view. (C) Occipital view excluding proximal cervical vertebrae. Abbreviations: ana, atlas neural arch; ar, articular; ax, axis; axi, axis intercentrum; bo, basioccipital; br, braincase; cv3, cervical vertebra 3; ect, ectopterygoid; fr, frontal; j, jugal; la, lacrimal; mx, maxilla; na, nasal; pa, parietal; pal, palatine; pbs, parabasisphenoid; pmx, premaxilla; po, postorbital; pra, prearticular; prf, prefrontal; pt, pterygoid; q, quadrate; qj, quadratojugal; sq, squamosal; vo, vomer. Images of this figure have also been presented in *Spiekman et al. (2020)*.

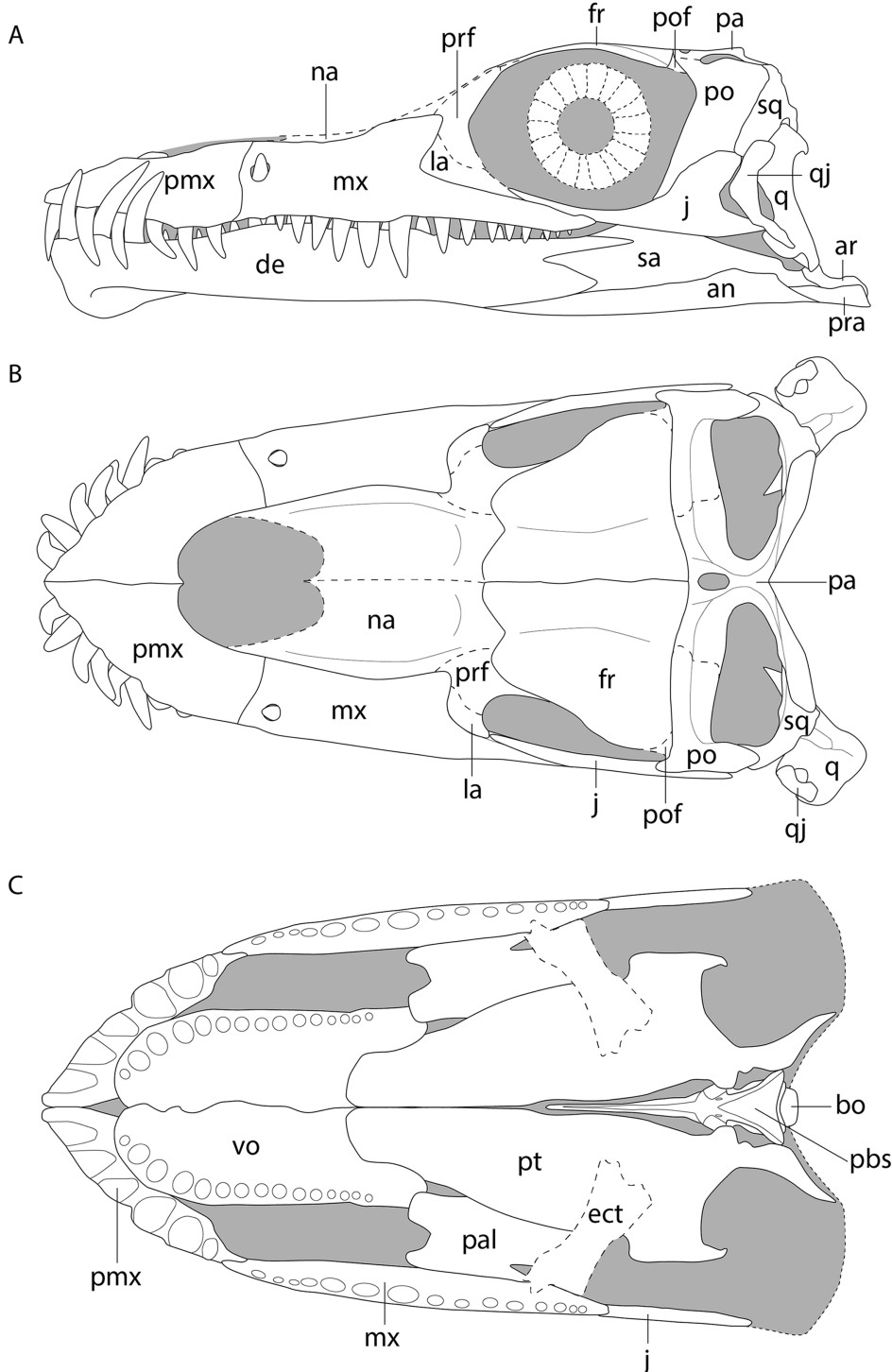

**Figure 6 Reconstruction drawing of the skull of *Tanystropheus hydroides* largely based on PIMUZ T 2790.** (A) Left lateral view. (B) Dorsal view. (C) Ventral view. Abbreviations: an, angular; ar, articular; bo, basioccipital; de, dentary; ect, ectopterygoid; fr, frontal; j, jugal; la, lacrimal; mx, maxilla; na, nasal; pa, parietal; pal, palatine; pbs, parabasisphenoid; pmx, premaxilla; po, postorbital; pof, postfrontal; pra, prearticular; prf, prefrontal; pt, pterygoid; q, quadrate; qj, quadratojugal; sq, squamosal; vo, vomer. Images of this figure have also been presented in *Spiekman et al. (2020)*.

posteriorly from the mandibular rami, premaxillae and maxillae, and as such protected the bones underneath from breakage and distortion. The length of the skull is 138 mm (from the tip of the premaxilla to the right retroarticular process; the posterior extent of the skull cannot be established in-situ).

*Premaxilla*

Both premaxillae are complete and in articulation at the anterior end of the snout (Fig. 2A). Each bears six alveoli, as is also the case in *Tanystropheus longobardicus* (*Spiekman & Scheyer, 2019*). The lateral surface of the premaxilla is plate-like, and the premaxilla maintains its height along most of its anteroposterior length but anteriorly gradually tapers to a point (Fig. 7). The nasals probably only connected to the premaxillae on their anterolateral margin (Fig. 6B). No clear prenarial process is present. Instead, there is a small posterior extension on the medial end of the bone, which does not bear an articulation surface for the nasal to form an internarial bar. The prenarial process of *Tanystropheus longobardicus* and *Macrocnemus* spp. is also incipient, and has been reduced completely in rhynchosaurs, *Teyujagua paradoxa*, and the allokotosaurs *Azendohsaurus madagaskarensis*, *Pamelaria dolichotrachela*, and *Shringasaurus indicus* among early archosauromorphs (*Dilkes, 1998*; *Flynn et al., 2010*; *Miedema et al., 2020*; *Nosotti, 2007*; *Pinheiro, Simão-Oliveira & Butler, 2019*; *Sengupta, Ezcurra & Bandyopadhyay, 2017*). In contrast, the prenarial process is well-established and elongate in *Protorosaurus speneri*, *Prolacerta broomi*, *Dinocephalosaurus orientalis*, and *Pectodens zhenyuensis* (*Gottmann-Quesada & Sander, 2009*; *Li et al., 2017*; *Modesto & Sues, 2004*; *Rieppel, Li & Fraser, 2008*).

A postnarial process is also absent in *Tanystropheus hydroides* and the suture between the premaxilla and maxilla is consequently almost vertical and directly posterior to the last alveolus of the premaxilla (Figs. 7A and 7B). The posteriormost part of the premaxillary body is labiolingually flattened, indicating that this part would have overlapped the maxilla laterally. This represents the opposite morphology of that recently described for rhynchosaurs, in which the maxilla laterally overlaps the premaxilla distinctly (see supplemental figure 11 of *Pritchard et al., 2018*). The premaxilla of *Tanystropheus longobardicus* bears a pronounced posteriorly directed postnarial process that would have articulated on the dorsolateral surface of the anterior part of the maxilla (e.g. MSNM BES SC 1018, PIMUZ T 2484; *Nosotti, 2007*). A similar postnarial process is also present in *Prolacerta broomi*, *Azendohsaurus madagaskarensis*, and *Teyujagua paradoxa* in which this process forms a simple articulation with the maxilla (*Flynn et al., 2010*; *Pinheiro, Simão-Oliveira & Butler, 2019*; *Spiekman, 2018*). The premaxilla of *Macrocnemus bassanii* also has an elongate postnarial process, but additionally bears a posteromedial process, and these two processes form a complicated articulation with the maxilla (*Miedema et al., 2020*). Since the medial surface of the premaxilla cannot be observed for any known specimen, it is unclear whether *Tanystropheus longobardicus* possessed a similar posteromedial process.

As in *Macrocnemus* spp., there is no lingual contribution of the premaxilla to the palate in *Tanystropheus hydroides* (*Miedema et al., 2020*; Fig. 7D). No foramina are present on the

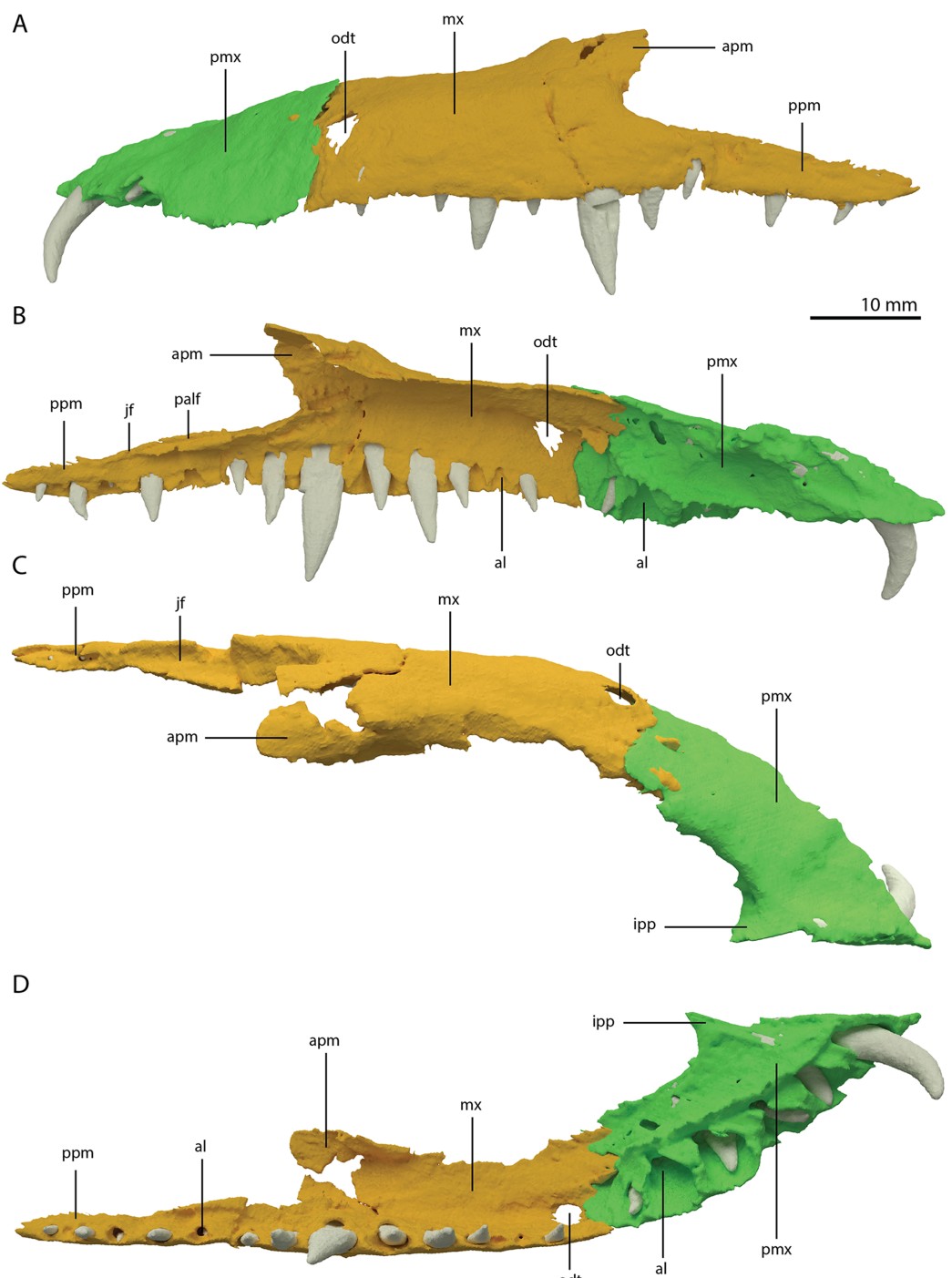

**Figure 7 Articulated digital reconstruction of the left premaxilla and maxilla of PIMUZ T 2790.** (A) Lateral or labial view. (B) Medial or lingual view. (C) Dorsal view. (D) Ventral view. Abbreviations: al, alveolus; apm, ascending process maxilla; ipp, incipient prenarial process; jas, jugal facet; mx, maxilla; odt, opening for dentary tooth; palf, palatine facet; pmx, premaxilla; ppm, posterior process maxilla.

premaxilla. In contrast, several small neurovascular foramina line the premaxilla of *Tanystropheus longobardicus* (MSNM BES SC 1018; *Nosotti, 2007*).

*Maxilla*

The left maxilla is complete except for its anteriormost portion, which is broken. The anteriormost portion of the right maxilla is similarly broken and it additionally misses the posteriormost part of its posterior process. The left maxilla preserves 15 alveoli, whereas only 11 are present on the less complete right element. Even though the anterior portion of both maxillae are somewhat poorly preserved, it is clear that they do not taper. Instead, each has a tall, almost vertical anterior margin (Fig. 2B and 7). The anterior part of the dorsal margin is largely horizontal and would have articulated with the lateral margin of the nasal (Fig. 6B). Posteriorly, the dorsal margin of the maxilla rises to form an ascending process with a distinctly concave posterior margin. This morphology occurs widely among non-archosauriform archosauromorphs, with the notable exception of *Protorosaurus speneri* (*Gottmann-Quesada & Sander, 2009*). The dorsal margin of the posterior process of the maxilla is wide in both bones and bears a concave articulation facet, anteriorly for the lacrimal and perhaps the prefrontal, and posteriorly for the anterior process of the jugal (Figs. 7B and 7C). On its medial side the dorsal margin of the posterior process is thickened at approximately its mid-length, forming a facet for the lateral margin of the palatine, as well as possibly the distal end of the ectopterygoid (Fig. 7B). The posterior process of the left maxilla is long, being almost subequal in anteroposterior length to the rest of the maxilla. Anteriorly, both maxillae bear a large opening, through which dentary tooth 10 pierced. A similar opening can also be seen in *Tanystropheus hydroides* specimen PIMUZ T 2819 (see supplemental figure 1B of *Spiekman et al., 2020*). No other foramina can be identified on the lateral surface of the maxilla.

*Septomaxilla*

A septomaxilla was previously tentatively assigned to *Tanystropheus hydroides* and *Tanystropheus longobardicus* (*Wild, 1973*). However, no evidence for such an element could be found in the SRμCT data of PIMUZ T 2790, and none of the bones that would surround a septomaxilla (i.e. the premaxilla, vomer and nasal), bear any articulation facets for such a bone. Nevertheless, it cannot be excluded that a small septomaxilla was present in *Tanystropheus hydroides* when taking into consideration the poor preservation of the vomer and the nasal in PIMUZ T 2790. Similarly, the presence of a septomaxilla cannot be determined confidently for *Tanystropheus longobardicus* (*Nosotti, 2007*). Septomaxillae occur in several early archosauromorphs, including *Prolacerta broomi* and the early rhynchosaur *Mesosuchus browni* (*Modesto & Sues, 2004*; SAM-PK-6536, pers. observ. SNFS).

*Nasal*

There are several flat and plate-like bone fragments present anterior to the frontals, which are preserved in a higher plane than the pterygoids and vomers. These fragments are therefore identified as parts of the nasals (Figs. 3A and 5A). They are clearly concave in the transverse plane. Only a short portion of the straight medial margin of the left nasal could be identified. No other margins are preserved. The reconstruction of the nasal of

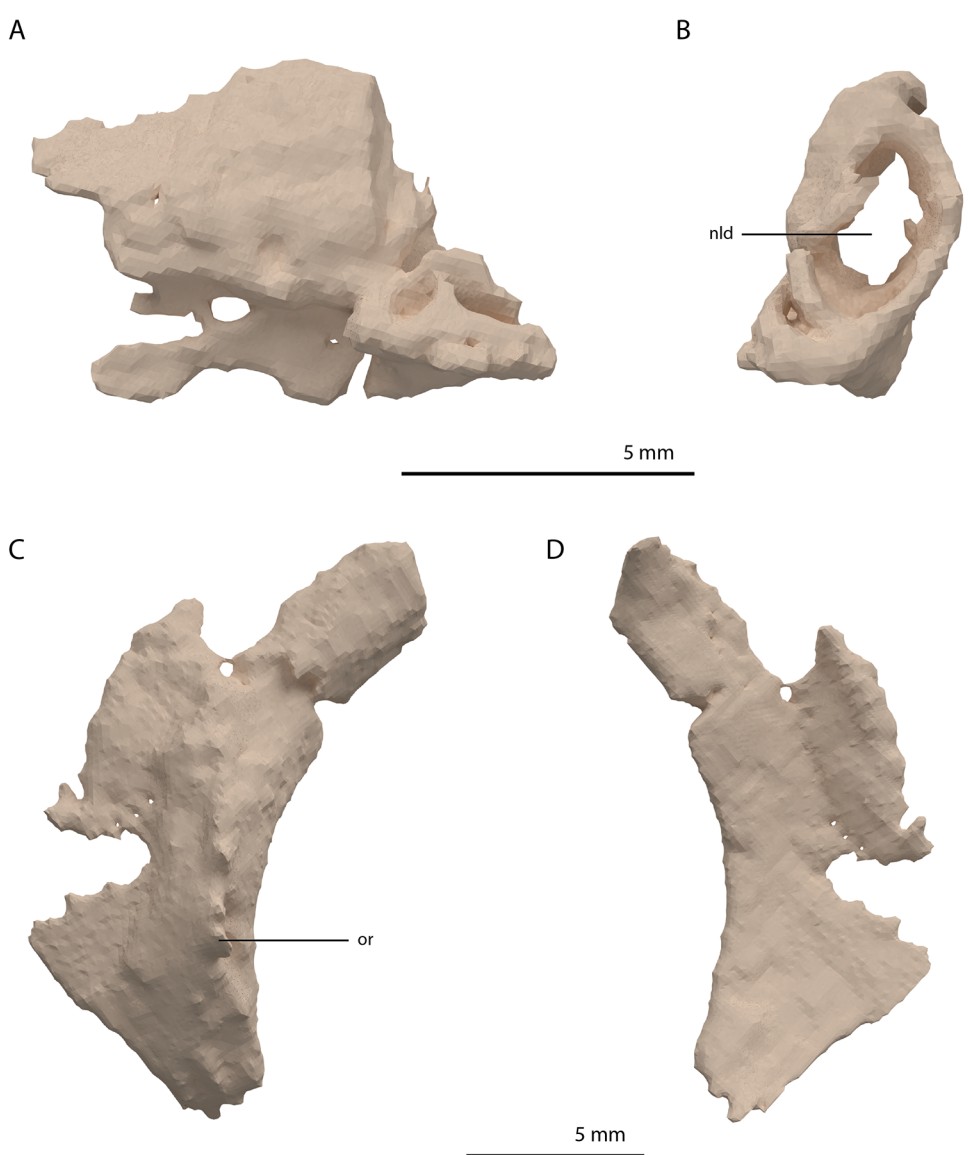

**Figure 8 Digital reconstruction of the left lacrimal and left prefrontal of PIMUZ T 2790.** (A) Left lacrimal in lateral view. (B) Left lacrimal in posterior view. (C) Left prefrontal in lateral view. (D) Left prefrontal in medial view. Abbreviations: nld, nasolacrimal duct; or, orbital rim.

*Tanystropheus hydroides* (Fig. 6B), which is based on inferences from PIMUZ T 2790, PIMUZ T 2819, and PIMUZ T 2787, as well as comparisons to *Tanystropheus longobardicus*, was discussed in *Spiekman et al. (2020)* and is expanded upon in the discussion section below.

### Lacrimal

Directly posterior and medial to the ascending process of the left maxilla, a fragmented bone is preserved which is identified as the left lacrimal (Figs. 8A and 8B). Although its margins are incomplete it bears a large oval-shaped posterior opening, which is the foramen for the naso-lacrimal duct. Another bone with a similar association with the right

maxilla is somewhat bigger than the left lacrimal. However, it is very poorly preserved and cannot be identified confidently.

The prefrontal had a broad anterior and dorsal contact with the frontal, nasal, and maxilla, as can be deduced from the SRμCT data of PIMUZ T 2790 and the better-preserved prefrontal of *Tanystropheus hydroides* specimen PIMUZ T 2819 (see supplemental figure 1A of *Spiekman et al., 2020*). Therefore, the lacrimal was likely restricted to the ventral side of the prefrontal and contacted the maxilla on the ventral part of the posterior margin of the ascending process and along the posterior process of the latter. It possibly also reached the anterior process of the jugal. Based on the prefrontal of PIMUZ T 2819 it seems likely that the lacrimal formed part of the anteroventral margin of the orbit. The lacrimal of *Tanystropheus longobardicus* is best-preserved in MSNM BES SC 1018 and also shows a large posterior opening transmitting the naso-lacrimal duct, albeit comparatively much smaller than in *Tanystropheus hydroides* (*Nosotti, 2007*).

*Prefrontal*

The right prefrontal is missing but a partial left prefrontal is preserved anterolaterally to the left frontal. It has a clear orbital rim formed by a distinctly raised ridge (Figs. 8C and 8D), which is similar to that observed in the right prefrontal of PIMUZ T 2819 and in other non-archosauriform archosauromorphs, including *Tanystropheus longobardicus* (e.g. MSNM BES SC 1018). The prefrontal would have formed the anterodorsal margin of the orbit. The remaining edges of the prefrontal are broken and poorly preserved. They are very likely incomplete, in part because the element was crushed over the left surangular. The prefrontal was orientated in the 're-assembled' skull based on the position of this bone in PIMUZ T 2819, which is in partial articulation (*Wild, 1973*).

*Frontal*

Both frontals are preserved next to each other, posterior to the left mandibular ramus. They are very broad elements, being almost equal in width and anteroposterior length (Figs. 2D and 9). They are at their widest posteriorly and become slightly but steadily narrower anteriorly. This is also the condition in *Tanystropheus longobardicus* (PIMUZ T 2484; figure 4 of *Spiekman & Scheyer, 2019*). In other non-archosauriform archosauromorphs the frontals are considerably less wide relatively, resulting in a much narrower interorbital region. There is a clear sagittally orientated depression on the medial portion of the dorsal surface and the bone has a distinct convex curvature lateral to this depression (Fig. 9C). This curvature forms the rounded dorsal margin of the orbit. Both in *Tanystropheus hydroides* and *Tanystropheus longobardicus*, the contribution of the frontal to the margin of the orbit is remarkably large (see figure 3 of *Spiekman et al., 2020*). In other non-archosauriform archosauromorphs, the contribution of the frontal to the orbital margin is either very small (*Macrocnemus bassanii*, PIMUZ T 4822; *Prolacerta broomi*, BP/1/471; *Mesosuchus browni*, SAM-PK-6536; *Howesia browni*, SAM-PK-5884; *Shringasaurus indicus*, *Sengupta, Ezcurra & Bandyopadhyay, 2017*; and *Teyujagua paradoxa*, *Pinheiro, Simão-Oliveira & Butler, 2019*) or considerable, yet

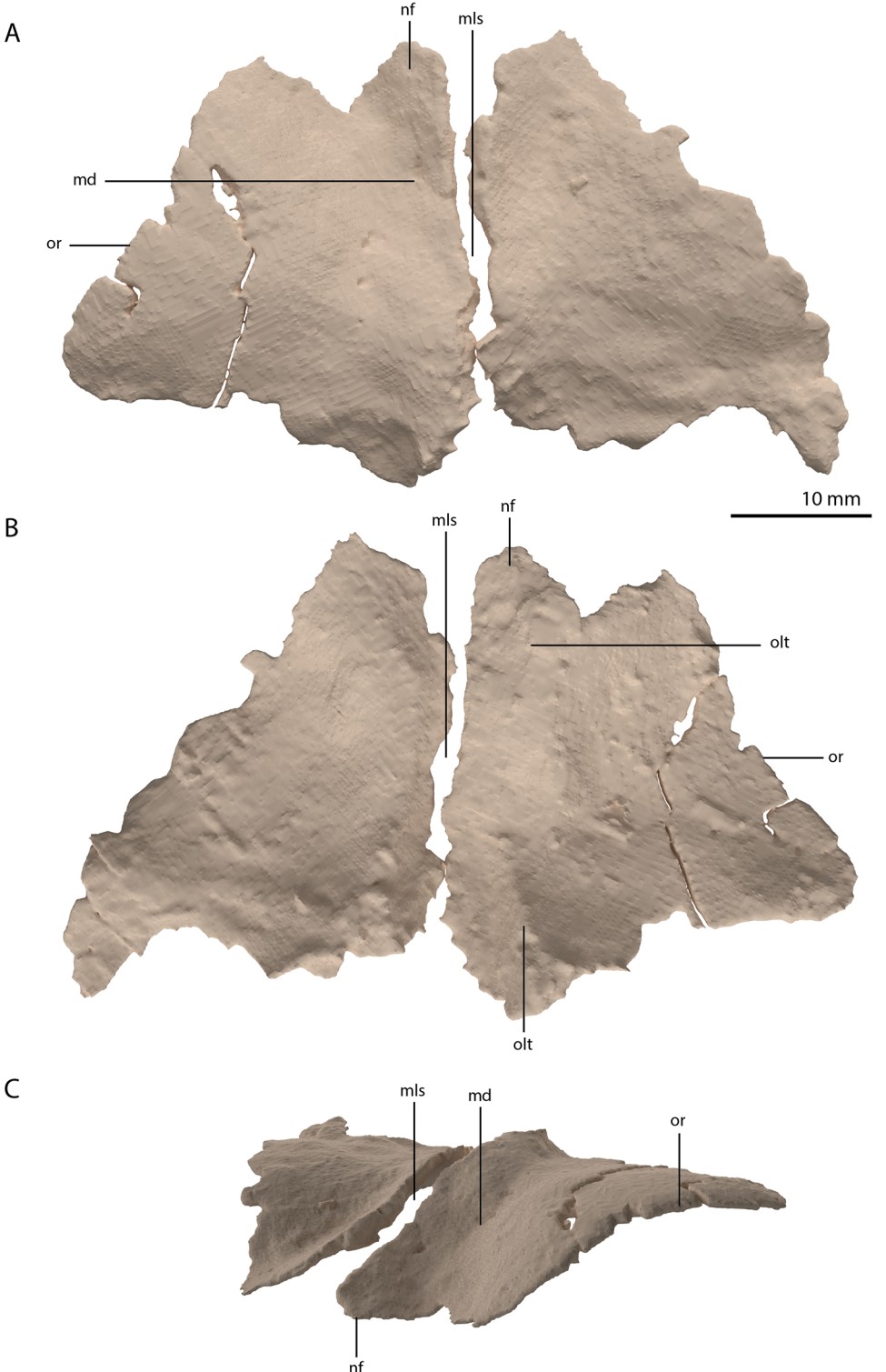

**Figure 9 Digital reconstruction of the frontals of PIMUZ T 2790.** (A) Dorsal view. (B) Ventral view. (C) Oblique left anterolateral view. Abbreviations: md, medial depression; mls, midline suture; nf, nasal facet; olt, olfactory tract; or, orbital rim.

distinctly smaller than that seen in *Tanystropheus hydroides* and *Tanystropheus longobardicus* (*Protorosaurus speneri*, NMK S 180, *Gottmann-Quesada & Sander, 2009*; *Azendohsaurus madagaskarensis*, *Flynn et al., 2010*; and *Teraterpeton hrynewichorum*, *Sues, 2003*). The frontals would have contacted the nasals anteriorly, the prefrontals anterolaterally, the parietals and postorbitals posteriorly, and the postfrontals and likely part of the postorbitals posterolaterally (Fig. 6B). They would have extended little beyond the level of the orbit, both posteriorly and anteriorly. The left frontal is complete, but the anterolateral part of the right frontal is broken. It was previously suggested that the frontals of the large specimens of *Tanystropheus* from Monte San Giorgio (now *Tanystropheus hydroides*) were possibly fused (*Wild, 1973*). However, PIMUZ T 2790 clearly reveals that the frontals are unfused and that the suture between them was straight and simple, in contrast to the interdigitating suture seen *Tanystropheus longobardicus* (PIMUZ T 2484; figure 4A of *Spiekman & Scheyer, 2019*). On the ventral surface of both frontals a faint sagittally oriented ridge is visible, which corresponds to the depression of the dorsal surface and likely represents the margin of a shallow gutter transmitting the olfactory tract (Fig. 9B). It is constricted at about the anteroposterior mid-length of both bones and reaches somewhat further laterally at its anterior end than at the posterior end. Although the ridge is quite faint, it is most pronounced posteriorly. There is no depression on either frontal to accommodate the olfactory bulb as has been observed for *Tanystropheus longobardicus* and *Macrocnemus bassanii* (*Ezcurra, 2016*).

*Parietal*

The parietals are fused but broken into three pieces that became scattered. The main piece is located directly posterior to both frontals and the other two pieces, which represent the two posterolateral processes of the fused parietals, are located to the right and directly below the main body. The anterior part of the fused parietals is largely missing. A partial left anterolateral process is preserved, whereas the right process is completely absent. The bone has been reassembled in the digital reconstruction (Figs. 5 and 10). The anterolateral process is roughly equal in width to the posterolateral processes, indicating that it framed the anterior margin of the supratemporal fenestra completely. The distal portion of this process is dorsoventrally flattened and likely overlapped the medial process of the postorbital, since this postorbital process was relatively long and would have reached close to the midline of the skull (see below). The parietal most likely overlapped the postorbital based on the configuration preserved in PIMUZ T 2819, in which the anterior margin of the fused parietals contacted the frontals in a roughly straight transverse suture across the width of these elements in dorsal view (figure 4B of *Spiekman & Scheyer, 2019*). The posterior portion of the pineal foramen is well-preserved and shows that it was large and with a marked rim. Posterior to the pineal foramen a low sagittal crest runs along the midline of the fused parietals. From this midline crest, the fused parietals slope down steeply on both sides to form a surface area for the attachment of the jaw adductor musculature. These surfaces, the supratemporal fossae, make up most of the dorsolateral side of the main body of the fused parietals. The posterolateral processes are dorsoventrally tall and almost entirely transversely orientated. Distally, the

posterolateral processes slightly expand dorsoventrally. On the anterior surface of both posterolateral processes, a distinct articular surface for the medial process of the squamosals is present (Figs. 10D and 10E). The medial margin of this surface is orientated laterodorsally to medioventrally. It can be inferred from the tight fit between the parietal and squamosal that a supratemporal bone was certainly absent in *Tanystropheus hydroides*. The shape of the fused parietals of PIMUZ T 2790 corresponds with that seen in the well-preserved fused parietals exposed in dorsal view in the *Tanystropheus hydroides* specimen PIMUZ T 2819 (see figure 4B of *Spiekman & Scheyer, 2019*). From our new findings, it can be inferred that the anterolateral processes of the fused parietals of PIMUZ T 2790 are wider than interpreted for this specimen by *Wild (1973)*. Instead, the bones identified there as the postfrontals represent parts of the anterolateral processes of the parietals, as was also reconstructed for this specimen in figure 3 of *Jiang et al. (2011)*. The postfrontals were most likely not clearly exposed in dorsal view in *Tanystropheus hydroides*. The morphology of the parietals differs strongly from that of *Tanystropheus longobardicus*, which is best represented in PIMUZ T 2484 (see figure 4A of *Spiekman & Scheyer, 2019*). In *Tanystropheus longobardicus* the parietals are unfused in the midline and lack the pronounced anterolateral processes. No clear supratemporal fossae are present, and the main body of the parietals is relatively much wider compared to *Tanystropheus hydroides*.

Among non-archosauriform archosauromorphs fused parietals also occur in *Dinocephalosaurus orientalis* (IVPP-V13767), *Protorosaurus speneri* (NMK S 180, *Gottmann-Quesada & Sander, 2009*) and rhynchosaurs (*Butler et al., 2015*; *Dilkes, 1995*; *Dilkes, 1998*). The presence of a pineal foramen is variable among early archosauromorphs, and it is absent in *Macrocnemus* spp. (*Miedema et al., 2020*), *Trilophosaurus buettneri* (*Spielmann et al., 2008*), *Euparkeria capensis* (SAM-PK-5867), and in some specimens of *Prolacerta broomi* (e.g. BP/1/471) and *Proterosuchus* spp. (e.g. SAM-PK-K10603). From PIMUZ T 2819 it can be inferred that the pineal foramen was completely enclosed by the parietals on their anterior portion in *Tanystropheus hydroides* (figure 4B of *Spiekman & Scheyer, 2019*). This condition is common among early archosauromorphs, although the pineal foramen in *Tanytrachelos ahynis* (NMS G.2017.11.1), *Prolacerta broomi* (BP/1/5880), *Proterosuchus fergusi* (BP/1/3993), and *Teyujagua paradoxa* (*Pinheiro, Simão-Oliveira & Butler, 2019*) is positioned at approximately mid-length of the parietals, whereas the pineal foramen of *Azendohsaurus madagaskarensis* is enclosed posteriorly by the parietals and anteriorly by the frontals (*Flynn et al., 2010*). The large lateral extension of the anterolateral processes of the parietals in *Tanystropheus hydroides* is unique among non-archosauriform archosauromorphs but is somewhat reminiscent of the pronounced anterolateral processes seen in erythrosuchids (*Butler et al., 2019a*, *2019b*; *Gower, 2003*). The large and roughly laterally facing supratemporal fossae of the fused parietals in combination with dorsoventrally tall posterolateral processes seen in *Tanystropheus hydroides* represent a similar morphology to that of the comparatively large-sized early archosauromorphs *Azendohsaurus madagaskarensis* and *Dinocephalosaurus orientalis* (IVPP-V13767; *Flynn et al., 2010*; *Rieppel, Li & Fraser, 2008*). It is also present to a lesser degree in the parietals of *Protorosaurus speneri*, in which

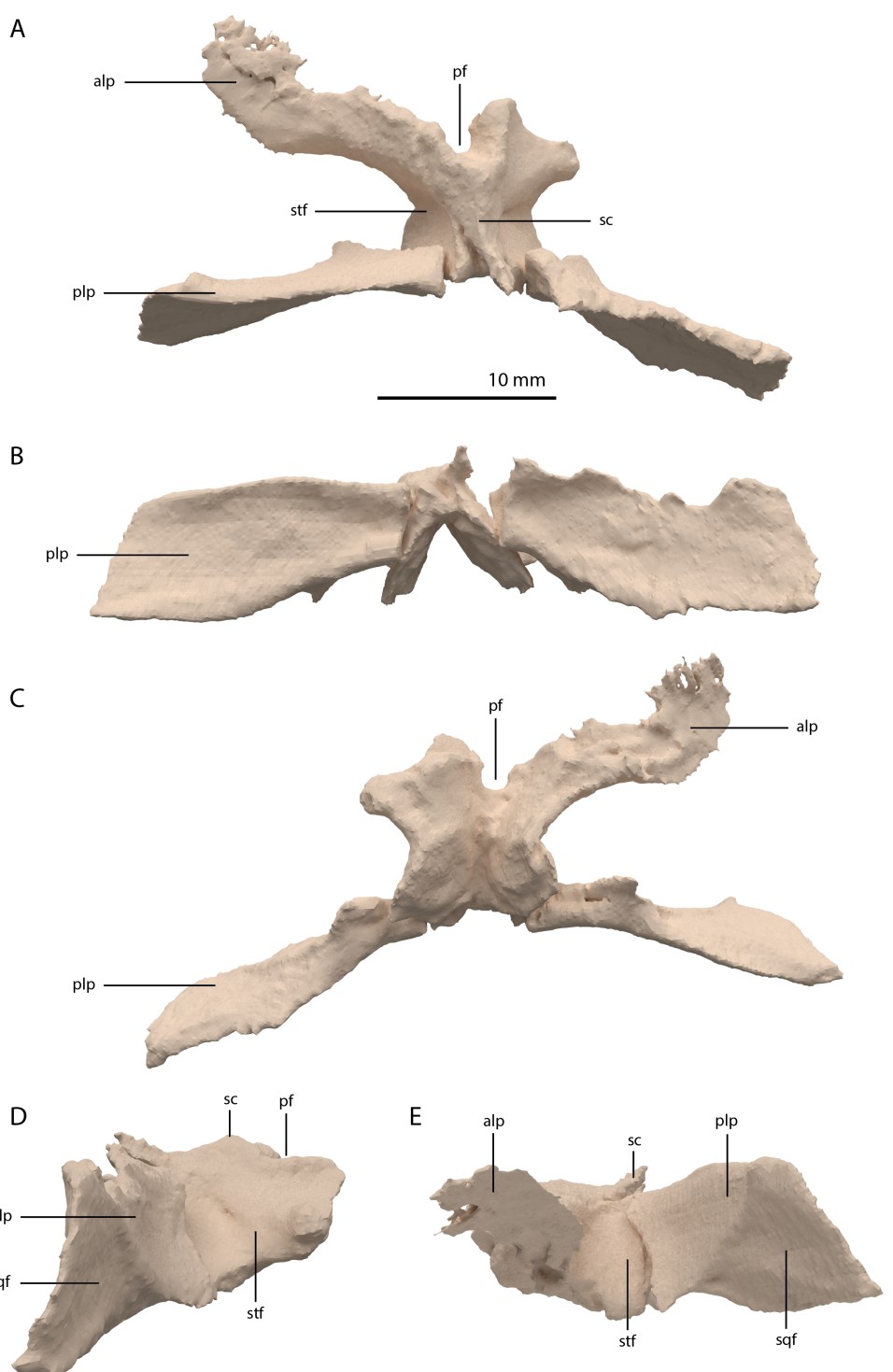

**Figure 10 Digital reconstruction of the parietal of PIMUZ T 2790.** (A) Dorsal view. (B) Posterior or occipital view. (C) Ventral view. (D) Right lateral view. (D) Left lateral view. Abbreviations: alp, ante-rolateral process; pf, pineal foramen; plp, posterolateral process; sc, sagittal crest; sqf, squamosal facet; stf, supratemporal fossa.                      

the supratemporal fossae are also quite large but largely dorsally facing, and which possess narrower posterolateral process (NMK S 180; *Gottmann-Quesada & Sander, 2009*). However, the morphology of *Tanystropheus hydroides* differs distinctly from that seen in the parietals of smaller early archosauromorphs (e.g. *Macrocnemus bassanii*, *Prolacerta broomi*, *Jesairosaurus lehmani*, and *Tanystropheus longobardicus*; PIMUZ T 2484; *Jalil, 1997*; *Miedema et al., 2020*; *Modesto & Sues, 2004*). In these taxa, the supratemporal fossae form less of a depression and are largely dorsally facing, and the posterolateral processes are much narrower. Both the supratemporal fossae and the posterolateral processes of the parietals are important muscle attachment sites for the jaw adductor musculature. However, these differences among early archosauromorphs appear to be more strongly correlated with size rather than phylogeny or feeding strategies, since closely related taxa exhibit strongly different morphologies (e.g. *Tanystropheus hydroides* and *Tanystropheus longobardicus*), whereas relatively large-sized taxa with a widely different diet (e.g. the piscivorous *Dinocephalosaurus orientalis* and the herbivorous *Azendohsaurus madagaskarensis*) show a similar morphology.

### Postfrontal

A postfrontal could not be identified in PIMUZ T 2790. The width at the posterior end of the frontal might indicate that this element was comparatively small and mostly visible in lateral view (Figs. 5A, 6A and 6B). A postfrontal in *Tanystropheus hydroides* had previously been identified in PIMUZ T 2819 (*Wild, 1973*). However, as discussed above these elements in fact represent the elongate anterolateral process of the fused parietals (see also figure 3B of *Jiang et al., 2011*). The lack of an identifiable postfrontal in any available specimen of *Tanystropheus hydroides* precludes any further interpretation without ambiguity. The postfrontal of *Tanystropheus longobardicus* is known from PIMUZ T 2484 (figure 4A of *Spiekman & Scheyer, 2019*). This element is small and triangular and articulates posteromedially with the parietal and anteromedially with the frontal (*Nosotti, 2007*). The postfrontal framed the posterodorsal margin of the orbit, but its articulation with the postorbital is unclear.

### Postorbital

The postorbital is a triradiate bone with two very elongate processes, and one shorter process (Fig. 11). Both postorbitals are preserved, each directly posterolateral to their respective frontals. The right element is the more complete of the two. The two long processes are the ventral and medial processes, of which the ventral process is slightly longer. Both processes are straight and form a slightly acute angle with each other, indicating a very abrupt and sharp transition between the lateral and dorsal surfaces of the postorbital region of the skull. This configuration differs strongly from the postorbital in all other known non-archosauriform-archosauromorphs, in which the ventral and medial or dorsal processes of the postorbital generally form a crescent shape. This sharp transition is further corroborated by the shape of the squamosal, as discussed below. As a result, the medial process was extensive, reaching almost to the midline of the skull. The medial process of the right postorbital is very thin but is incomplete posteriorly. This can be

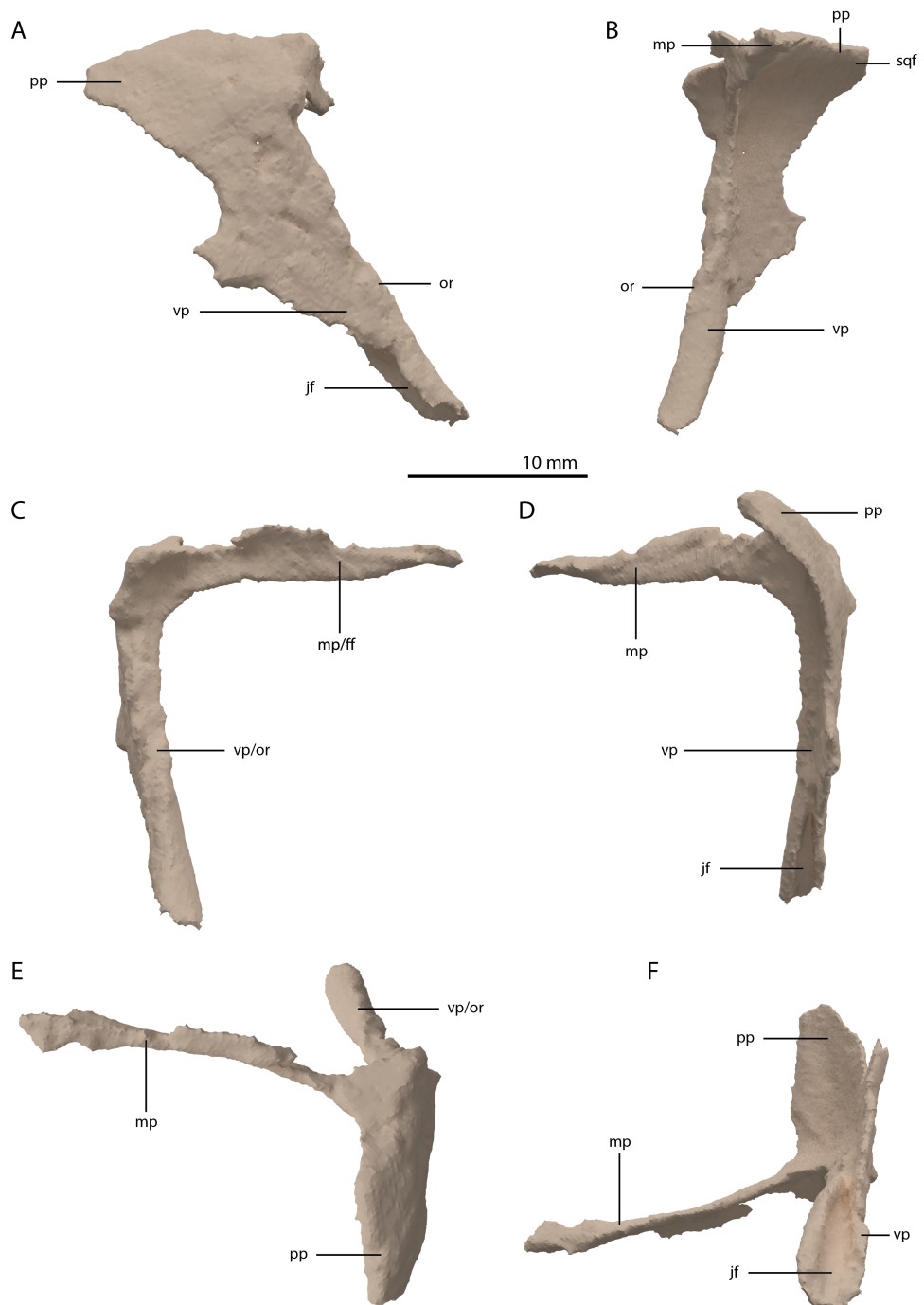

**Figure 11 Digital reconstruction of the right postorbital of PIMUZ T 2790.** (A) Lateral view. (B) Medial view. (C) Anterior view. (D) Posterior view. (E) Dorsal view. (F) Ventral view. Abbreviations: ff, frontal facet; jf, jugal facet; mp, medial process; or, orbital rim; pp, posterior process; sqf, squamosal facet; vp, ventral process.

inferred from the medial process of the less complete left postorbital, which is considerably broader (Fig. 5A). The medial process has a vertically orientated and flat anterior surface that would have formed a long transverse suture with the posterior margins of the

frontal and possibly the postfrontal (Fig. 11C). The ventral process tapers distally, where it bears a clear articulation surface for the ascending process of the jugal on its posterior surface (Figs. 11D and 11F). This facet is deeper and more conspicuous than that observed in *Prolacerta broomi* (BP/1/5066) and similar to that of *Macrocnemus bassanii* (*Miedema et al., 2020*). Although it is partially broken, it seems likely that the ventral process gradually widened posterodorsally and would have been confluent with the posterior process, forming a dorsoventrally broad suture with the squamosal. The posterior process is largely laterally facing, with its dorsal margin forming part of the lateral margin of the supratemporal fenestra. The anterior part of the bone where the medial and ventral processes meet is somewhat thickened. The identification of the postorbital in several *Tanystropheus longobardicus* specimens (PIMUZ T 2791, in PIMUZ T 2484 and MSNM BES SC 265) was recently re-interpreted based on the shape of the postorbital in *Tanystropheus hydroides* (Methods S1 of *Spiekman et al., 2020*). The postorbital of *Tanystropheus longobardicus* is also preserved in MSNM BES SC 1018 and, like *Tanystropheus hydroides*, bears a long ventral process, with a groove on its posterior surface that received the ascending process of the jugal. The medial process of MSNM BES SC 1018 was probably also elongate, whereas the posterior process was comparatively much shorter, as in *Tanystropheus hydroides*. However, due to the lack of three-dimensionally preserved skulls, the exact shape of the postorbital and its articulation with the surrounding bones remains unclear for *Tanystropheus longobardicus*.

*Jugal*
The left jugal is missing in PIMUZ T 2790, but an apparently almost complete right jugal can be observed through external observation (i.e. without the use of SRμCT data) on the specimen lateral to the posterior part of the right mandibular ramus (Fig. 2C). However, parts of this element could not be recovered from the SRμCT data; the jugal has thus been partially reconstructed. The parts that were visible in the SRμCT data are the main body of the jugal, including the base of the anterior and ascending processes, and the complete posterior process, as well as the anterior half of the anterior process and the posterodorsal end of the ascending process (Figs. 12A and 12B). Filling in the missing parts of the jugal based on the well-preserved left jugal of PIMUZ T 2819 (see supplemental figure 1A of *Spiekman et al., 2020*) resulted in a nearly identical reconstruction of the jugal as is visible in the specimen externally (Figs. 12C and 12D). Its shape is virtually identical to that of *Tanystropheus longobardicus* (*Nosotti, 2007*). The anterior process is quite long and curved and tapers to a sharp point anteriorly. It framed the entire ventral margin of the orbit based on the overall length of the process and the clear jugal facet present on the posterior process of the maxilla. The posterior process is directed posteriorly with a largely straight ventral margin and a curved dorsal margin, which meet at the tapered end of the process. Although the process is quite long, no facet is present, and it did not contact any bone posteriorly, and the infratemporal bar was therefore incomplete. The jugal of all known non-archosauriform archosauromorphs bears a posterior process, except for *Dinocephalosaurus orientalis* (IVPP-V13767) and *Pectodens zhenyuensis* (IVPP-V18578). In none of these taxa does this

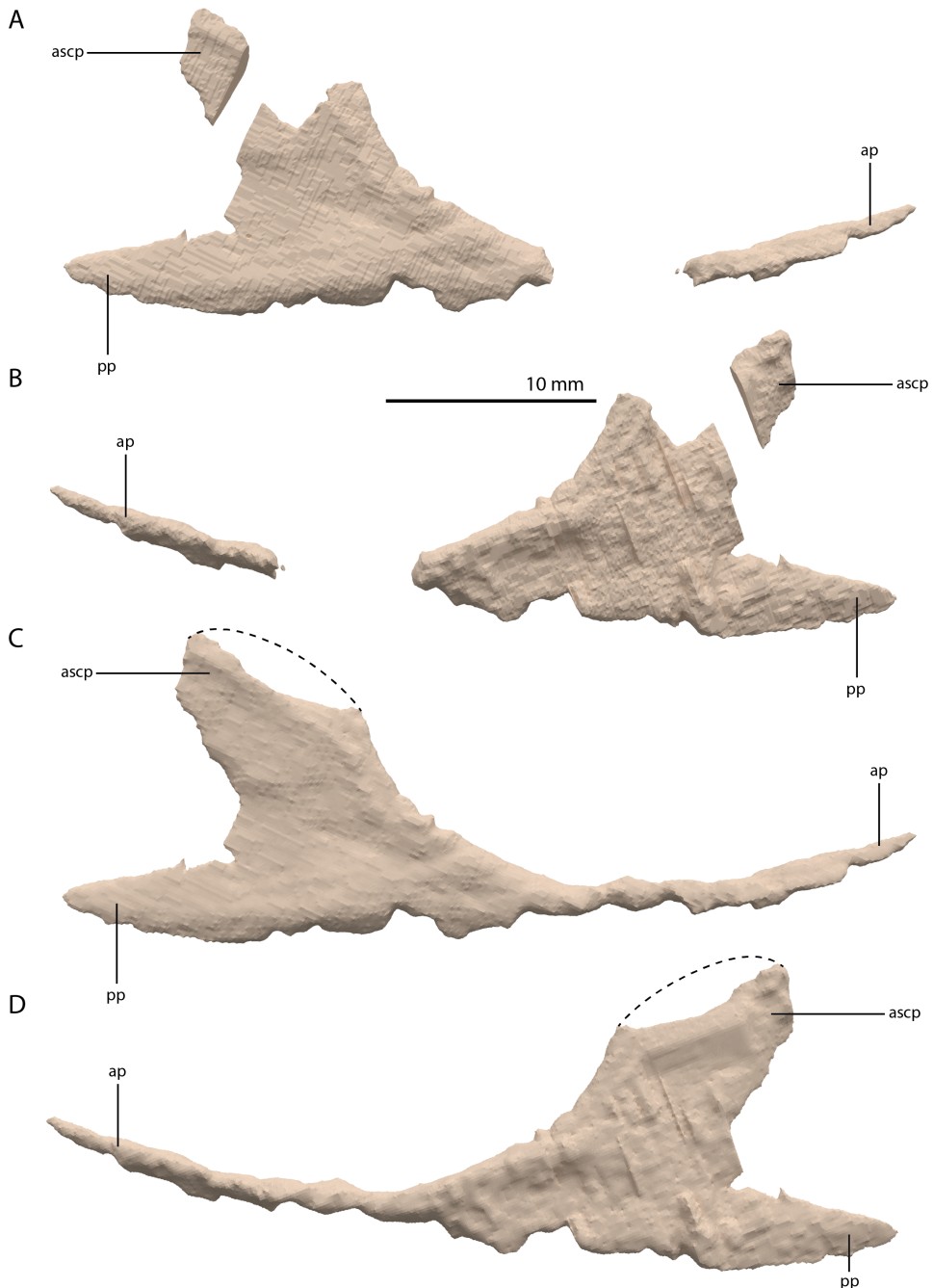

**Figure 12 Digital reconstruction of the right jugal of PIMUZ T 2790.** (A) Incomplete jugal as visible in the SRμCT data in lateral view. (B) Incomplete jugal as visible in the SRμCT data in medial view. (C) Jugal with missing portions reconstructed in lateral view. (D) Jugal with missing portions reconstructed in medial view. The stippled line indicates the dorsal margin of the ascending process as inferred from the well-preserved jugal of the *Tanystropheus hydroides* specimen PIMUZ T 2819. Abbreviations: ap, anterior process; ascp, ascending process; pp, posterior process.

process connect to the quadrate or quadratojugal, and the presence of a complete infratemporal bar is considered a synapomorphy of Archosauriformes (*Pinheiro, Simão-Oliveira & Butler, 2019*). *Trilophosaurus buettneri* represents an exception in that the infratemporal fenestra is completely absent in this taxon (*Spielmann et al., 2008*). The ascending process is somewhat posterodorsally orientated. The posterior margin of the ascending process formed the anterior margin of the infratemporal fenestra (Figs. 4 and 6A). Although the dorsal margin in PIMUZ T 2790 is absent, the complete jugal of PIMUZ T 2819 indicates that it was convex. This margin of the ascending process connected to the posteroventral margin of the postorbital along its entire length. At its base it fitted into the concave articulation facet on the ventral process of the postorbital. The configuration of the postorbital region indicates that the dorsal tip of the ascending process of the jugal connected to the anteroventrally expanded anterior process of the squamosal (Figs. 4 and 6A). Together with the postorbital, these three bones formed a wide postorbital bar, and the infratemporal fenestra was consequently small. A wide postorbital bar is also present in *Dinocephalosaurus orientalis* (IVPP-V13767), *Pectodens zhenyuensis* (IVPP-V18578), and *Jesairosaurus lehmani* (ZAR 06). The postorbital bar of *Azendohsaurus madagaskarensis* appears to be somewhat wider than that seen in most archosauromorphs, but less so than in the abovementioned taxa (*Flynn et al., 2010*). In contrast, other non-archosauriform archosauromorphs show a slender postorbital bar (e.g. *Langobardisaurus pandolfii*, MFSN 1921; *Macrocnemus bassanii*, *Miedema et al., 2020*; *Fuyuansaurus acutirostris*, IVPP-V17983; *Protorosaurus speneri*, *Gottmann-Quesada & Sander, 2009*; *Prolacerta broomi*, *Modesto & Sues, 2004*; *Mesosuchus browni*, *Dilkes, 1998*). The exact configuration of the postorbital bar in *Tanystropheus longobardicus* is unclear since no well-preserved squamosal is currently known for this species (*Spiekman et al., 2020*, Methods S1).

*Squamosal*

Both squamosals are preserved. The right one is complete, whereas the left element is largely complete but missing the end of the medial process and its anterior process is badly broken. The right squamosal is located underneath the right frontal and directly anterior to the right posterolateral process of the fused parietals. The left squamosal is surrounded by the left postorbital and the left anterolateral process of the fused parietals posteriorly, the left quadrate ventrally and anteriorly, and the left frontal dorsally. The overall shape of the squamosal is that of a curved plate-like bone formed by an anteriorly and a medially directed process (Fig. 13). The anteriorly directed process is dorsoventrally tall, especially anteriorly, where it formed a broad suture with the postorbital and almost certainly contacted the ascending process of the jugal ventrally. On the lateral surface of the anterodorsal tip, a clear triangular-shaped facet received the posterior process of the postorbital. The shape of the facet indicates that ventral to it, the anterior margin of the squamosal was partially covered by the postorbital in lateral view. In most archosauromorphs, including the tanystropheid *Macrocnemus bassanii*, the anterior portion of the squamosal can be distinguished into a discrete anterior process, which articulates with the posterior process of the postorbital, and a ventral process which

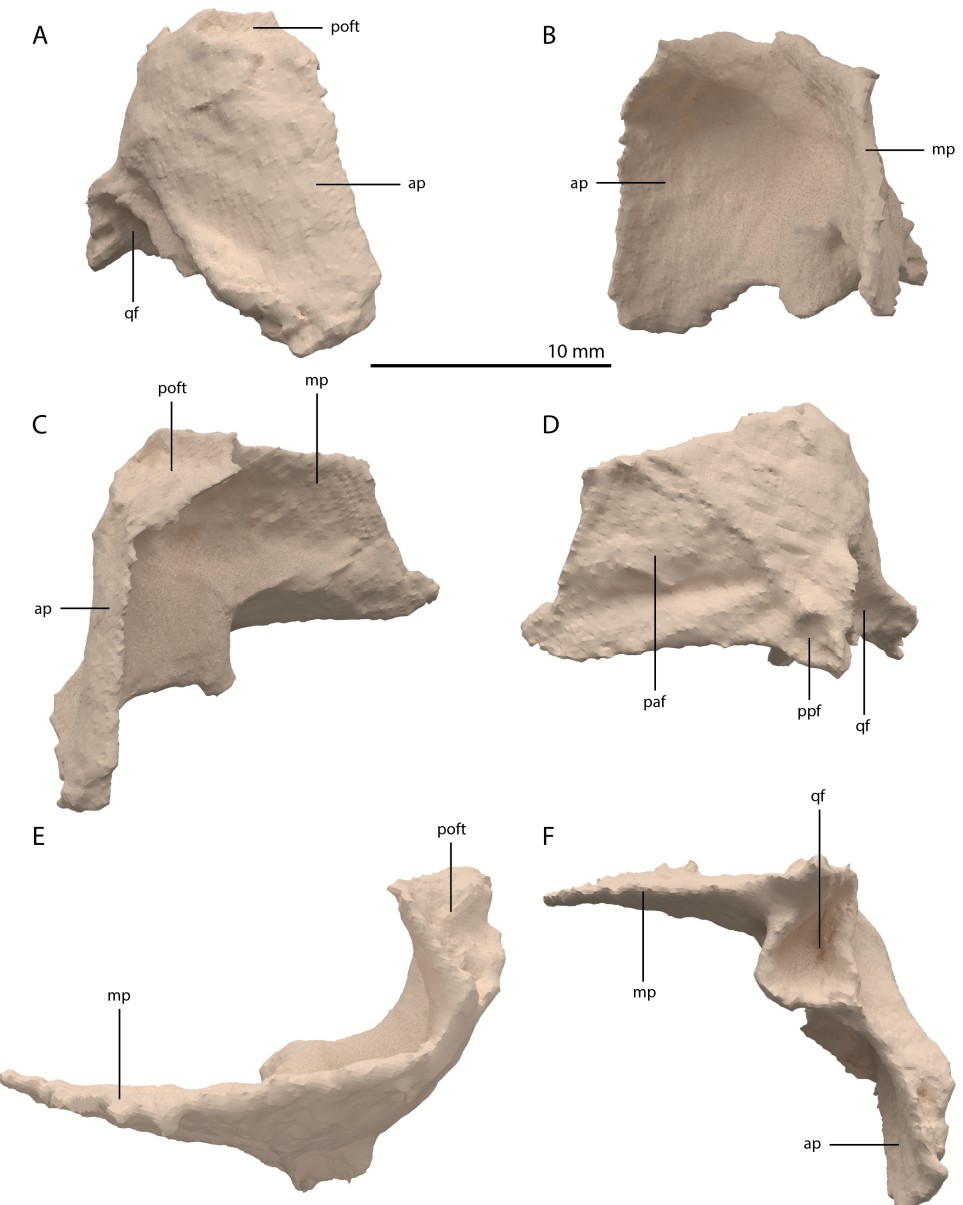

**Figure 13 Digital reconstruction of the right squamosal of PIMUZ T 2790.** (A) Lateral view. (B) Medial view. (C) Anterior view. (D) Posterior view. (E) Dorsal view. (F) Ventral view. Abbreviations: ap, anteriorly directed process; mp, medial process; paf, parietal facet; poft, postorbital facet; ppf, paroccipital process facet; qf, quadrate facet.

is located directly anterior to the quadrate (*Dilkes, 1998*; *Miedema et al., 2020*; *Modesto & Sues, 2004*). The anteriorly directed process of the right squamosal of PIMUZ T 2790 is certainly complete but does not exhibit two distinct processes. Instead, it seems most likely that the large plate-like anterior portion of the squamosal is homologous to both the anterior and ventral processes present in *Macrocnemus bassanii* and that these processes have become confluent in *Tanystropheus hydroides*. Although distinctly less tall than the anteriorly directed process, the medial process of the squamosal is also flat and

dorsoventrally tall. The posterior side of its distal half bears a large facet for the posterolateral process of the parietal (Fig. 13D). In dorsal view, the angle formed between the anterior and medial process is approximately 90 degrees (Fig. 13E). On its posteroventral side the squamosal bears a peculiar articular facet. This facet would have accommodated the dorsal head of the quadrate. It forms a large, very deep, and roughly pyramidal concavity. Its medial and lateral margins are raised, the former of which in particular forms a distinct ridge. The location and shape of this facet differs distinctly from that of *Macrocnemus bassanii* and *Prolacerta broomi*. In these taxa this socket has a similar shape to that in a ball and socket joint, and it is formed on the ventral side of the posterior process of the squamosal (*Miedema et al., 2020*; *Modesto & Sues, 2004*). A posterior process of the squamosal is absent in *Tanystropheus hydroides*. Directly medial to the quadrate facet, a small concavity is located on the posterior surface of the squamosal, which might represent an articulation facet of the distal end of the paroccipital process of the opisthotic (Fig. 13D). Directly anterior or lateral to the quadrate facet another anteroventrally orientated concavity is present, which is demarcated anteriorly by a low ridge on its ventral part.

*Quadrate*
The right quadrate is broken and only partially preserved to the right of the right frontal on the external surface of the specimen. The left quadrate, however, is very well-preserved and complete apart from the dorsolateral tip, which is broken off (Fig. 14). It is located underneath the left frontal, the left squamosal, and the quadrate ramus of the left pterygoid. The shaft is slightly sigmoidal in lateral view as the posterior margin is clearly concave on its dorsal portion and a straight to slightly convex on its ventral part (Fig. 14A). From the shaft, a very thin but wide pterygoid ramus is extended anteromedially. Its dorsal margin extends horizontally from the base of the dorsal head of the quadrate and forms a 90-degree angle with the medial margin. The medial margin is straight along its dorsal third before gradually but continuously decreasing in width ventrally until it terminates at the base of the ventromedial condyle (=entocondyle) of the quadrate. The surface of the pterygoid ramus bears a distinct fossa seen in posterior view, which results in an equally distinct convexity in anterior view. The morphology of the pterygoid ramus is similar in overall shape and orientation to that of the best-known quadrate of *Tanystropheus longobardicus*, preserved in PIMUZ T 2484 (Fig. 15). However, the ramus is considerably shorter comparatively in *Tanystropheus longobardicus*, and the presence of the fossa cannot be established due to the small size and compression of the specimen. The pterygoid ramus differs strongly from the short anteriorly directed ramus of *Macrocnemus bassanii* (*Miedema et al., 2020*), but shows similarities to the pterygoid ramus of *Prolacerta broomi* (BP/I/5066) and possibly *Protorosaurus speneri* (*Gottmann-Quesada & Sander, 2009*), although clear observation for *Protorosaurus speneri* is considerably hampered by the flattening of NMK S 180, the only known specimen exhibiting this feature. The dorsal end of the shaft of the quadrate of PIMUZ T 2790 bears a very conspicuous posteroventrally directed hook (Fig. 14). A hooked dorsal head of the quadrate is known for the allokotosaurs *Pamelaria dolichotrachela*,

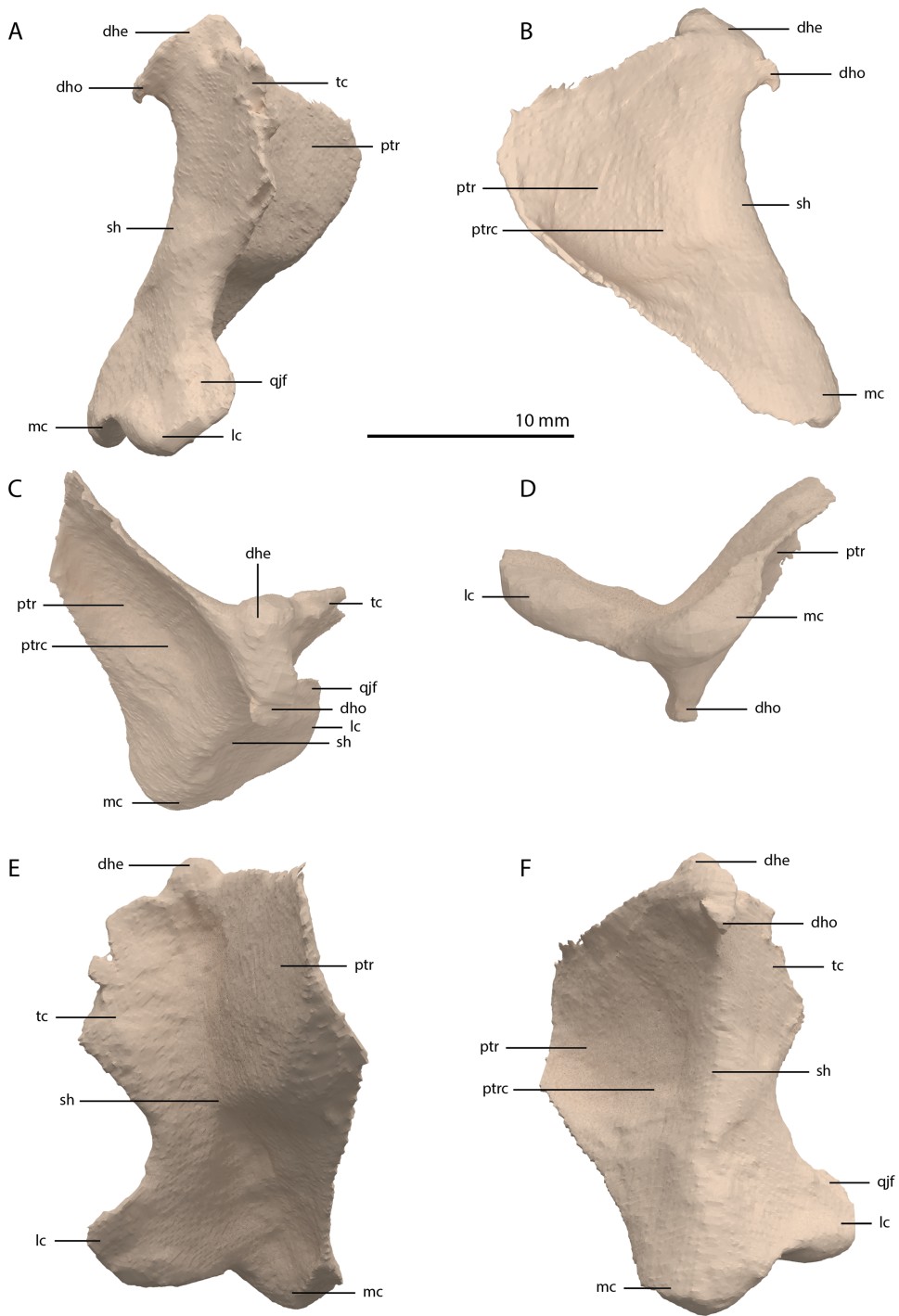

**Figure 14 Digital reconstruction of the right quadrate of PIMUZ T 2790.** (A) Lateral view. (B) Medial view. (C) Dorsal view. (D) Ventral view. (E) Anterior view. (F) Posterior view. Abbreviations: dhe, dorsal head; dho, dorsal hook; lc, lateral condyle; mc, medial condyle; ptr, pterygoid ramus; ptrc, pterygoid ramus concavity; qjf, quadratojugal facet; sh, shaft; tc, tympanic crest.

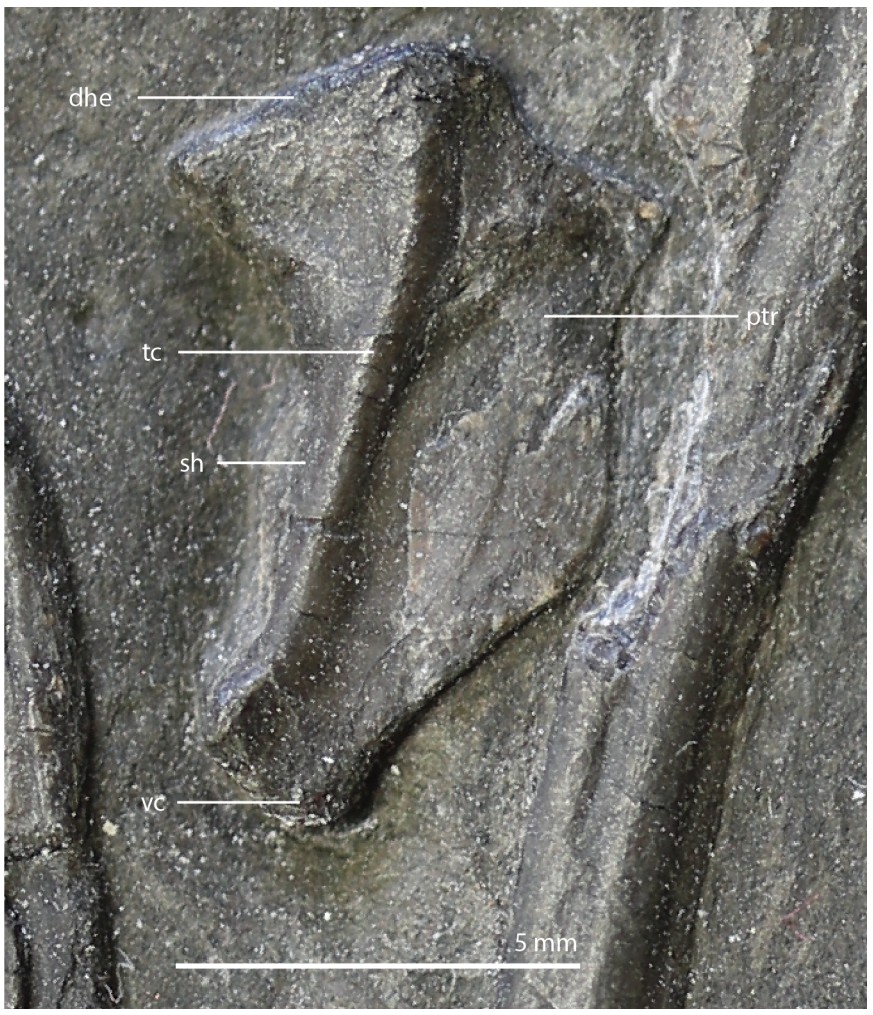

**Figure 15 The right quadrate of *Tanystropheus longobardicus* specimen PIMUZ T 2484 in lateral view, revealing a morphology similar to that of *Tanystropheus hydroides*.** Abbreviations: dhe, dorsal head; ptr, pterygoid ramus; sh, shaft; tc, tympanic crest; vc, ventral condyle.

*Shringasaurus indicus*, and *Azendohsaurus madagaskarensis* but in these taxa this hook is not as conspicuous as in *Tanystropheus hydroides* (*Flynn et al., 2010*; *Sen, 2003*; *Sengupta, Ezcurra & Bandyopadhyay, 2017*). Although not hooked as in the abovementioned taxa, the dorsal head of the quadrate is also posteroventrally expanded in *Tanystropheus longobardicus* (Fig. 15). Anterolateral to this hook, the majority of a short tympanic crest is located, which has also been identified in certain rhynchosaurs (e.g. *Mesosuchus browni*, *Dilkes, 1998*), allokotosaurs (e.g. *Azendohsaurus madagaskarensis*, *Flynn et al., 2010*), *Prolacerta broomi* (*Modesto & Sues, 2004*), and *Macrocnemus bassanii* (*Miedema et al., 2020*). Ventral to the tympanic crest the quadrate is constricted before widening laterally towards the ventrolateral condyle (=ectocondyle). A quadrate foramen was previously identified for both *Tanystropheus longobardicus* and *Tanystropheus hydroides* (as the large morphotype of *Tanystropheus longobardicus* in *Wild, 1973*). However, such a foramen is absent in PIMUZ T 2790. We were also not able to corroborate the presence of this

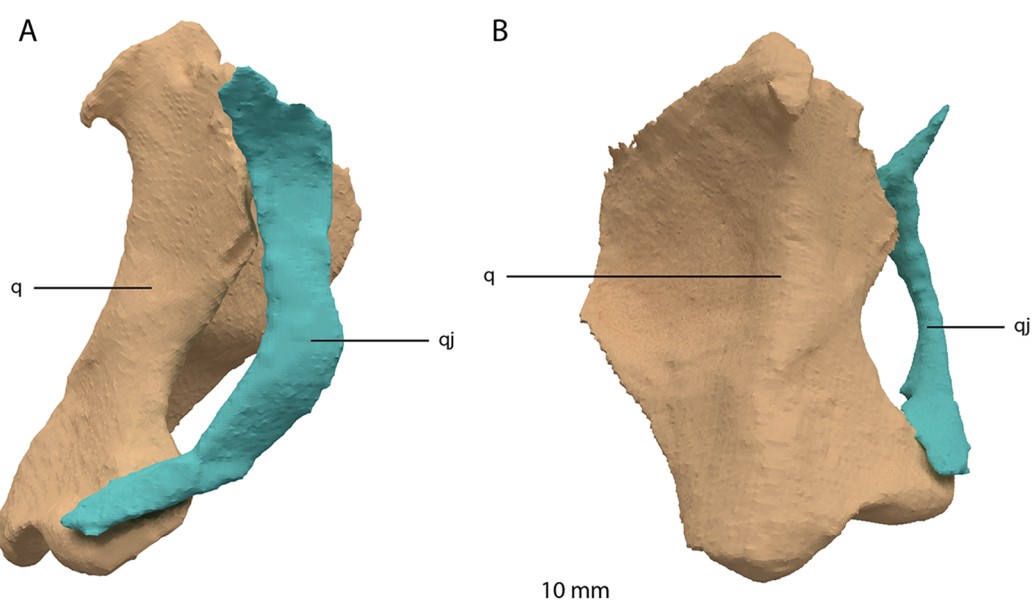

**Figure 16 Articulated digital reconstruction of the right quadrate and quadratojugal of PIMUZ T 2790.** (A) Lateral view. (B) Medial view. Abbreviations: q, quadrate; qj, quadratojugal.

foramen in the specimens in which it was considered to be present, PIMUZ T 2484 for *Tanystropheus longobardicus*, and PIMUZ T 2787 for *Tanystropheus hydroides*. This foramen was therefore absent in *Tanystropheus hydroides*, and likely also absent in *Tanystropheus longobardicus*. Both ventral condyles of the quadrate are rounded and are separated by a concavity (Fig. 14F). The lateral condyle (=ectocondyle) is wider than the medial condyle (=entocondyle), whereas the medial condyle projects further ventrally than the lateral one, as is also the case in *Macrocnemus bassanii* and the allokotosaurs *Pamelaria dolichotrachela* and *Azendohsaurus madagaskarensis* (*Flynn et al., 2010*; *Miedema et al., 2020*; *Sen, 2003*). The medial condyle would have articulated with the glenoid fossa of the articular. The skull reconstruction reveals that the quadrate was orientated somewhat posteroventrally to anterodorsally, as well as lateroventrally to mediodorsally (Figs. 4–6). This angled orientation of the quadrate is also known and considerably more pronounced in *Proterosuchus fergusi* (*Ezcurra & Butler, 2015*). The dorsolateral surface of the lateral condyle bears a faint, somewhat rectangular-shaped facet (Fig. 14A). Here, the ventral footplate of the quadratojugal would have attached to the quadrate (Fig. 16).

*Quadratojugal*
Two small, curved and rod-shaped elements are identified as the quadratojugals, which were previously considered to be absent in both *Tanystropheus hydroides* and *Tanystropheus longobardicus* (*Nosotti, 2007*; *Wild, 1973*). The left quadratojugal is located directly anterior to the left prefrontal and dorsal to the left surangular, whereas the right quadratojugal is located anterolaterally to the poorly preserved right quadrate and lateral to the posterior part of the right mandibular ramus. The quadratojugal is a flattened, rod-like bone with a helical curvature (Fig. 16). The ventral end is thin and would have

articulated on the dorsolateral surface of the lateroventral condyle of the quadrate. The dorsal end articulated with the squamosal and possibly the laterodorsal part of the quadrate. Because of the curvature of the bone, the articular surface of the dorsal end almost faces in the direct opposite direction of the ventral articulation. There is no anterior process of the quadratojugal and it therefore did not connect to the jugal (Fig. 4). This corresponds largely to the configuration seen in many early archosauromorphs (e.g. *Macrocnemus bassanii*, *Mesosuchus browni* and *Prolacerta broomi*; *Dilkes, 1998*; *Miedema et al., 2020*; *Modesto & Sues, 2004*), in which the quadratojugal is also curved and has a similar position relative to the quadrate. However, the quadratojugal of these taxa appear to lack the helical or twisting curvature present in *Tanystropheus hydroides*. The morphology of the quadratojugal of allokotosaurs differs distinctly from that of other early archosauromorphs, including *Tanystropheus hydroides*. In *Azendohsaurus madagaskarensis* and *Teraterpeton hrynewichorum* the quadratojugal is roughly straight (*Flynn et al., 2010*; *Sues, 2003*), whereas in *Trilophosaurus buettneri* the infratemporal fenestra is completely missing and the quadratojugal possibly had a triangular shape (*Spielmann et al., 2008*).

*Vomer*
Both vomers are fragmentary and are surrounded by the mandible, premaxillae, and maxillae. The tooth bearing outer margins of both bones are intact, but most of their medial surfaces are lost, probably because they were exposed on the surface of the specimen during excavation and preparation (Fig. 17). There are 15 alveoli preserved on the right vomer and 14 on the left. The vomers were very thin and only thickened around the tooth bearing lateral margin. They were wide and enclosed the palate anteriorly and laterally and restricted the internal choanae to relatively narrow openings (Fig. 5B). The morphology of the vomers corresponds to that of the well-preserved vomers of PIMUZ T 2787 (see figure 4G of *Spiekman & Scheyer, 2019*). The vomers of *Dinocephalosaurus orientalis* are likely equally broad as those of *Tanystropheus hydroides*, but were probably edentulous (*Rieppel, Li & Fraser, 2008*). The vomers of other early archosauromorphs, including *Tanystropheus longobardicus*, are generally much narrower and bear one or more rows of small teeth (*Dilkes, 1998*; *Flynn et al., 2010*; *Miedema et al., 2020*; *Modesto & Sues, 2004*; *Spiekman et al., 2020*). Therefore, the vomeral morphology of *Tanystropheus hydroides* appears to be unique among early archosaurmorphs, and the large recurved teeth along the lateral margin of the bone likely represent a feeding adaptation.

*Palatine*
A plate-like bone is preserved directly anteroventral to the left frontal and dorsal to the transverse flange of the left pterygoid. It is incomplete, with the straight medial margin being the only complete margin of the element (Fig. 18). Based on its position in the specimen and overall shape, which is in correspondence with the palatines of PIMUZ T 2787 (see figure 4G of *Spiekman & Scheyer, 2019*), it is tentatively identified as the left palatine. No right palatine could be identified. The element is edentulous, thin, roughly flat, and anteroposteriorly longer than transversely wide.

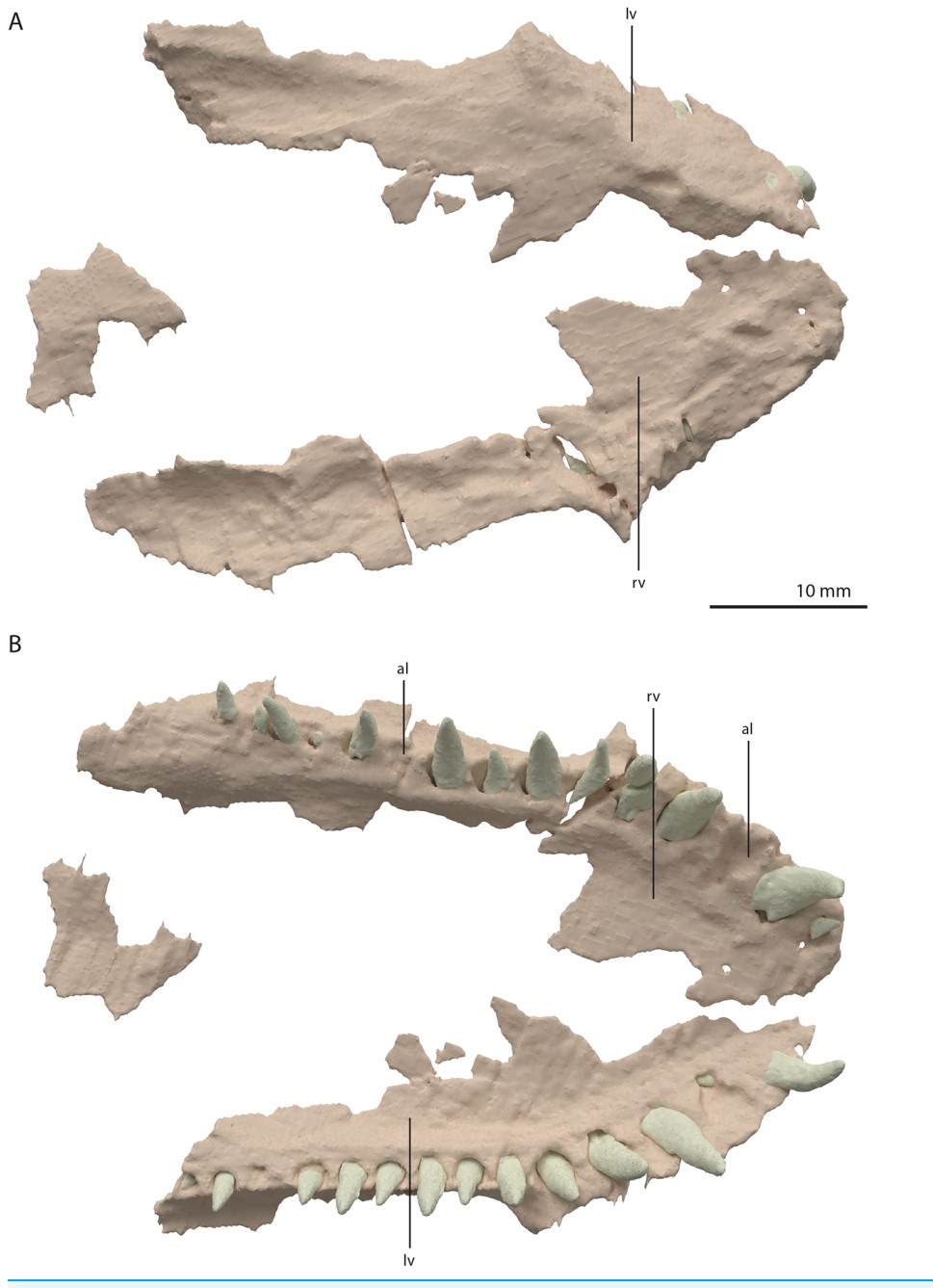

**Figure 17 Digital reconstruction of the vomers of PIMUZ T 2790.** (A) Dorsal view. (B) Ventral view. Abbreviations: al, alveolus; lv, left vomer; rv, right vomer.

*Ectopterygoid*

The left ectopterygoid could not be identified, but directly to the right of the parabasisphenoid and anterior to the right quadrate, an element is preserved that is tentatively identified as a complete right ectopterygoid (Fig. 19). This element is distinctly different from the ectopterygoid seen in other archosauromorphs (e.g. *Azendohsaurus madagaskarensis*, *Mesosuchus browni*, and *Macrocnemus bassanii*; *Dilkes, 1998*;

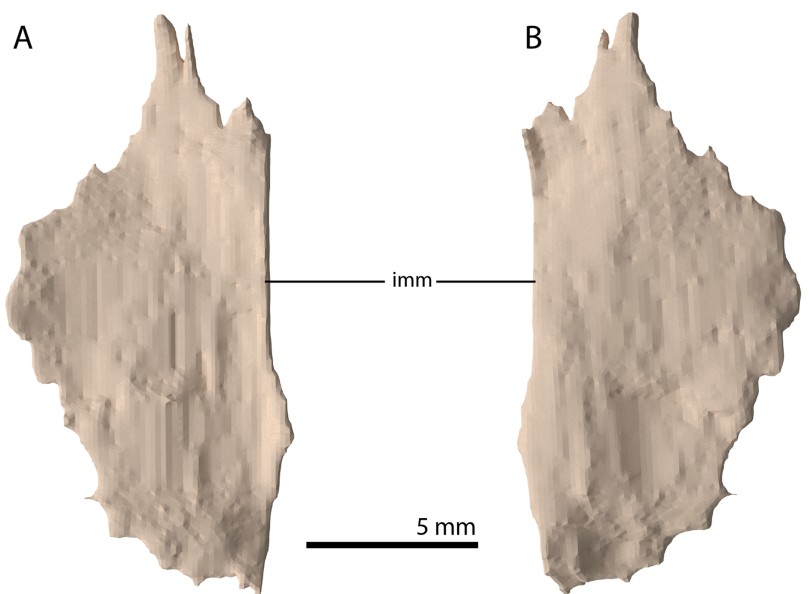

**Figure 18 Digital reconstruction of the putative left palatine of PIMUZ T 2790.** (A) Dorsal view. (B) Ventral view. Abbreviation: imm, intact medial margin.

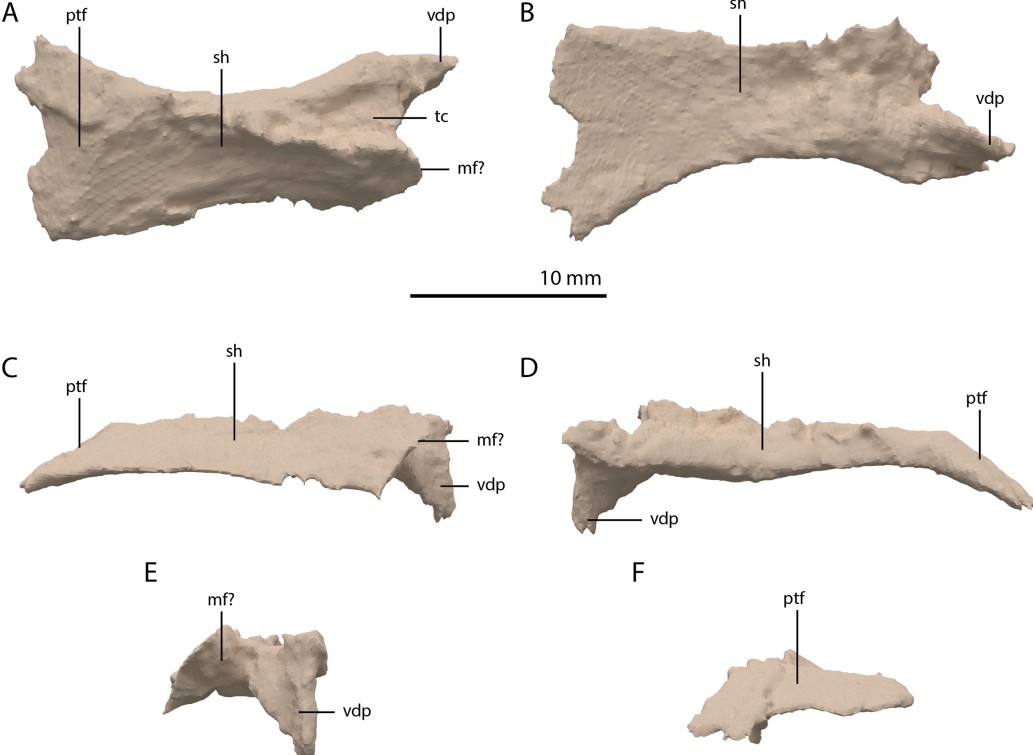

**Figure 19 Digital reconstruction of the element tentatively interpreted as the right ectopterygoid of PIMUZ T 2790.** (A) Dorsal view. (B) Ventral view. (C) Anterior view. (D) Posterior view. (E) Lateral view. (F) Medial view. Abbreviations: mf, maxilla facet; ptf, pterygoid facet; sh, shaft; tc, triangular concavity; vdp, ventrally deflected process.

*Flynn et al., 2010*; *Miedema et al., 2020*). Nevertheless, the element is identified as an ectopterygoid due to its relative position in the skull and its shape and size, which allow it to articulate with the pterygoid and maxilla in the 're-assembled' skull model (Figs. 5B and 6C). The element is a dorsoventrally flattened bone with a plate-like shaft. Its medial end is flattened and curved ventrally (Fig. 19D). No clear articulation facet with the pterygoid is present and the ectopterygoid would have overlapped considerably with the pterygoid ventrally. It would thus have formed a loose and possibly movable connection to the pterygoid just anterior to its transverse flange at the posterior part of the palatal ramus. The anterior or medial margin of the ectopterygoid shaft is gently concave and somewhat thickened, whereas the posterior or lateral margin is straight and thin. On the lateral end of the dorsal surface of the shaft a triangular concavity is located anteriorly (Fig. 19A). If our interpretation is correct, this facet would have received the posterior part of the palatine. The lateral margin of the ectopterygoid is formed by a small flat surface that would have articulated with the medial side of the posterior process of the maxilla (Fig. 19E). Anterior or lateral to this, the ectopterygoid is projected slightly further anteriorly to form a ventrally deflected process of unknown function.

*Pterygoid*

Both pterygoids are preserved and are located below and anterior to the frontals. Both are largely complete, with only the anterior third of the bones being broken and partially missing. Both pterygoids are completely edentulous. In this, and in their overall shape, the pterygoids conform to the morphology of the pterygoids in PIMUZ T 2787 (see figure 4D of *Spiekman & Scheyer, 2019*). The right pterygoid is slightly more complete than the left (Fig. 5B). Even though its anterior part is broken, it is clear that the palatal ramus (=anterior process) of the pterygoid of PIMUZ T 2790 is wide along its entire length. In this feature and the complete absence of the pterygoid teeth, *Tanystropheus hydroides* differs from all other early archosauromorphs for which the pterygoid is known, including *Tanystropheus longobardicus*, but with the possible exception of *Dinocephalosaurus orientalis* (*Rieppel, Li & Fraser, 2008*; *Spiekman et al., 2020*). The shape and size of the palatal rami suggests that the pterygoids contacted each other anteriorly. The pterygoid is concave in the transverse plane, with a concavity in the centre of the bone and a somewhat dorsally inclined lateral portion (Figs. 20E and 20F). The pterygoid is similarly concave in the sagittal plane, with the pterygoid being the lowest at the level of the transverse process and the palatal and quadrate rami being slightly inclined dorsally (Figs. 20C and 20D). The lateral surface of the transverse flange is distinctly rugose and dorsoventrally thickened (Fig. 20C). This surface is orientated posteroventrally to anterodorsally. The angle between the anterior and lateral margins of the transverse flange is roughly right-angled, whereas that between the posterior and lateral margins is acute. At the base of the quadrate ramus, the articulation facet for the basipterygoid process of the parabasisphenoid can be clearly made out on the medial surface (Fig. 20D). It is framed by a dorsally directed upper lip and a medially directed lower lip. Directly anterior to this facet a concavity is present, which might have facilitated the articulation of the ventral foot plate of the epipterygoid. However, a clear articulation surface cannot be discerned.

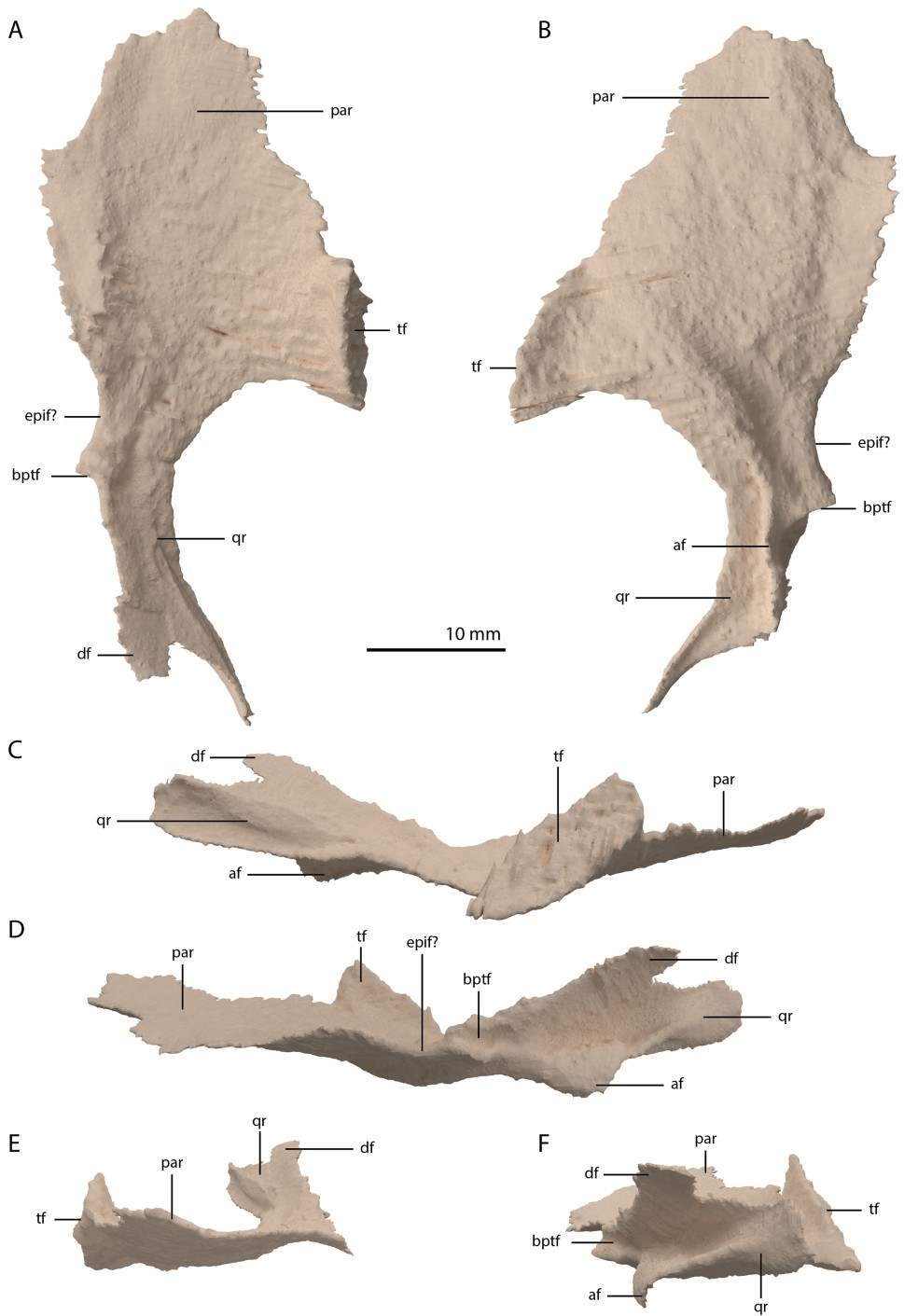

**Figure 20 Digital reconstruction of the right pterygoid of PIMUZ T 2790.** (A) Dorsal view. (B) Ventral view. (C) Lateral view. (D) Medial view. (E) Anterior view. (F) Posterior view. Abbreviations: af, arcuate flange; bptf; basipterygoid facet; df, dorsal flange; epif, epipterygoid facet; par, palatal ramus; qr, quadrate ramus; tf, transverse flange.

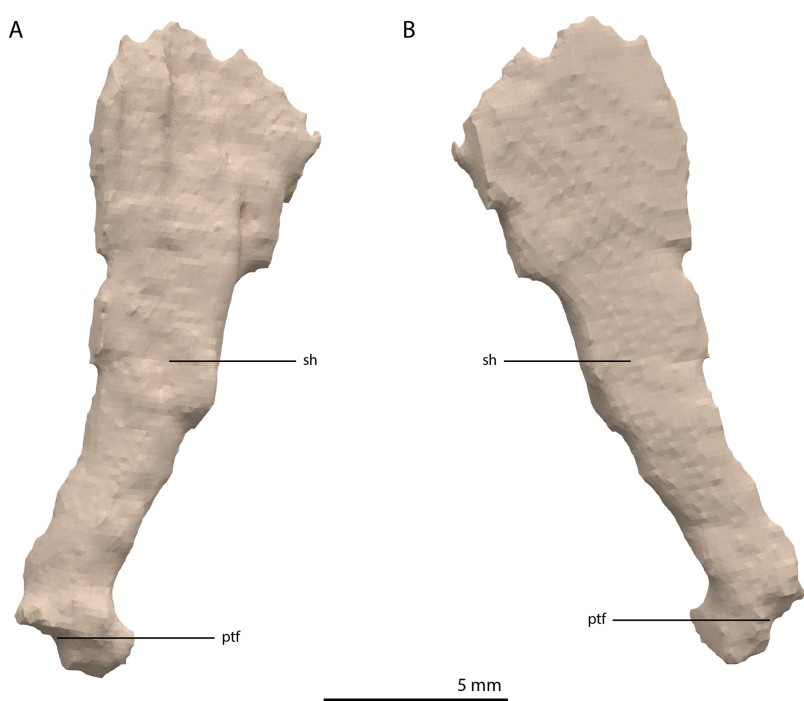

**Figure 21 Digital reconstruction of the right epipterygoid of PIMUZ T 2790.** (A) Lateral view. (B) Medial view. Abbreviations: ptf, pterygoid facet; sh, shaft.

The quadrate ramus has a posterolateral orientation and is somewhat dorsally inclined. From the main ramus project two thin flanges, a dorsomedially orientated dorsal flange and a ventromedially orientated arcuate flange (sensu *Ford & Benson, 2018*). The dorsal flange is larger and reaches further posteriorly along the ramus (Figs. 20C and 20D). The dorsal flange is straight whereas the arcuate flange is curved ventrally and even slightly laterally at its distal end (Fig. 20F). The dorsomedial orientation of the dorsal flange seems to indicate it did not directly contact the pterygoid wing of the quadrate. Anterior to the quadrate ramus, the arcuate flange transitions into a low ridge that is anterolaterally orientated and almost reaches the transverse flange (Fig. 20B).

*Epipterygoid*
Both epipterygoids are preserved. The left element is located in between the left anterolateral process of the fused parietals and the left postorbital. The right is preserved directly posterior to the ascending process of the right jugal. Both are complete, except for the middle part of the shaft of the left element, which is broken. The epipterygoid is a lateromedially flattened, columnar bone that has a gradual anteroposterior expansion towards its dorsal end and a more abrupt expansion on its ventral end (Fig. 21). The expansion on the dorsal end is larger than on the ventral end. The ventral end is rounded with a ventrolaterally facing flat surface that likely connected to the pterygoid directly anterior to the facet for the basipterygoid process on this bone. The shape of the epipterygoid differs from that described for *Macrocnemus bassanii*, in which the shaft

bears a distinct posterior expansion, the shaft has an oval rather than flattened cross-section, and in which the epipterygoid is not expanded dorsally (*Miedema et al., 2020*).

*Basioccipital*

The basioccipital is located below the anterior part of the right frontal and anterior to the two fused braincase elements. This element, which forms the posteroventral part of the braincase (Fig. 22), is largely complete and distinctly deformed from left to right in posterior view (Fig. 23). Nevertheless, the original morphology of the bone can still be inferred. The occipital condyle contribution of the basioccipital is ventrally concave at its base. The dorsal surface of the occipital condyle contribution bears two large dorsolaterally directed, concave facets for the articulation of the condylar contributions of the exoccipitals. As can be seen from the morphology of the exoccipitals, they contributed substantially to the occipital condyle and they possibly even excluded the basioccipital from contributing to the floor of the foramen magnum (Figs. 22E and 23C), which was possibly also the case for *Tanystropheus longobardicus* (PIMUZ T 2484). The combined shape of the exoccipitals and the basioccipital gave the condyle a hemispherical shape (Fig. 22E). Anterior to the occipital condyle on the ventral surface, two ridges run from the occipital neck ventrolaterally towards the basal tubera of the basioccipital (Fig. 23A). The surface between these ridges is concave. The surfaces lateral to these ridges are also concave and face posterolaterally. The basal tubera of the basioccipital are rounded and more medially located than the basal tubera of the parabasisphenoid. A transverse ridge that is slightly depressed in its centre connects the basal tubera. Such a ridge is absent in *Macrocnemus bassanii* and *Prolacerta broomi* but common in allokotosaurs and non-archosaurian archosauriforms (*Evans, 1987*; *Ezcurra, 2016*; *Flynn et al., 2010*; *Sen, 2003*). In contrast to *Euparkeria capensis* (*Sobral et al., 2016*), posterior to this ridge, the contribution of the basioccipital to the median pharyngeal recess appears to be minimal. On the anterior surface of the basioccipital indentations are present that are similar to those recently described for *Macrocnemus bassanii* (Fig. 23B; *Miedema et al., 2020*). However, the compression of the bone has distorted these structures, preventing a detailed description of their shape.

*Parabasisphenoid*

The para–and basisphenoid are fully fused into a single element, the parabasisphenoid, which forms the anteroventral portion of the braincase (Fig. 22). It is preserved anterior to the right frontal and the right quadrate. It is virtually complete and somewhat deformed from right to left in anterior view. The cultriform process is long and straight and tapers to a sharp point anteriorly (Fig. 24). It is somewhat dorsoventrally constricted at its base as in *Prolacerta broomi* and *Euparkeria capensis* (*Evans, 1986*; *Sobral et al., 2016*) among other archosauromorphs, but from this constriction the cultriform process gains in height anteriorly until approximately one-third of its anteroposterior length (Figs. 24C and 24D). From there, it slowly decreases in dorsoventral height towards its anterior terminus. It bears a concave trough on its dorsal surface forming a V-shaped cross section (Fig. 24B). The basipterygoid processes are prominent but short and are facing

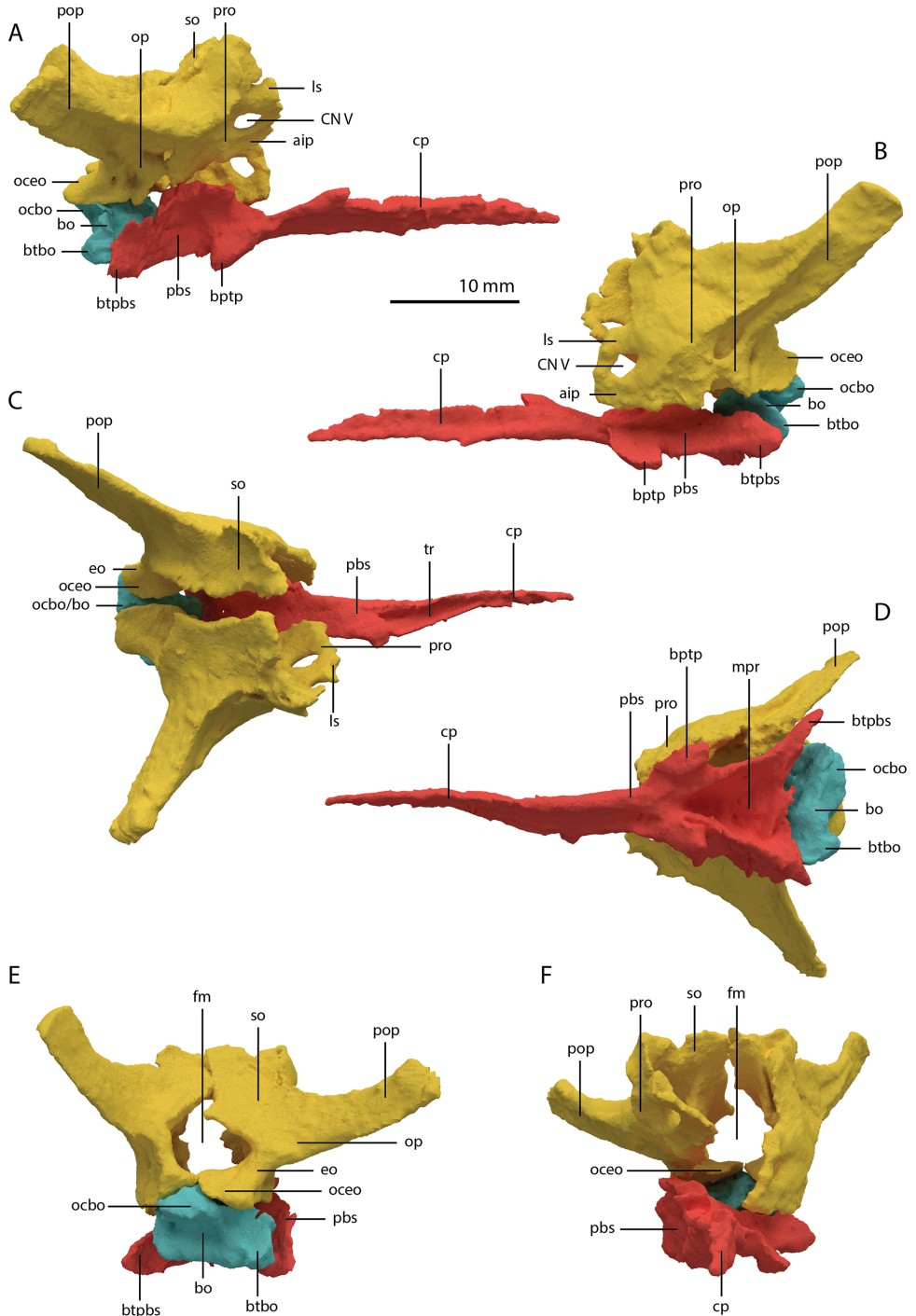

**Figure 22 Articulated digital reconstruction of the braincase of PIMUZ T 2790.** (A) Right lateral view. (B) Left lateral view. (C) Dorsal view. (D) Ventral view. (E) Posterior or occipital view. (F) Anterior view. Abbreviations: aip, anterior inferior process; bo, basioccipital; bptp, basipterygoid process; btbo, basal tuber basioccipital; btpbs, basal tuber parabasisphenoid; CN, cranial nerve; cp, cultriform process; eo, exoccipital; fm, foramen magnum; ls, laterosphenoid; mpr, median pharyngeal recess; ocbo, basioccipital contribution occiput; oceo, exoccipital contribution occiput; op, opisthotic; pbs, parabasisphenoid; pop, paroccipital process; pro, prootic; so, supraoccipital; tr, trough.

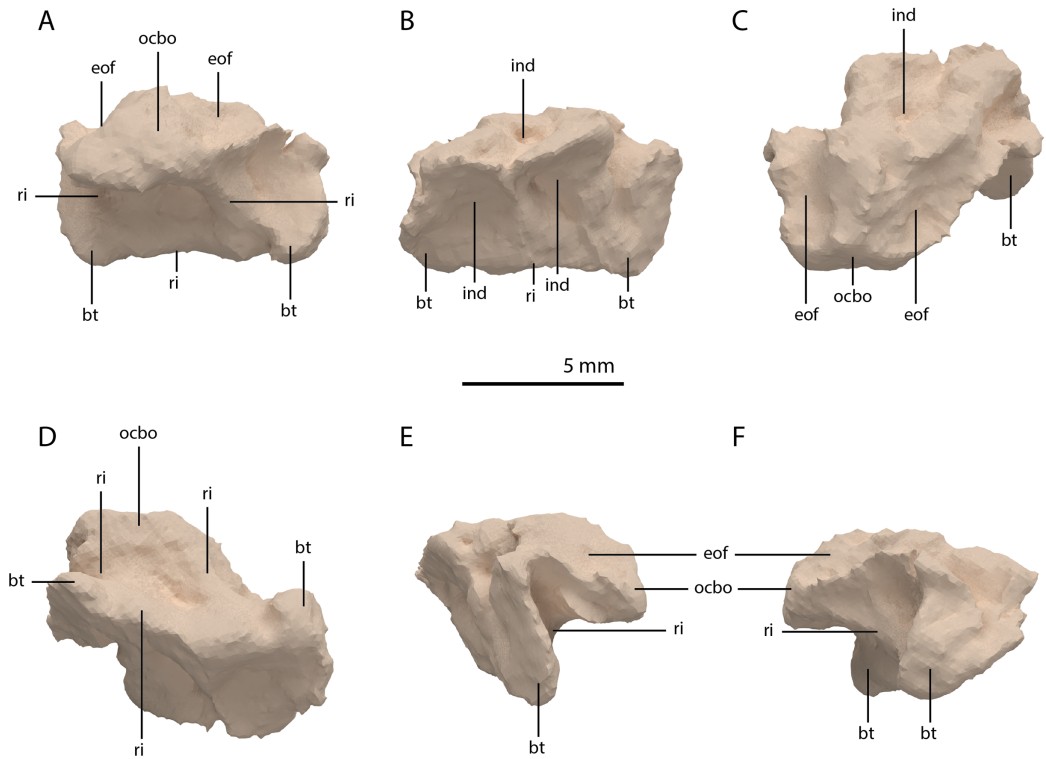

**Figure 23 Digital reconstruction of the basioccipital of PIMUZ T 2790.** (A) Posterior or occipital view. (B) Anterior view. (C) Dorsal view. (D) Ventral view. (E) Left lateral view. (F) Right lateral view. Abbreviations: bt, basal tuber; eof, exoccipital facet; ind, indentation; ocbo, basioccipital contribution occipital condyle; ri, ridge.

anterolaterally (Fig. 24A). Distinct, but thin, parasphenoid crests run along the ventral surface of the main body posterolaterally towards the basal tubera. No foramina were found on the ventral surface of the main body, but these could have been present but simply not visible in the SRμCT data, since they are generally present in early archosauromorphs (*Ezcurra, 2016*). The parasphenoid crests are connected by a deeply concave bony plate, the median pharyngeal recess (sensu *Nesbitt, 2011*). This character was originally identified in archosauriforms, but was recently determined to be present in the non-archosauriform archosauromorphs *Bentonyx sidensis* and *Azendohsaurus madagaskarensis* (*Ezcurra, 2016*). There was no intertuberal plate as in *Prolacerta broomi* and *Azendohsaurus madagaskarensis* (*Evans, 1986*; *Flynn et al., 2010*).

The entire parabasisphenoid has a horizontal orientation, similar to most early diapsids and non-archosauriform archosauromorphs, but in contrast to *Azendohsaurus madagaskarensis* and most early archosauriforms, in which the posterior portion of the parabasisphenoid is orientated anteroventrally (*Flynn et al., 2010*; *Gower & Sennikov, 1996*). The basal tubera of the parabasisphenoid are thin and open posteriorly (Fig. 24E). The contribution of the parabasisphenoid to the basal tubera is wider by comparison than that of the basioccipital contribution so that there may have been an open space in this region, the so-called pseudolagenar recess as described for several archosauriforms (*Gower & Sennikov, 1996*). However, because the elements are not preserved in

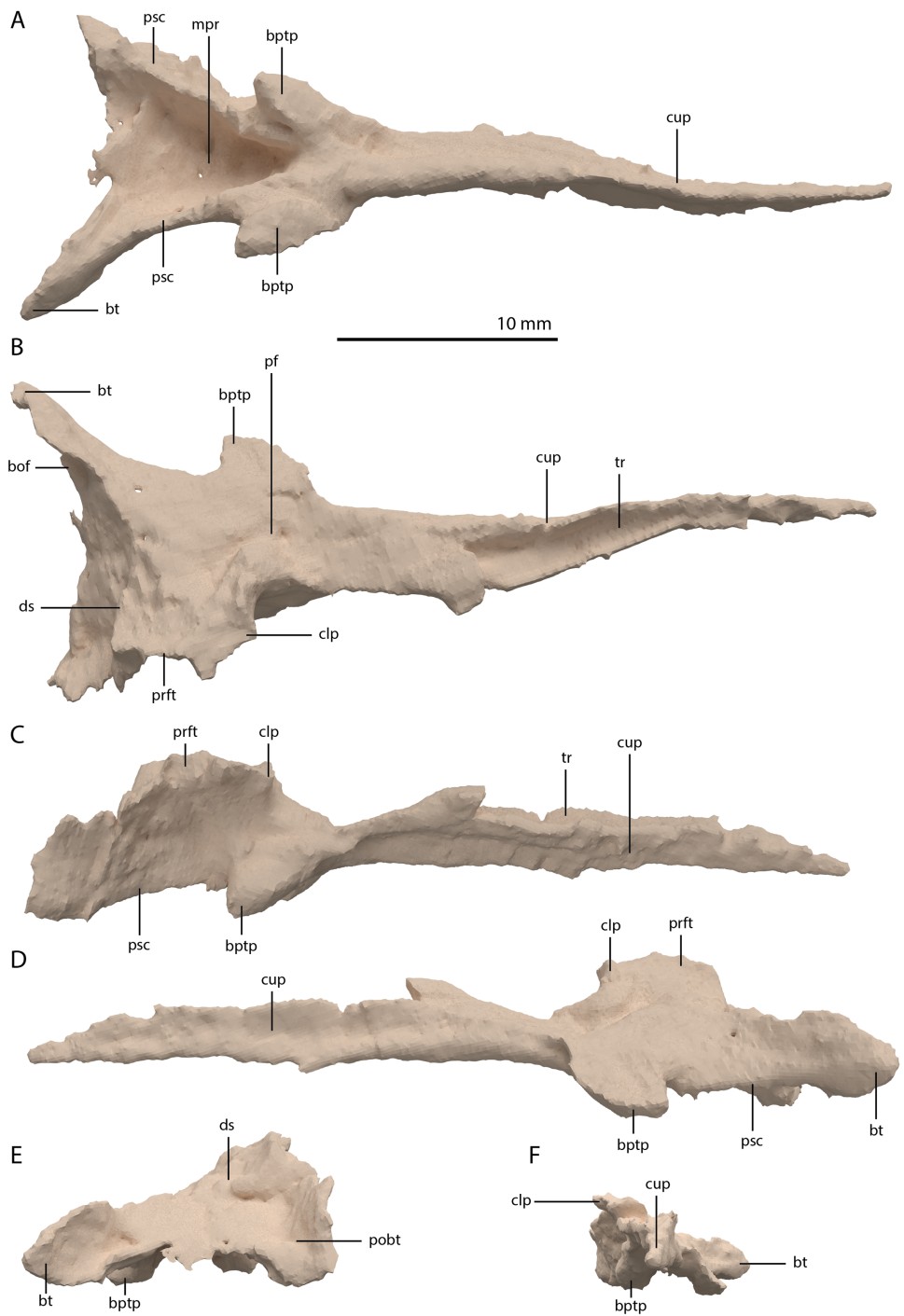

**Figure 24 Digital reconstruction of the parabasisphenoid of PIMUZ T 2790.** (A) Ventral view. (B) Dorsal view. (C) Right lateral view. (D) Left lateral view. (E) Posterior view. (F) Anterior view. Abbreviations: bof, basioccipital facet; bptp, basipterygoid process; bt, basal tuber; clp, clinoid process; cup, cultriform process; ds, dosum sella; mpr, median pharyngeal recess; pf, pituitary fossa; pobt, posterior opening basal tubera; prft, prootic facet; psc, parasphenoid crest; tr, trough.

articulation, this cannot be stated with certainty. There is no evidence for pneumatic foramina, as were recently discovered in *Mesosuchus browni* (*Sobral & Müller, 2019*). Of the dorsal portion of the parabasisphenoid, only the right side is largely preserved. A clinoid process with an anteriorly concave margin is located dorsal to the right basipterygoid process (Figs. 24C and 24D). Posterior to the clinoid process, the lateral margin remains tall at first, before gently sloping down ventrally towards the basal tubera. This, in combination with the morphology of the prootic, indicates that the contact between these two bones was continuous posterior to the clinoid process. Anteromedial to the clinoid process, a shallow concavity represents the pituitary fossa (=hypophyseal fossa; Fig. 24B). No foramina can be observed on the dorsal surface of the parabasisphenoid, but as for the ventral surface, their absence cannot be assumed. Posterior to the right clinoid process the dorsal surface of the parabasisphenoid is interrupted abruptly by the vertical slope of the dorsum sellae.

*Articulated Braincase elements*
The exoccipital, opisthotic, supraoccipital, prootic, and laterosphenoid were all preserved in full articulation. No sutures between these bones were discernible from the SRμCT data. The individual elements were instead distinguished based on their morphology. The observation of sutures in well-ossified archosauromorph braincases is often ambiguous (*Gower & Sennikov, 1996*; *Sobral et al., 2016*), and among non-archosauriform archosauromorphs, several braincase sutures between the exoccipital, opisthotic, supraoccipital, and prootic could also not be discerned in *Mesosuchus browni* despite the aid of μCT data (*Sobral & Müller, 2019*). The articulated braincase is split into two pieces along the midline. Both pieces are still closely associated and are located anterior to the fused parietals and right squamosal, posterior to the two pterygoids and ventral to the right frontal. The left piece is distorted, with the dorsal surface facing laterally, the exoccipital having been tilted slightly dorsally, and the prootic medially. The right piece is virtually undistorted, and therefore the description of the elements below is based on this side (Fig. 25).

*Exoccipital*
Ventrally the exoccipital bears a large flat ventromedially orientated surface, the occipital foot, that articulated on the dorsolateral surface of the basioccipital (Fig. 25). Although the foot extended far medially and posteriorly, the extent to which both exoccipitals may have touched each other ventrally is unclear. The ventrolateral and lateral margin of the foramen magnum was certainly formed by the exoccipitals. However, due to the lack of observable sutures, the dorsal extent of the exoccipitals cannot be determined. Anterior to the exoccipital foot, but not visible in occipital view, a small, ventrolaterally orientated foramen is present; the opening for the hypoglossal nerve (CN XII, Fig. 25E). Anterior to this, a larger oval-shaped opening, the metotic foramen, is present, which forms the passageway for the glossopharyngeal nerve, vagus nerve, and accessory nerve (CN IX, CN X, and CN XI, respectively). The metotic foramen is framed by the exoccipital posteriorly and the ventral ramus of the opisthotic anteriorly in archosauromorphs and

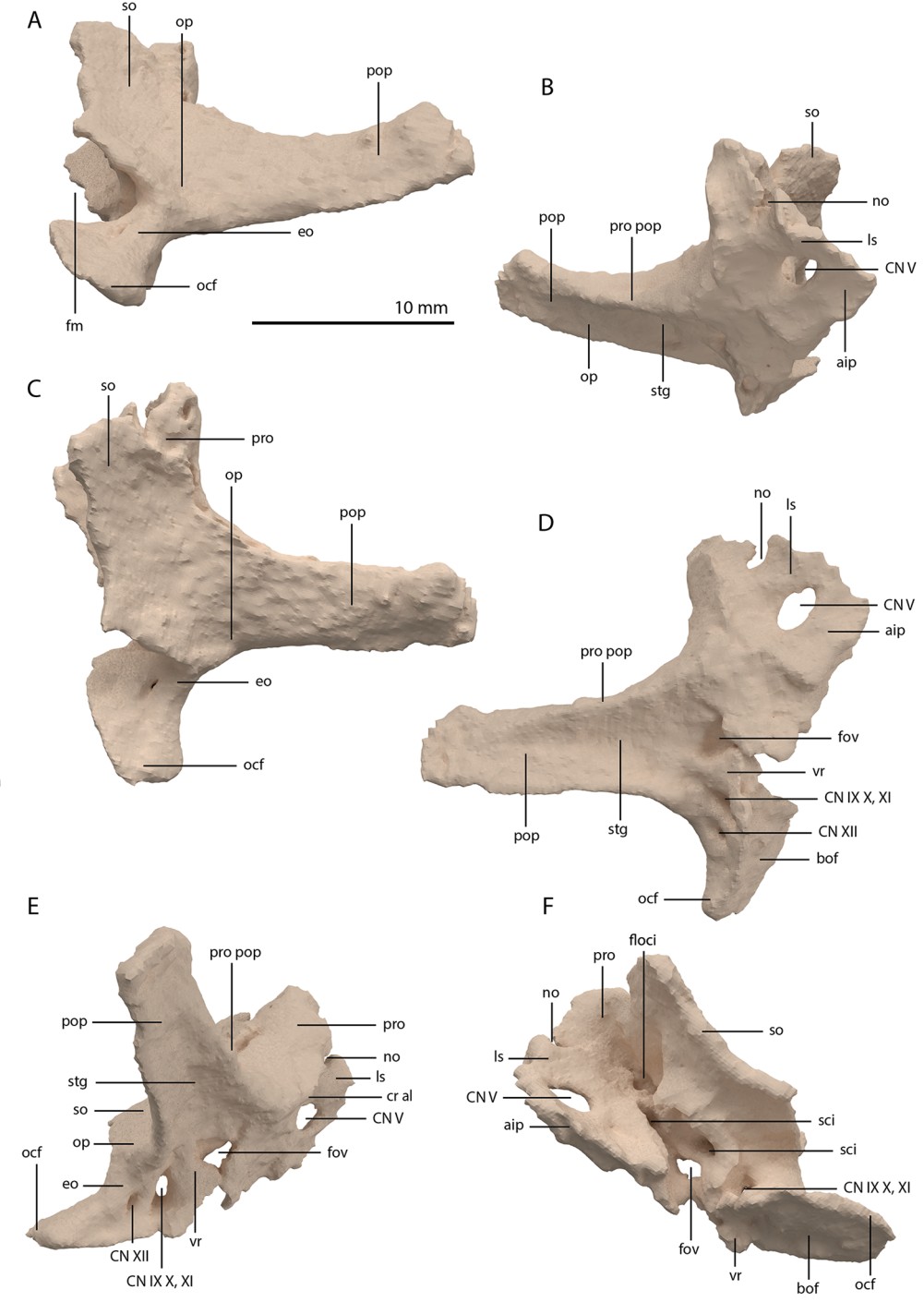

**Figure 25 Digital reconstruction of the right articulated braincase of PIMUZ T 2790, consisting of part of the supraoccipital, the exoccipital, opisthotic, prootic, and laterosphenoid.** (A) Posterior or occipital view. (B) Anterior view. (C) Dorsal view. (D) Ventral view. (E) Lateral view. (F) Medial view. Abbreviations: aip, anterior inferior process; bof, basioccipital facet; CN, cranial nerve; cr al, crista alaris; fm, foramen magnum; floci, flocculus indentation; fov, fenestra ovalis; eo, exoccipital; ls, laterosphenoid; no, notch; ocf, occipital foot; op, opisthotic; pop, paroccipital process; pro, prootic; pro pop, prootic contribution paroccipital process; sci, semi-circular canal indentation; so, supraoccipital; stg, stapedial groove; vr, ventral ramus.

other diapsids (*Evans, 1986*; *Gardner, Holiday & O'Keefe, 2010*; *Gower, 1997*; *Sobral & Müller, 2019*; *Sobral et al., 2016*), and thus demarcates the anteroventral extent of the exoccipital. The exoccipital and opisthotic fully enclose the metotic foramen, which is thus not framed by the basioccipital.

*Opisthotic*

The paroccipital process of the opisthotic projects laterally and slightly posteriorly (Fig. 25). The posterodorsal surface of the process is flattened, whereas the anteroventral surface bears a large ridge running along its dorsal margin which terminates close to the distal end of the process. This ridge forms the dorsal margin of a stapedial groove and corresponds to the ventral ridge or keel described for the paroccipital processes of PIMUZ T 2819 (Figs. 25B, 25D and 25E; *Wild, 1973*). The paroccipital process is virtually straight and maintains its width distally. Likely, the distal end of the paroccipital process connected to a concavity on the squamosal. In *Tanystropheus longobardicus*, the paroccipital process is much shorter and dorsoventrally expands at its distal end (see figure 4A of *Spiekman & Scheyer, 2019*). The opisthotic connects to the exoccipital ventrally, the supraoccipital dorsomedially, and the prootic anteriorly, but none of these sutures are visible in the SRμCT data. Ventrally the ventral ramus of the opisthotic, which is clearly visible in lateral view, frames the anterior margin of the metotic foramen posteriorly and the posterior margin of the fenestra ovalis, through which the stapes was connected to the inner ear, anterodorsally (Fig. 25E). The fenestra ovalis is irregularly shaped, but, in contrast to *Prolacerta broomi* and *Youngina capensis* (*Evans, 1986*; *Gardner, Holiday & O'Keefe, 2010*), its margin is well-ossified. The fenestra ovalis and metotic foramen were previously identified in *Tanystropheus hydroides* based on the elements preserved in PIMUZ T 2819 (*Wild, 1973*). However, a comparison of this specimen to the braincase of PIMUZ T 2790 presented here, reveals that this previous identification was incorrect. The ventralmost foramen visible in PIMUZ T 2819, which was previously identified as the opening for the hypoglossal nerve, actually represents the metotic foramen and the opening dorsal to this, which was considered to be the metotic foramen, is the fenestra ovalis, with the two being separated by the ventral ramus of the opisthotic. The space between the opisthotic-exoccipital complex and the prootic is the result of the slight deformation that likely occurred during compression of the specimen, rather than the fenestra ovalis. The ventral ramus is narrow at its base, widens ventrally, and expands posteriorly where it contacts the anteroventral part of the exoccipital below the metotic foramen. The ventral ramus is incomplete anteroventrally in the right opisthotic as can be deduced from the complete ventral ramus of the left element. Following the morphology in the latter, the ventral ramus is anteroposteriorly expanded at its distal end but does not form a rounded, bulbous process. It thus differs from *Macrocnemus bassanii*, in which the ventral ramus is not expanded, and from *Prolacerta broomi*, in which the ventral ramus has a large bulbous head (*Evans, 1986*; *Miedema et al., 2020*). As for most of the details of the braincase, the morphology of the ventral ramus of the opisthotic is currently unknown for *Tanystropheus longobardicus*. Because the suture between the opisthotic and supraoccipital cannot be observed, the nature of the articulation between these bones is unclear in PIMUZ T 2790.

*Prootic*

The prootic contacts the opisthotic posteroventrally, the supraoccipital posterodorsally, the parabasisphenoid ventrally, the laterosphenoid anteriorly, and likely the parietal anterodorsally. The anteroventral portion of the prootic is tilted somewhat laterodorsally to medioventrally in the right element, which might be the result of slight deformation (Fig. 25). However, all the different structures present on the prootic appear intact. Although a clear suture cannot be established, the posterior outline of the prootic can be delineated because its margin is somewhat raised compared to the opisthotic, whereas the margin of the supraoccipital is somewhat raised above the prootic along their connection. This shows that a narrow tapering process of the prootic contributed to the anterior side of the paroccipital process along the anterior ridge (Figs. 25B, 25D and 25E). This contribution tapers to a point almost halfway along the distal extent of the process. Posteroventrally, the prootic connects with the ventral ramus of the opisthotic to enclose the fenestra ovalis. The crista prootica, which is a ridge running posterodorsally to anteroventrally along the lateral surface of the prootic in archosauromorphs, cannot confidently be identified. In contrast, a sharply anterodorsally curving ridge, the crista alaris (sensu *Sobral & Müller, 2019*), can be clearly discerned (Fig. 25E). The foramen for the exit of the facial nerve (CN VII) is apparently not preserved on either prootic. The large and oval-shaped opening for the trigeminal nerve (CN V) is completely enclosed by the prootic and anteriorly by the laterosphenoid. The portion of the prootic ventral to CN V is the rectangular-shaped anterior inferior process, of which the anterior margin would have connected to the clinoid process of the parabasisphenoid. The connection between the prootic and parabasisphenoid likely continued posteriorly to the clinoid process, closing off the lateral wall of the braincase. This character and the considerable contribution of the prootic to the anterior side of the paroccipital process represent derived characters in *Tanystropheus hydroides* that are also present in early archosauriforms but absent in *Prolacerta broomi*, *Macrocnemus bassanii*, and *Youngina capensis* (*Evans, 1986*; *Evans, 1987*; *Gardner, Holiday & O'Keefe, 2010*; *Gower & Sennikov, 1996*; *Miedema et al., 2020*).

*Laterosphenoid*

The presence of a laterosphenoid is widespread among archosauriforms and has also been identified for *Azendohsaurus madagaskarensis* (*Clark et al., 1993*; *Flynn et al., 2010*; *Gower & Sennikov, 1996*). However it is absent in other non-archosauriform archosauromorphs such as *Macrocnemus bassanii* and *Prolacerta broomi*, as well as in non-saurian diapsids such as *Youngina capensis* (*Evans, 1986*; *Gardner, Holiday & O'Keefe, 2010*; *Miedema et al., 2020*). The presence of a laterosphenoid in *Tanystropheus hydroides* represents the phylogenetically earliest occurrence of this element in the archosauromorph lineage. It is located immediately anterior to the prootic and identified as the bone anteriorly and dorsally enclosing CN V (Fig. 25). As in *Azendohsaurus madagaskarensis*, the laterosphenoid is small and not anteriorly expanded as in archosauriforms. The dorsal part of the anterior margin of the bone bears a distinct notch, which might have transmitted branches of the trigeminal nerve (*Sobral, Hipsley &*

*Müller, 2012*). The suture between the laterosphenoid and the prootic cannot be observed and therefore the exact outline of the bone is unclear.

*Supraoccipital*

The supraoccipital is located dorsal to the exoccipital, dorsomedial to the opisthotic, and posterodorsal to the prootic (Figs. 25A and 25C). The supraoccipital is incomplete and broken into two uneven parts. Therefore, it appears that the break did not occur along a suture and that the supraoccipital was most likely fused. This is supported by the only other known supraoccipital of *Tanystropheus hydroides*, present in PIMUZ T 2787 and visible in anterior view, which is fused (*Wild, 1973*). Little remains of the supraoccipital on the left side of the braincase and it is heavily distorted. The right half is virtually undistorted and complete except for a portion of the medial side. Dorsally the supraoccipital would have connected to the parietals, although no clear facet is present. The supraoccipital is large and plate-like in shape and appears more sloped than in most early archosauromorphs, in which the orientation is more vertical. However, it is unclear whether this represents the original orientation or if it is the result of distortion. The posterodorsal surface of the supraoccipital is largely flat but slightly concave, although it is also unclear whether this is due to deformation. A low sagittal crest along the midline of the supraoccipital was described for PIMUZ T 2787 based on an X-ray (*Wild, 1973*). Presumably, this was also the case for PIMUZ T 2790, but this cannot be corroborated because the medial portion of the supraoccipital is not preserved. A small convexity also runs along the midline of the supraoccipital in the closely related taxa *Tanystropheus longobardicus* (PIMUZ T 2484; figure 4A of *Spiekman & Scheyer, 2019*) and *Macrocnemus bassanii* (*Miedema et al., 2020*). Laterally the supraoccipital thickens distinctly on its anteroventral surface, where the supraoccipital contributes to the otic capsule (Fig. 25F).

*Stapes*

A small rod-like bone preserved between the left frontal dorsally, and the dorsal end of the left quadrate ventrally might represent the left stapes. Its tentative identification is based on its overall shape and size, and because the structure fits neatly within the stapedial groove of the paroccipital process (Fig. 26). It is a thin element, which would correspond to the stapes of other archosauromorphs, but contrasts with the much more robust stapes of non-saurian diapsids such as *Youngina capensis* (*Ezcurra, 2016*; *Gardner, Holiday & O'Keefe, 2010*). The element is apparently incomplete and one of its ends is forked. These two prongs could represent part of the margin of the stapedial foramen. The stapes was previously tentatively identified for *Tanystropheus longobardicus* in PIMUZ T 2485, PIMUZ T 2482, and MSNM BES SC 1018 (*Nosotti, 2007*; *Wild, 1973*). The morphology of these elements roughly corresponds to the tentative stapes described here, but a stapedial foramen cannot be established in any of these specimens.

*Endocast*

The excellent preservation of the right side of the braincase allows for the reconstruction of its endocast, which includes parts of the cerebellum, pons, medulla oblongata, cranial

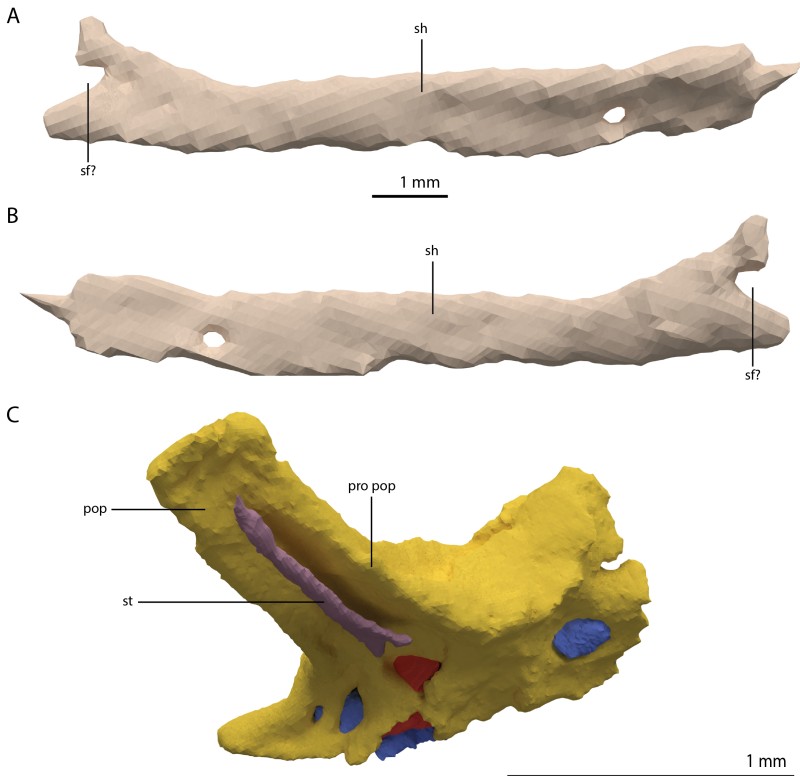

**Figure 26 Digital reconstruction of the element tentatively identified as the left stapes of PIMUZ T 2790.** (A) Anterior or posterior view. (B) Opposite view of (A). (C) The element in reconstructed articulation with the braincase. The fused braincase element is indicated in yellow. The endocast is indicated in blue. The endosseous labyrinth is indicated in red. The braincase, endocast, and endosseous labyrinth were mirrored for this reconstruction. Abbreviations: sf, stapedial foramen; pop, paroccipital process; pro pop, prootic contribution paroccipital process; sh, shaft; st, stapes.

nerves, and the endosseous labyrinth (Fig. 27). The flocculus is remarkably large and extends laterally between the anterior semicircular canal and the vestibule of the endosseous labyrinth. Large flocculi are also present in the early archosauromorphs *Mesosuchus browni* and *Euparkeria capensis*, whereas it is only poorly developed in *Proterosuchus fergusi* (*Brown et al., 2019*; *Sobral & Müller, 2019*; *Sobral et al., 2016*). Posterior to the flocculus, the cerebellum is constricted before expanding again posteriorly towards the medulla oblongata. As described for the braincase elements, three bundles of cranial nerves could be reconstructed. The passage for the facial nerve (CN VII) was not identified. Anterior to the endosseous labyrinth and ventral to the flocculus, the trigeminal nerve (CN V) diverges from the pons. The bundle of the glossopharyngeal nerve, vagus nerve, and accessory nerve (CN IX, CN X, and CN XI) is located directly posterior to the cochlea of the endosseous labyrinth. Posterior to this, a smaller canal would have carried the hypoglossal nerve (CN XII). A small canal in the exoccipital connects the bundle of the CN IX, X, and XI to CN XII. However, it is unclear whether this represents an original feature or is the result of deterioration or deformation of the bone.

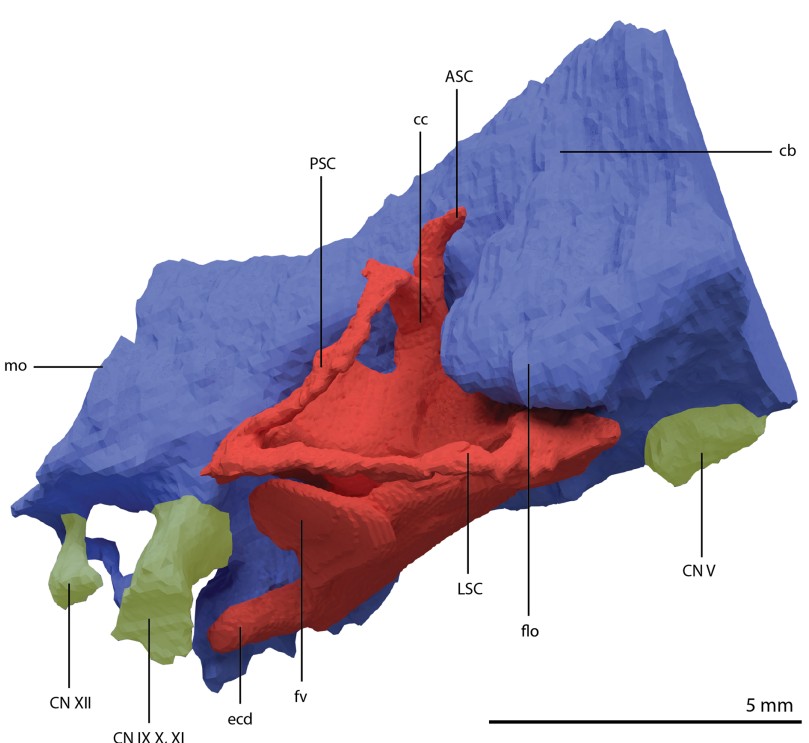

**Figure 27 Digitally reconstructed endocast and endosseous labyrinth of the right fused braincase element of PIMUZ T 2790 in lateral view.** The endocast is indicated in blue, the endosseous labyrinth in red, and the cranial nerves in yellow. Abbreviations: ASC, anterior semicircular canal; cb, cerebellum; cc, common crus; ecd, endosseous cochlear duct; CN, cranial nerve; flo, flocculus; fv, fenestra vestibuli; LSC, lateral semicircular canal; medulla oblongata; PSC, posterior semicircular canal.

*Endosseous labyrinth*

Like all other archosauromorphs, the endosseous labyrinth is enclosed by the opisthotic, prootic, and the supraoccipital (*Sobral & Müller, 2016*). However, the degree to which each element encloses each part of the labyrinth is unclear because no sutures between these elements are discernible. All semicircular canals and the common crus are gracile (Fig. 27). The anterior semicircular canal (ASC) is incomplete. Because it must have curved around the large flocculus, the shape of the ASC can be inferred from the flocculus and the preserved part of the ASC directly dorsal to the common crus (see also figure 1D of *Spiekman et al., 2020*). The ASC was quite tall and elongate and would have been larger than the posterior semicircular canal (PSC). The greater length of the ASC compared to the PSC has been suggested to be a derived crocodylomorph feature, and in non-archosaurian archosauromorphs these canals are approximately similar in size (*Brown et al., 2019*; *Pierce, Williams & Benson, 2017*). However, the ASC is also considerably longer than the PSC in the recently described endosseous labyrinth of the early rhynchosaur *Mesosuchus browni* (*Sobral & Müller, 2019*). The PSC is less arched than the ASC and exhibits sinusoidal curvature. The lateral semicircular canal (LSC) is similar in length to the PSC and does not exhibit a continuous curvature, instead bending inwards at its approximate mid-point. In the 're-assembled' skull of PIMUZ T 2790, the

LSC is oriented at approximately the same angle as the horizontal plane of the skull as estimated from the tooth row of the maxilla (angle = ~2°). However, this measurement might deviate slightly, as it is dependent on the orientation of the braincase in relation to the rest of the skull, which can only be approximated in the 're-assembled' skull. The endosseous cochlear duct is well defined and posteroventrally orientated. It is relatively elongate, similar to that of *Youngina capensis*, and considerably longer than in *Mesosuchus browni* and *Proterosuchus fergusi* (*Brown et al., 2019*; *Gardner, Holiday & O'Keefe, 2010*; *Sobral & Müller, 2019*). The extent of the cochlea of *Euparkeria capensis* is unclear (*Sobral et al., 2016*).

### Cranial openings

#### External naris

The lack of any distinct prenarial process of the premaxilla indicates that the external nares were likely confluent. However, the lack of anything but a very fragmentary nasal makes any additional inferences with regards to the external naris based on PIMUZ T 2790 impossible. The nasals and external nares have been reconstructed for *Tanystropheus hydroides* based on inferences from PIMUZ T 2787 and PIMUZ T 2819 (Fig. 6B; see also the discussion on the nasal below and in *Spiekman et al. (2020)* and supplemental figures 1A and 2B therein). This reveals that the external nares were framed only by the premaxillae and nasals and confirms that they were confluent. Confluent external nares are also known for *Tanystropheus longobardicus* (*Spiekman et al., 2020*), the allokotosaurs *Azendohsaurus madagskarensis* (*Flynn et al., 2010*), *Shringasaurus indicus* (*Sengupta, Ezcurra & Bandyopadhyay, 2017*) and *Pamelaria dolichotrachela* (*Sen, 2003*), *Teyujagua paradoxa* (*Pinheiro, Simão-Oliveira & Butler, 2019*), and rhynchosaurs (*Dilkes, 1998*). However, in these taxa the external nares are anterodorsally facing, whereas in the flat-snouted *Tanystropheus hydroides* the external nares face entirely dorsally.

#### Orbit

The outline of the posterodorsal margin of the orbit is unclear due to the lack of information on the shape of the postfrontal and its articulation with the surrounding bones. Nevertheless, the orbit appears to be subcircular and slightly longer anteroposteriorly than tall dorsoventrally (Figs. 4 and 6A). The dorsal margin of the orbit is largely formed by the curved lateral margin of the wide frontal. The anterior margin of the orbit is framed by the prefrontal dorsally and the lacrimal ventrally. The curved anterior process of the jugal forms the ventral margin of the orbit. The ventral process of the postorbital largely covers the jugal anteriorly and thus forms most of the posterior margin of the orbit. The overall arrangement of the bones framing the orbit results in a largely laterally and slightly anteriorly directed orbit.

#### Temporal fenestra

The three-dimensional preservation of PIMUZ T 2790 allows for a clear rendition of the temporal fenestrae of *Tanystropheus hydroides*. The supratemporal fenestra is considerably wider than it is long (Figs. 5A and 6B). At least the medial half of its anterodorsal margin was formed by the anterolateral process of the parietal, with the

margin of the lateral half being either formed by the parietal or the medial process of the postorbital. Although the shape of the postfrontal is unknown for *Tanystropheus hydroides*, it seems unlikely that the postfrontal contributed to the margin of the supratemporal fenestra based on the shape and articulation of the postorbital and parietal. The medial margin of the supratemporal fenestra was very short and formed by the lateral portion of the main body of the parietal. The supratemporal fossa of the parietal is strongly sloped ventrally and very tall, presenting a large surface area for the attachment of jaw adductor musculature (Figs. 10A, 10D and 10E). Posteriorly, the margin of the supratemporal fenestra is formed by an equal contribution of the dorsoventrally tall posterolateral process of the parietal medially and the similarly shaped medial process of the squamosal laterally (Fig. 5A). The lateral margin is formed by the posterior process of the postorbital anteriorly, and the anterior process of the squamosal posteriorly.

The infratemporal fenestra is relatively small in *Tanystropheus hydroides* among archosauromorphs because of the dorsoventrally tall postorbital bar and generally short length of the temporal region of the skull (Figs. 4 and 6A). The dorsal margin of the infratemporal fenestra has an oblique anteroventral to posterodorsal orientation, being formed by the squamosal and jugal. Because the ascending process of the jugal reaches the squamosal, the postorbital is excluded from the margin of the infratemporal fenestra. This configuration is also present in *Jesairosaurus lehmani* (ZAR 06), *Dinocephalosaurus orientalis* (IVPP-V13767), and *Pectodens zhenyuensis* (IVPP-V18578) among early archosauromorphs. The squamosal does not possess a separate ventral process and the posterior margin of the infratemporal fenestra was formed by the anterolateral margin of the quadrate and the quadratojugal. The horizontally orientated posterior process of the jugal formed the ventral margin of the fenestra but did not connect to any bone posteriorly and therefore the infratemporal fenestra was open ventrally.

*Choana*
The palatal region is only partially preserved in PIMUZ T 2790 (Fig. 5B), but the morphology of the palatal elements can additionally be inferred from the well-preserved elements of PIMUZ T 2787 (see figure 4D and 4G of *Spiekman & Scheyer, 2019*). The combined information from these two specimens allows for a complete reconstruction of the palate, which reveals the size and shape of the ventral openings of the skull (Fig. 6C). The choana is long but narrow and framed by a row of large teeth on both the lateral and medial side. It is enclosed laterally by the maxilla, anteriorly by the premaxilla, medially by the vomer, and posteriorly by the palatine.

*Suborbital fenestra*
Because our identification of the ectopterygoid is tentative, we are not able to confidently interpret the shape of the suborbital fenestra. It was likely framed by the palatine, ectopterygoid, pterygoid, and maxilla (Figs. 5B and 6C). A clear facet on the medial side of the ventral surface of the palatine indicates how this bone articulated with the palatal ramus of the pterygoid and suggests that a small vacuity might have also been present anteriorly between both elements and the vomer.

*Posttemporal fenestra*

The preservation of the posterolateral processes of the parietals and the braincase elements including the paroccipital process of the opisthotic allows for the detailed reconstruction of the occipital region of *Tanystropheus hydroides* (Fig. 5C). This reveals the presence of a slit-like posttemporal fenestra with a curved ventral margin. It is framed by the posterolateral process of the parietal dorsally, the supraoccipital medially, and the paroccipital process of the opisthotic ventrally.

### Mandible

Both mandibular rami have been preserved and are mostly complete and in articulation with each other. In the right ramus, the dentary is detached from the more posterior elements and overall, the preservation of the left mandibular ramus is better than that of the right. Therefore, its reconstruction is largely based on the left ramus (Fig. 28), supplemented by information from the right ramus and the mandibles of other *Tanystropheus hydroides* specimens.

*Dentary*

Both dentaries have been slightly displaced and are located beneath their corresponding premaxilla and maxilla. The dentary makes up roughly two-thirds of the surface of the mandible in lateral view. It is a long, labiolingually thin bone. The mandibular symphysis is exclusively formed by the dentaries and both mandibular rami were strongly connected by an interdigitating suture, as is indicated by the complex pattern of ridges and grooves on the medial surface on the anterior end of the dentary (Figs. 28B and 28C) and. Here, the dentary is also conspicuously expanded ventrally, forming a distinct keel that decreases in height posteriorly and terminates in between the level of the third and fourth alveolus. The presence of this keel is autapomorphic for *Tanystropheus hydroides* among non-archosauriform archosauromorphs. No foramina could be identified on either of the dentaries of PIMUZ T 2790. However, up to nine foramina were previously identified in specimens that are now referred to *Tanystropheus hydroides* in PIMUZ T 2819 and PIMUZ T 2793, and at least one of these was considered to open into the Meckelian groove medially (*Wild, 1973*). The broken surfaces of the dentaries in these specimens hamper clear observation of these foramina, but it is possible that at least a few small foramina were indeed present. Foramina on the anterior part of the dentary are unambiguously present in *Tanystropheus longobardicus* (*Nosotti, 2007*). In total, 18 alveoli can be identified in the left dentary and 17 in the less complete right one. The three anteriormost are the largest. In lateral view, the alveolar margin appears to be straight to slightly convex posteriorly, with a slight concavity in between the three large anterior alveoli and the remaining alveoli. On the medial surface, the Meckelian canal is dorsoventrally high posteriorly, taking up more than half of the medial surface of the dentary (Fig. 28C). Anteriorly the canal narrows, especially around the level of the eighth alveolus. Further anteriorly, approximately at the level of the third alveolus, the canal expands again and finally terminates near the symphysis. The posterior margin of both dentaries is incomplete, and therefore the sutures of this bone with the surangular and

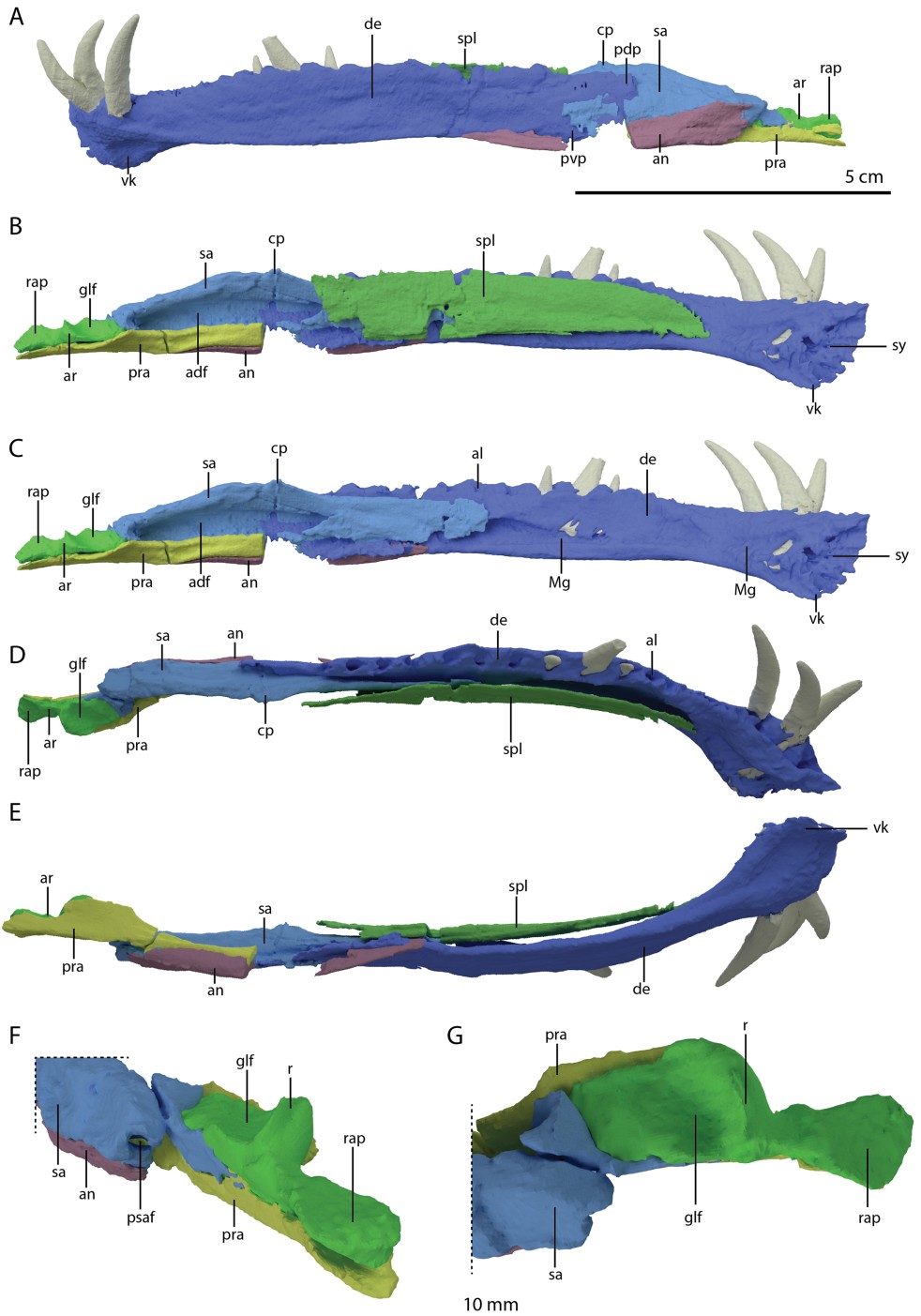

**Figure 28 Digital reconstruction of the left mandibular ramus of PIMUZ T 2790.** (A) Lateral view. (B) Medial view including splenial. (C) Medial view excluding splenial. (D) Dorsal view. (E) Ventral view. (F) Close-up of the posterior part of the mandibular ramus, including the glenoid fossa and retroarticular process, in oblique posterolateral view. (G) Close-up of the posterior part of the mandibular ramus in dorsal view. Abbreviations: adf, adductor fossa; al, alveoli; an, angular; ar, articular; cp, coronoid process; de, dentary; glf, glenoid fossa; Mg, Meckelian groove; pdp, posterodorsal process; pra, prearticular; psaf, posterior surangular foramen; pvp, posteroventral process; r, ridge; rap, retroarticular process; sa, surangular; spl, splenial; sy, symphysis; vk, ventral keel.

angular cannot be established. However, the dentary of the largest known specimen of *Tanystropheus hydroides*, PIMUZ T 2793, is intact, and bears tapering posterodorsal and posterocentral processes (sensu *Ezcurra, 2016*; see figures 18 and 19 of *Wild, 1973*). PIMUZ T 2790 clearly shows that the dentary overlapped the surangular and angular laterally, as had also previously been observed by *Wild (1973)*. In contrast, in *Tanystropheus longobardicus* the surangular is considered to overlap the dentary laterally, based on specimens MSNM BES SC 265, MSNM BES SC 1018, and PIMUZ T 3901 (*Nosotti, 2007*).

*Splenial*
Both splenials are preserved medial to their corresponding dentary and are incomplete posteriorly. The splenial has an elongate plate-like shape and does not curve around the dentary ventrally as in *Macrocnemus bassanii* or *Tanystropheus longobardicus* (*Miedema et al., 2020*; *Nosotti, 2007*), and was thus not visible in lateral view (Figs. 28A and 28E). The anteromedial curvature of the splenial follows that of the dentary. Based on their corresponding curvature, the splenial likely extended anteriorly to approximately the level of the third alveolus of the dentary. The bone is dorsoventrally tall, covering the dentary almost completely in medial view (Fig. 28B). In contrast, the splenial of *Tanystropheus longobardicus* is much shorter dorsoventrally and does not reach as far anteriorly (MSNM BES SC 1018; *Nosotti, 2007*). The posterior extent of the splenial is unclear, although based on the right element, it possibly reached the adductor fossa.

*Angular*
The right angular is located posteromedial to the posterior extent of the left dentary. It is slightly disarticulated from the surangular. The right angular, although less complete than the left, appears to still be in its natural articulation with the surangular and dentary. It is broken in two pieces due to a large break in the ventral margin of the left mandibular ramus directly behind the posterior extent of the dentary. In lateral view the angular clearly extended further anteriorly than the surangular on the left mandibular ramus (Fig. 28A). Although the angular wraps around the surangular ventrally, the angular is only slightly exposed medially and is restricted to the ventral border of the mandible on this side (Figs. 28B, 28C, and 28E). Although its preservation is somewhat poor, the right angular shows most of the lateral exposure of the element. Its dorsal margin was oriented anteroventrally to posterodorsally and is largely straight to slightly concave. Based on the preservation in the left mandibular ramus, its dorsal border was likely connected to the lateral margin of the surangular shelf. The exact posterior extent is unclear, but it appears to have terminated anterior to the glenoid fossa at the posterior end of the surangular shelf. However, it is possible that the angular reached further posteriorly on its ventral side. In this regard, the contribution of the angular to the lateral surface of the mandible is much larger than that in *Macrocnemus bassanii*, and its dorsal margin is less curved than in other early archosauromorphs such as *Prolacerta broomi* and *Azendohsaurus madagaskarensis* (*Flynn et al., 2010*; *Miedema et al., 2020*; *Spiekman, 2018*).

*Surangular*

The left surangular is located dorsal to the left angular and directly posterior to the left dentary (Fig. 28A). The right surangular is located dorsal to the right angular and posteromedially to the right dentary. It was not possible to separate the surangular from the articular and prearticular in the right mandibular ramus (Fig. 4A), whereas in the left ramus, where these elements are largely in their natural articulation, they can clearly be distinguished (Fig. 4B). The left surangular is complete apart from its ventral margin, which is poorly preserved. The right surangular appears complete except for its anterior portion, which is broken. The anterior end of the surangular, together with the dentary, clearly indicates that there was no external mandibular fenestra as seen in *Teyujagua paradoxa* and archosauriforms (Fig. 28A; *Pinheiro, Simão-Oliveira & Butler, 2019*). The surangular was covered anteriorly by the dentary in lateral view and the splenial in medial view (Figs. 28A and 28B). A clear articulation surface on the ventral half of the lateral surface of the surangular indicates the outline of where the angular covered the surangular laterally. The dorsal margin of the surangular is straight anteriorly, but further posteriorly bears an anteroposteriorly long but low convexity. On this convexity, a very small process protrudes dorsally, which represents a small coronoid process. A separate coronoid bone could not be distinguished. However, it is possible that it tightly articulates with the surangular and that the suture between these two elements could not be discerned in the SRμCT data on either side. Therefore, the absence of a coronoid for *Tanystropheus hydroides* cannot be confirmed. It is similarly unclear whether the coronoid represented a separate element in *Tanystropheus longobardicus* (*Nosotti, 2007*). On both surangulars, a large posteriorly opening foramen is present directly anterior to the glenoid fossa (Fig. 28F). This foramen represents the posterior surangular foramen sensu *Ezcurra (2016)* and is present in various archosauriforms and in *Prolacerta broomi*, *Azendohsaurus madagaskarensis*, *Eohyosaurus wolvaardti*, and *Teyujagua paradoxa* among non-archosauriform archosaurmorphs (*Butler et al., 2015*; *Flynn et al., 2010*; *Pinheiro, Simão-Oliveira & Butler, 2019*; *Spiekman, 2018*). No other foramina could be identified on the lateral surface of either surangular. In lateral view an inconspicuous surangular shelf can be discerned, which is reminiscent of the shelf seen in non-rhynchosaurid rhynchosaurs (e.g. *Mesosuchus browni*, *Dilkes, 1998*), but distinctly less pronounced than those seen in *Teyujagua paradoxa* and several early Archosauriformes (*Ezcurra, 2016*; *Pinheiro, Simão-Oliveira & Butler, 2019*). It is horizontal posteriorly, but directed anteroventrally further anteriorly, where it disappears posterior to the level of the coronoid process. Below the dorsal convexity of the surangular, a large and deep concavity is present on its medial surface (Figs. 28B and 28C). This represents the lateral wall of the adductor fossa, which is quite large in *Tanystropheus hydroides*, and would have formed the attachment site on the mandible for most of the adductor musculature. A posteriorly directed process extends from the posteromedial side of the surangular, which would have reached the anterior extent of the medial surface of the prearticular to form the medial wall of the adductor chamber.
*Prearticular*

Both prearticulars are preserved, although it was not possible to isolate the right element from its surrounding bones in the SRμCT data. The right prearticular is located medial to the surangular. The left prearticular is still in its natural articulation, covering the articular ventrally along the entire length of the latter (Figs. 28E and 28F). The anterior portion of the prearticular is covered by the angular ventrally and laterally.

The anteriormost part of the left prearticular is missing because of the same break that affects the left angular (Figs. 28B, 28C and 28E). Based on the prearticular from the right mandibular ramus, in which the anterior extent of the bone can be made out, the prearticular extends anteriorly to approximately the anterior margin of the adductor fossa. The prearticular is a thin, elongate bone that mediolaterally widens distinctly at the level of the glenoid fossa on the articular (Fig. 28E). It is largely flat and somewhat concave posteriorly with a deep groove that receives the articular dorsally. Further anteriorly, the bone becomes dorsoventrally tall and lateromedially thin, and fits tightly into the curved angular ventrally and the surangular medially, whilst forming most of the ventral margin of the adductor fossa. The prearticular extends roughly equally far posterior as the articular and thus contributes distinctly to the retroarticular process as in *Macrocnemus bassanii* (*Miedema et al., 2020*). Posterior to the posteriormost extent of the angular, the prearticular is clearly visible in lateral view (Fig. 28A).

*Articular*

As with the other posterior bones of the mandible, the articular is present in the right mandibular ramus but it cannot be separated from the surrounding elements. The left articular is complete and in articulation, fitting tightly on the dorsal surface of the prearticular (Fig. 28). The articular has a very short, blunt projection anterior to the glenoid fossa, which fits anterodorsally and laterally against the posterior part of the surangular and anteroventrally onto the prearticular. The ventral surface bears an anteroposteriorly directed keel that articulates tightly with the concave groove on the dorsal surface of the prearticular. The glenoid fossa, which forms the articulation with the ventromedial head of the quadrate, is framed by a raised margin (Figs. 28F and 28G). This margin is raised into a thin vertical ridge on its posterior side. The lateral part of the margin is also raised to form a ridge, particularly on its posterior part. The posterolateral margin in between the posterior and lateral margins is distinctly lower than its neighbouring ridges. The medial margin of the glenoid fossa is strongly displaced ventrally and only somewhat raised posteriorly. The rest of the medial margin, as well as the anterior margin, is not raised. The dorsal surface of the glenoid fossa is formed by a concavity that is orientated posterolaterally to anteromedially (Fig. 28F). Additionally, the fossa slopes laterodorsally to medioventrally. Posterior to the glenoid fossa a retroarticular process is developed on the lateral side of the articular. It forms approximately one-third of the anteroposterior length of the articular and is thus comparatively much larger than in *Macrocnemus bassanii* and *Dinocephalosaurus orientalis* (IVPP-V13767; *Miedema et al., 2020*; *Rieppel, Li & Fraser, 2008*; *Spiekman et al., 2020*), and similar to that of early rhynchosaurs such as *Howesia browni*, *Eohyosaurus*

*wolvaardti*, and *Mesosuchus browni* (*Butler et al., 2015*; *Dilkes, 1995*; *Dilkes, 1998*). It is mediolaterally constricted at its base and widens posteriorly (Fig. 28G). Its posterolateral margin is only very slightly upturned. The posterior end of the retroarticular process is slightly rounded, but less distinctly so than in *Macrocnemus bassanii* (*Miedema et al., 2020*).

### Dentition

The dentition on both upper and lower jaws is heterodont and characterized by very large, recurved fangs on the premaxilla and anterior part of the dentary and sharp, conical teeth of varying sizes on the maxilla and the remainder of the dentary (Figs. 4 and 29). Although not visible in the SRμCT data, external observation of the specimen reveals that there are distinct striations running along all the marginal teeth (Figs. 2A and 2B). The teeth are not serrated. In these characteristics, the teeth of *Tanystropheus hydroides*, as well as those of *Dinocephalosaurus orientalis*, are very similar to certain sauropterygians and indicate a piscivorous diet acquired through a similar feeding mechanism (*Rieppel, 2002*). All marginal teeth are distinctly labiolingually compressed. They exhibit a subthecodont tooth implantation (sensu *Fraser & Shelton, 1988*), marked by a strong and tall labial ridge and a strongly reduced lingual margin. The latter is particularly visible in the premaxilla and the maxilla (Figs. 7B and 7D). As mentioned above, the premaxilla bears six teeth, the maxilla 15, and the dentary likely 18, of which the anterior three teeth are enlarged fangs (Fig. 29). In some cases a replacement tooth is present lingual to the erupted tooth, indicating continuous tooth replacement. Because the size of each tooth is partially dependent of their growth stage, and because many teeth are missing in PIMUZ T 2790, it is hard to establish the relative size of the teeth throughout the marginal dentition. Nevertheless, the three fangs on each side of the mandible appear to be somewhat larger than the fangs on the premaxilla, and they interlock with each other to form a 'fish-trap' structure (sensu *Rieppel, 2002*). Based on the size of the alveoli, the first five teeth of the premaxilla were large fangs, with the sixth tooth being somewhat reduced in size (Fig. 29). Posterior to this, the anteriormost maxillary tooth is very small. From here, the size of the maxillary teeth increases gradually until the sixth maxillary tooth. Maxillary teeth six through eight are the largest ones. From the eighth tooth, the size of the teeth gradually decreases posteriorly, with the final, the 15th, maxillary tooth only protruding slightly ventral to the maxilla in lateral view (Figs. 7A and 29). This last tooth is located at the tapering end of the posterior process of the maxilla. In the dentary, the size pattern is quite different. It starts with the three large anterior fangs, followed by three strongly reduced alveoli that are barely visible (Figs. 28D and 29). Posterior to this, at the level of the last premaxillary tooth, the first of the larger conical teeth is located. As with the maxillary teeth, this tooth is still comparatively small and the size of the subsequent three teeth increases posteriorly. The large 10th dentary tooth pierces through the surface of the maxilla. Dentary teeth also pierce upper jaw bones in certain crocodylians and temnospondyls (*Cidade et al., 2017*; *Schoch, 1999*). After the 10th tooth, the alveoli gradually reduce in size posteriorly (Fig. 29). The posteriormost alveolus on the dentary is located at the same level as the third-to-last (13th) alveolus of the

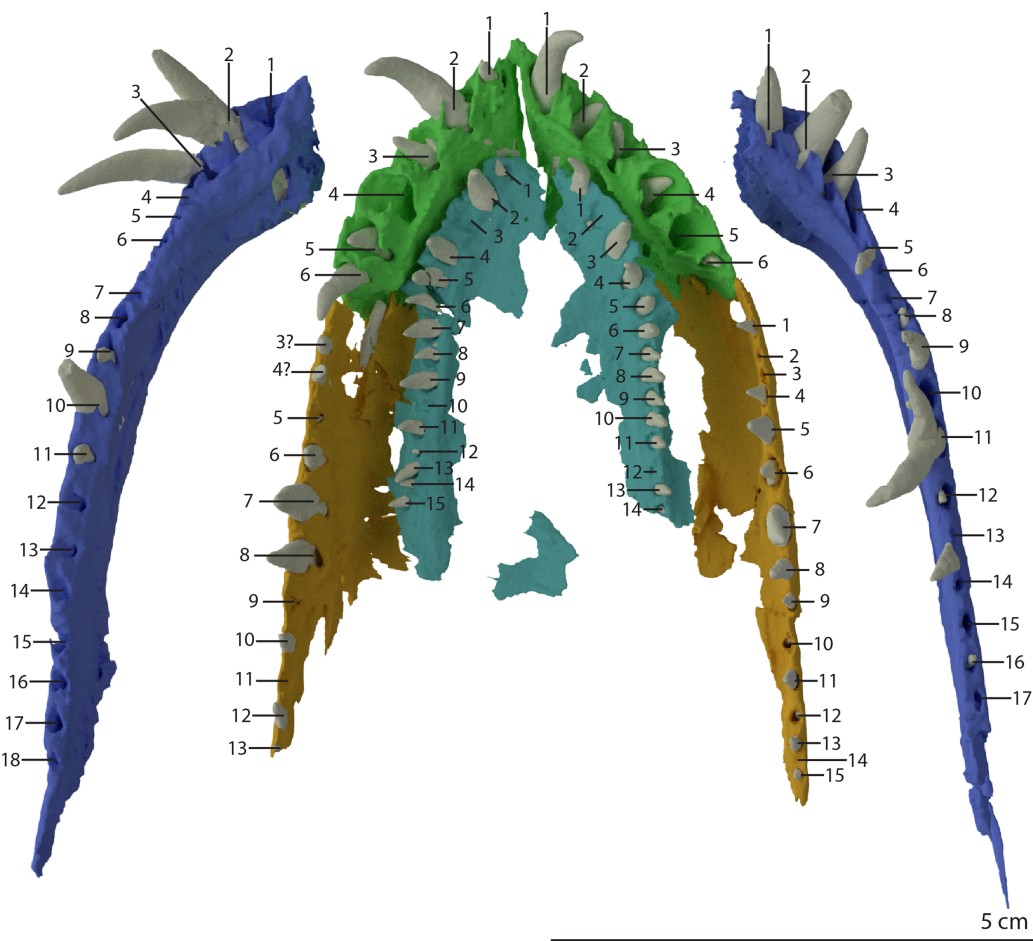

5 cm

**Figure 29 Digital reconstruction of the tooth bearing elements of PIMUZ T 2790.** The dentaries (in blue) are shown in dorsal view. The premaxillae (green), maxillae (orange), and vomers (turquoise) are shown in ventral view. The numbers indicate the position of each alveolus for each element counted from anterior to posterior.

maxilla. The conical dentition of the skull posterior to the fangs overlapped their counterparts on the mandible, rather than interlocking with them (Figs. 2B and 4A).

The vomerine teeth are smaller than most marginal teeth but still comparatively large for palatal teeth among archosauromorphs (Fig. 5B). They are recurved, being orientated ventrolabially at their base and ventrally at their distal end (Fig. 17B). They are homodont in shape and gradually decrease in size posteriorly. They are not serrated, but since the teeth can only be observed in the SRμCT data it cannot be assessed whether they were striated. The tooth implantation is subthecodont, with the anterior alveoli having a distinctly higher labial margin compared to the lingual one. Further posteriorly this distinction decreases until, at approximately the 7th alveolus counted from anterior on the right vomer, both margins are roughly equally well-developed. As mentioned above, the more complete right vomer bears 15 teeth positions (Fig. 29). This is likely the total number of teeth on the vomer, although it cannot be fully excluded that one or two additional tooth positions have been lost. As for the marginal dentition, vomerine

tooth replacement was continuous and small replacement teeth are in some cases preserved lingual to an erupted tooth in the alveolus.

### Postcranial skeleton

In addition to the skull, PIMUZ T 2790 preserves cervical vertebrae one through eight (Fig. 1A). Of these, the atlas-axis complex and anterior part of the third cervical were scanned together with the skull. This allows for the first detailed description of the atlas-axis complex of *Tanystropheus hydroides* (Fig. 30), as well as a description of the internal anatomy of the axis and the first postaxial cervical vertebra (Figs. 31 and 32). The largest vertebra is 190 mm in length (all lengths are provided in table 3 of *Wild, 1973*, PIMUZ T 2790 is specimen *p* therein).

*Atlas-axis complex*
The atlas-axis complex is preserved but disarticulated in PIMUZ T 2790, and consists of the atlas pleurocentrum, atlas intercentrum, two atlantal neural arches, axis intercentrum, and axis. The atlas pleurocentrum and intercentrum are preserved in between the axis ventrally and right postorbital dorsally. The right atlantal neural arch is located above the left quadrate and below the left frontal and squamosal, whereas the left atlantal neural arch is preserved underneath the right squamosal and the anterior end of the axis. Anterior to the atlas pleurocentrum and intercentrum, the axis intercentrum is preserved between the right posterolateral process of the fused parietals dorsally and the anterior part of the neural spine of the axis ventrally.

No proatlases could be identified. Proatlases are small elements that are present dorsal to the atlantal neural arches in various diapsids and, among non-archosauriform archosauromorphs, have been identified in *Macrocnemus bassanii*, *Trilophosaurus buettneri*, and *Azendohsaurus madagaskarensis* (*Miedema et al., 2020*; *Nesbitt et al., 2015*; *Spielmann et al., 2008*). Because these elements are small and easily disarticulated, it is possible that they were present in *Tanystropheus hydroides* but not preserved or identified in PIMUZ T 2790.

The atlantal neural arch is a tripartite element with a complex structure (Fig. 30). Its posterior process is elongate and formed the articulation with the small prezygapophysis of the axis. Anteriorly the neural arch has a dorsomedially and a ventromedially projected process. The dorsomedial process is a flattened wing-like structure, whereas the ventromedial process expands somewhat distally and ends in a flattened foot-like structure, which would have articulated with the atlas pleurocentrum (Fig. 30D). Together, the two anterior processes of the atlantal neural arch would have framed the neural canal anterior to the axis.

The atlas intercentrum is the smallest element of the complex and is crescent-shaped. It would have been located anteroventral to the atlas pleurocentrum and anterodorsal to the axis intercentrum (Fig. 30).

In the allokotosaurs *Trilophosaurus buettneri* and *Azendohsaurus madagaskarensis* the atlas pleurocentrum is partially or fully fused to the axis intercentrum, forming an odontoid complex (*Nesbitt et al., 2015*; *Spielmann et al., 2008*). However, as in *Macrocnemus bassanii* and *Mesosuchus browni*, these elements are present as separate

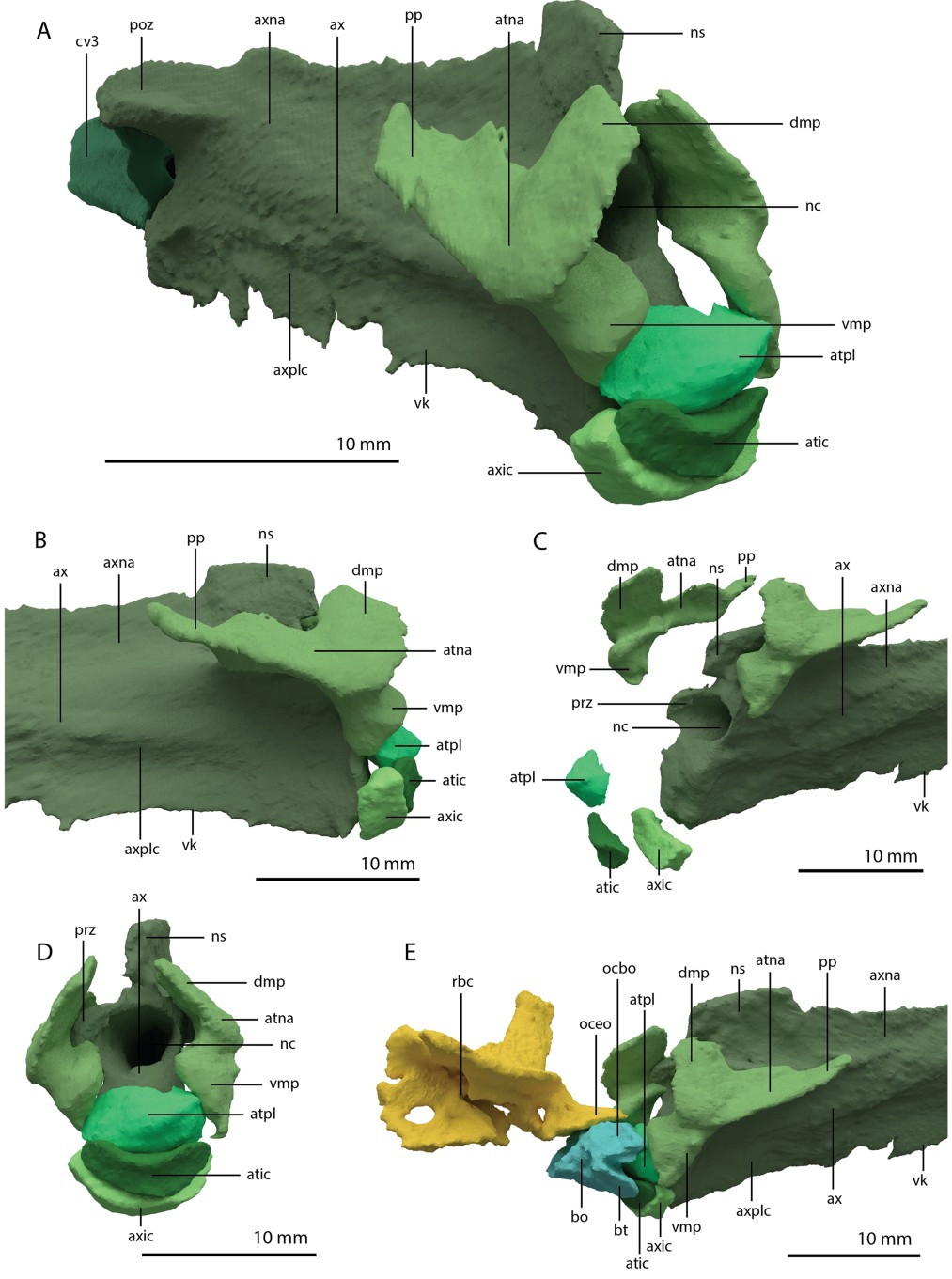

**Figure 30 'Re-assembled' digital reconstruction of the atlas-axis complex of PIMUZ T 2790.**
(A) Oblique right anterolateral view. (B) Right lateral view. (C) Disarticulated oblique left ante-
rolateral view. (D) Anterior view. (E) Oblique left anterolateral view of the atlas-axis complex in
articulation with the basioccipital and the right fused braincase element. Abbreviations: atic, atlas
intercentrum; atna, atlas neural arch; atpl, atlas pleurocentrum; ax, axis; axic, axis intercentrum; axna,
axis neural arch; axplc, axis pleurocentrum; bo, basioccipital; bt, basal tuber; cv, cervical vertebra; dmp,
dorsomedial process; nc, neural canal; ns, neural spine; ocbo, basioccipital contribution occiput; oceo,
exoccipital contribution occiput; poz, postzygapophysis; pp, posterior process; prz, prezygapophysis; rbc,
right braincase; vk, ventral keel; vmp, ventromedial process.

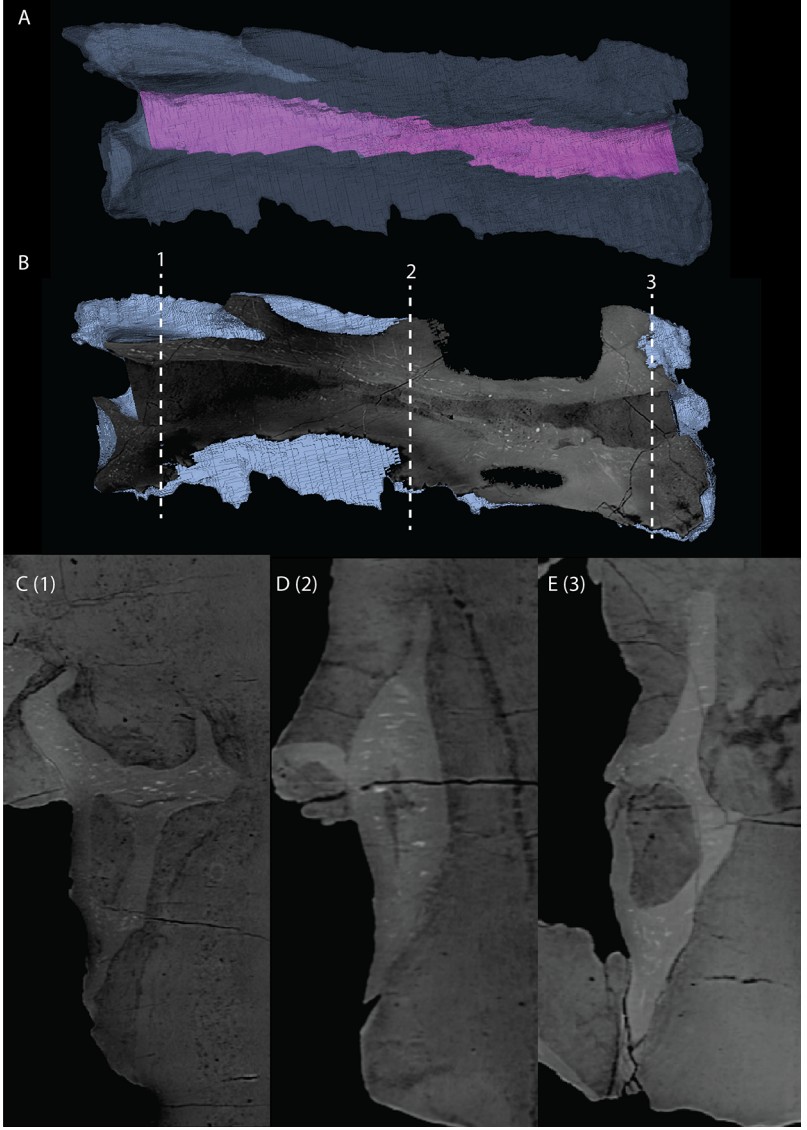

**Figure 31 Inner anatomy of the axis of PIMUZ T 2790.** (A) Transparent digital rendering of the axis, with the neural canal indicated in purple, in right lateral view. (B) Digital sagittal cross-section of the axis in right lateral view. The numbers above the stippled lines correspond to the numbers of the SRμCT slices in axial cross-section. (C) Digital axial cross-section of the posterior part of the axis at the level indicated by number 1 in the sagittal cross-section. (D) Digital axial cross-section of the middle part of the axis at the level indicated by number 2 in the sagittal cross-section. (E) Digital axial cross-section of the anterior part of the axis at the level indicated by number 3 in the sagittal cross-section.

elements in *Tanystropheus hydroides* (Fig. 30; *Dilkes, 1998*; *Miedema et al., 2020*). The atlas pleurocentrum has an oval cross-section and a flat posterior surface that articulated with the anterior surface of the centrum of the axis pleurocentrum. The anterior margin is distinctly convex and forms the odontoid process that would have articulated with the basioccipital directly below the occipital condyle, thus aligning the neural canal of the axis with the foramen magnum of the skull (Fig. 30E). The dorsal surface of the atlas

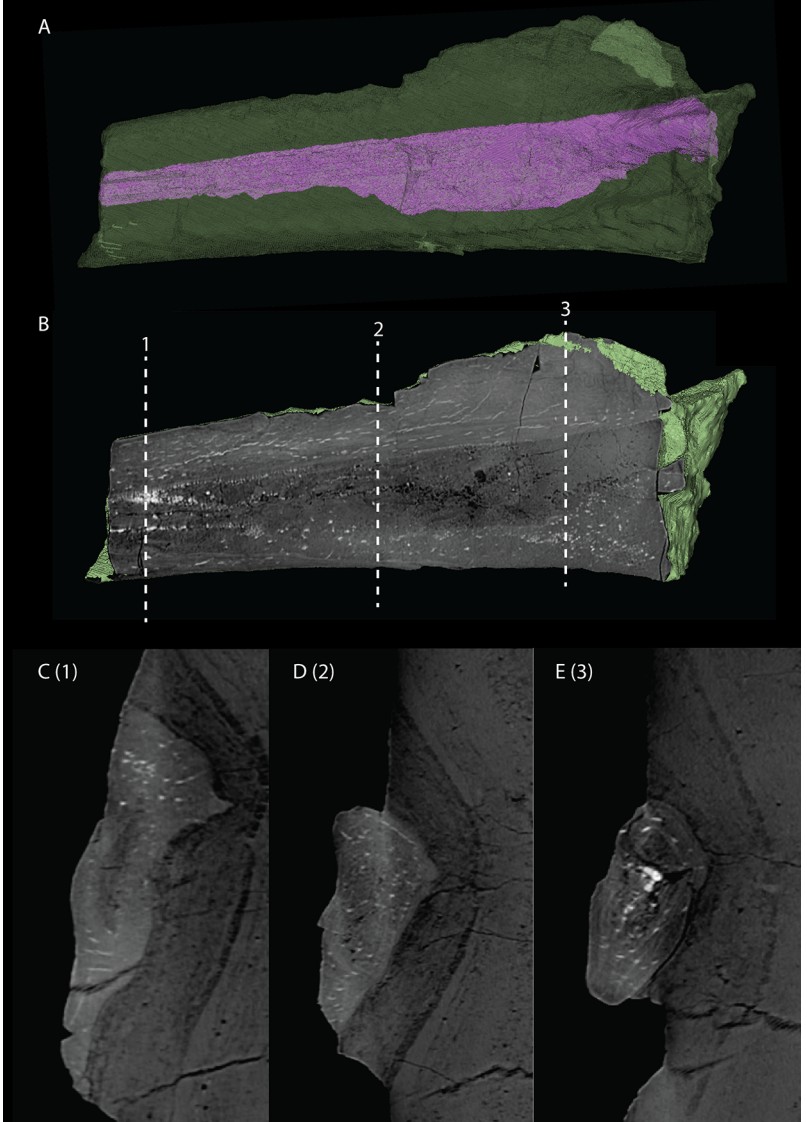

**Figure 32 Inner anatomy of the anterior part of the third cervical vertebra of PIMUZ T 2790.**
(A) Transparent digital rendering of the third cervical, with the neural canal indicated in purple, in right lateral view. (B) Digital sagittal cross-section of the third cervical in right lateral view. The numbers above the stippled lines correspond to the numbers of the SRµCT slices in axial cross-section. (C) Digital axial cross-section of the middle part of the third cervical at the level indicated by number 1 in the sagittal cross-section. (D) Digital axial cross-section of the middle to anterior part of the third cervical at the level indicated by number 2 in the sagittal cross-section. (E) Digital axial cross-section of the anterior part of the third cervical at the level indicated by number 3 in the sagittal cross-section.

pleurocentrum forms the floor of the neural canal and is flattened, whereas in *Macrocnemus bassanii* and *Azendohsaurus madagaskarensis* it is distinctly excavated (*Miedema et al., 2020*; *Nesbitt et al., 2015*).

The axis intercentrum has a similar shape to the atlas intercentrum but is considerably larger. It would have articulated with the axis posteriorly, the atlas intercentrum anteriorly, and the atlas pleurocentrum dorsally (Fig. 30). The anterior margin of the axis

intercentrum is flattened as in *Azendohsaurus madagaskarensis*, but in contrast to *Macrocnemus bassanii*, in which it is rounded (*Miedema et al., 2020*; *Nesbitt et al., 2015*).

At its anterior end the neural spine of the axis is slightly raised, but it is otherwise similar to that of the postaxial cervical vertebrae and very poorly developed (Fig. 30). Short anterolaterally projecting processes on the anterior end of the neural arch of the axis represent the poorly developed prezygapophyses, which would have received the atlantal neural arches dorsally (Figs. 30C and 30D). The postzygapophyses are typical to those of the post-axial cervical vertebrae, being well-developed and with distinct epipophyses, which extend posterior to the postzygapophyses (*Wild, 1973*). Between the postzygapophyses, a wide and elongate postzygapophyseal trough with a roughly straight posterior margin is preserved, similar to that described for the postaxial cervical vertebrae of *Tanystropheus* 'conspicuus' from the Upper Muschelkalk and *Tanystropheus* 'haasi' from Maktesh Ramon in Israel, now considered nomina dubia (sensu *Spiekman & Scheyer, 2019*). Such a trough has also been described for a dorsal vertebra now assigned to *Tanystropheus hydroides* (MSNM BES 215; *Nosotti, 2007*). In contrast to the postaxial cervical vertebrae, the pleurocentrum of the axis bears a distinct ventral keel (Fig. 30B). The neural canal of the axis is straight and does not run substantially through the pleurocentrum (Fig. 31), in contrast to the postaxial cervical (Fig. 32; *Edinger, 1924*; *Wild, 1973*).

No ribs are preserved in articulation with the atlas-axis complex. Four partial cervical ribs can be seen on the slab externally next to the third cervical vertebra (Fig. 1A). Three additional sections of cervical ribs are preserved underneath the skull and have been revealed by the SRµCT data (Fig. 3B, cervical ribs coloured in dark magenta). One rib segment is orientated almost perpendicular to the skull and has broken in three pieces over the right lower jaw and right pterygoid. Another rib fragment is orientated parallel and underneath the right jugal and is also broken over the right lower jaw. Finally, the smallest fragment is orientated parallel to the main axis of the skull and preserved underneath the right pterygoid. However, because these elements are broken and disarticulated, it is unclear whether they represent parts from the same or separate ribs, and therefore it cannot be established how many ribs in total were present in the anterior part of the neck. Thus, although it is almost certain that the axis bore ribs, it is unknown whether there were ribs associated with the atlas.

*Postaxial cervical vertebrae*
The anterior third of the first postaxial cervical vertebra was included in the SRµCT data and reveals an anatomy congruent with that of large-sized *Tanystropheus* specimens (Fig. 32; *Spiekman & Scheyer, 2019*). It is somewhat compressed transversely. The anterior articular surface of the centrum is flat to slightly concave and is taller than it is wide. The ventral surface of the centrum is flattened. The left prezygapophysis is absent but the right one is completely preserved and extends slightly anterior to the centrum. On the right side of the centrum, the two small articulation facets for the tuberculum and capitulum of the cervical rib are preserved near to each other. The neural spine is short but clearly present anteriorly, and gradually reduces in height posteriorly. It is virtually

absent at the posterior end of the section of the vertebra present in the SRµCT data. The inner anatomy visible in the slice data shows that the neural canal clearly passes through the pleurocentrum (Fig. 32), as has previously been described for postaxial cervical vertebrae of *Tanystropheus* spp. and the axis of *Macrocnemus bassanii* (*Edinger, 1924*; *Miedema et al., 2020*; *Wild, 1973*).

The remaining postaxial cervical vertebrae are also transversely compressed. Furthermore, large parts of the cervical vertebrae are reconstructed, obscuring the observation of many anatomical details externally, particularly at their anterior and posterior ends. Their overall morphology and relative lengths (specimen *p* in table 3 of *Wild, 1973*) corresponds to that of other *Tanystropheus hydroides* specimens.

## DISCUSSION

### Configuration of the nasals

Although no complete nasal is known for *Tanystropheus hydroides*, partial nasals can be observed in specimens PIMUZ T 2787 (supplemental figure 2B of *Spiekman et al., 2020*) and PIMUZ T 2819 (supplemental figure 1A of *Spiekman et al., 2020*). The fragmentary nasals of the latter show a similar clear concavity on their dorsal surfaces as described here in PIMUZ T 2790 (see also figure 3 of *Jiang et al., 2011*). The reconstruction of the nasals of *Tanystropheus hydroides* (Fig. 6B) has been inferred from these specimens and through comparison with *Tanystropheus longobardicus*. Complete but disarticulated nasals are preserved in *Tanystropheus longobardicus* specimen PIMUZ T 2484.

The elements lack the clear concavity on the dorsal surface of the nasal seen in PIMUZ T 2790 and PIMUZ T 2819 of *Tanystropheus hydroides*, but it is uncertain whether this represents a true morphological feature because the elements of PIMUZ T 2484, including the nasals, are strongly compressed. Both nasals of PIMUZ T 2484 form a single, tapering process anteriorly (supplemental figure 2D of *Spiekman et al., 2020*). The posterior margin of the nasals and the anterior margins of the corresponding frontals in this specimen allow the articulation between these elements to be inferred confidently (supplemental figure 2D and 2E of *Spiekman et al., 2020*). Their configuration implies that the anteriorly directed process of the nasal in *Tanystropheus longobardicus* represents the lateral process rather than the anterior process (sensu *Pinheiro, Simão-Oliveira & Butler, 2019*) as previously considered. Therefore, it did not form an internarial bar, but instead contacted the dorsal margin of the maxilla and premaxilla. The lack of both a prominent anterior process on the nasal and a well-developed prenarial process on the premaxilla suggests the complete absence of an internarial bar and the presence of confluent external nares in *Tanystropheus longobardicus* (see figure 3E of *Spiekman et al., 2020*). The partially visible nasal in PIMUZ T 2787 of *Tanystropheus hydroides* is preserved directly anterior to the left frontal and it therefore most likely represents the left nasal. However, since the nasal almost certainly was somewhat displaced posteriorly, it cannot be excluded that it represents the right nasal that was also laterally displaced. As in *Tanystropheus longobardicus* only a single clear process projects anteriorly on the nasal and a well-developed prenarial process of the premaxilla is also absent in *Tanystropheus hydroides*. If the nasal of PIMUZ T 2787 indeed represents the left nasal, this would

imply that this process projects anterolaterally from the nasal and would therefore represent the lateral process. Since this corresponds to the configuration present in the congeneric species *Tanystropheus longobardicus*, we consider this the most parsimonious configuration. Consequently, as in *Tanystropheus longobardicus*, the anterior process of the nasal was only very weakly developed and the external nares were also confluent in this species (Fig. 6B).

## The mandible of *Tanystropheus longobardicus*

The reconstruction of the lower jaw of *Tanystropheus hydroides*, based on the well-preserved left mandibular ramus of PIMUZ T 2790, allows for the reinterpretation of the posterior part of the mandible of *Tanystropheus longobardicus*. This was previously reconstructed based on the well-preserved, but strongly compressed and cracked mandible of MSNM BES SC 1018 (see figures 13 and 53B of *Nosotti, 2007*). In this specimen the splenial was interpreted as reaching far posteriorly, approximately to the level of the glenoid fossa in medial view. However, the posterior part of what was interpreted as the splenial has the same morphology as the angular in PIMUZ T 2790 (Fig. 28). Both form the ventral margin of the mandible ventral to the adductor fossa, and both bear the same distinct concave dorsal surface, which receives the ventral margin of the element dorsal to it in a deep groove. Therefore, this structure is reinterpreted as part of the angular and correspondingly, the splenial of *Tanystropheus longobardicus* likely did not reach much further posteriorly than the anterior margin of the adductor fossa. Similarly, the bone previously identified as the angular in medial view in MSNM BES SC 1018 corresponds in position and morphology to the prearticular of PIMUZ T 2790, and we therefore re-identify it as the angular in medial view. The portion of the mandible previously identified as the prearticular of MSNM BES SC 1018 also represents part of the prearticular, which was likely considered as a separate element due to cracks widely present in the specimen.

The sutures on the lateral side of the mandible of *Tanystropheus longobardicus* are hard to establish in all available specimens. They were previously reconstructed largely based on the left mandibular ramus of PIMUZ T 2484, which is broken into two pieces (*Nosotti, 2007*; *Wild, 1973*). However, the reconstructions of this specimen by *Wild (1973)* and *Nosotti (2007)* differ in some respects (see figures 49A and 49B of *Nosotti, 2007*). The suture between the angular and surangular was reconstructed as being positioned more dorsally by *Nosotti (2007)* than by *Wild (1973)*. A foramen observed in PIMUZ T 2484 could correspond to the posterior surangular foramen that we identified in PIMUZ T 2790 (Fig. 28F). In the reconstruction presented by *Wild (1973)*, this foramen is also placed on the surangular, whereas in the arrangement by *Nosotti (2007)* this foramen is located on the angular. No foramen is known on the angular in early archosauromorphs, but the presence of a foramen is much more common on the lateral surface of the surangular (*Ezcurra, 2016*). Furthermore, the distribution of the sutures in the reconstruction by *Wild (1973)*, both for the angular and the surangular, and for the dentary with both these elements, corresponds more closely with that of *Tanystropheus hydroides* than that hypothesised by *Nosotti (2007)*, and it is therefore preferred here. However, the

interpretation of the sutures of the lateral side of the mandible in *Tanystropheus longobardicus* remains somewhat equivocal.

## Palaeobiology and functional morphology of the skull

Due to the highly unusual morphology of *Tanystropheus*, interpretations of its palaeobiology are contentious. Originally considered to be a terrestrial animal (*Peyer, 1931*), a detailed study of all the *Tanystropheus* material from Monte San Giorgio (*Wild, 1973*) suggested that the 'juveniles', now known to represent skeletally mature individuals of *Tanystropheus longobardicus*, were terrestrial and used their tricuspid dentition to feed on insects. *Tanystropheus hydroides*, then considered to represent the adult morphotype of *Tanystropheus longobardicus*, was considered to be largely aquatic, actively pursuing its prey in the water. A detailed study of the functional morphology of the neck of *Tanystropheus* (*Tschanz, 1986*, *1988*) revealed that it represented a rigid, stiffened structure that was strongly limited in its flexibility, particularly in the dorsoventral plane. This was deduced from the relative elongation of the individual cervical vertebrae, the lack of attachment sites for the epaxial musculature due to the reduction of the neural spines of the cervical vertebrae, and the presence of hyperelongate and very thin cervical ribs that would have overlapped each other parallel to the cervical vertebral column to form rigid bundles that supported the long neck. A subsequent histological study of the cervical ribs of *Tanystropheus* found that they consisted largely of lamellar bone (*Jaquier & Scheyer, 2017*), thus representing rigid structures that could have supported the neck ventrally as suggested by *Tschanz (1986)*. In contrast, the posterior processes of the cervical ribs of the terrestrial and long-necked sauropod dinosaurs consist primarily of longitudinal collagen fibres, which indicates that they represent ossified tendons (*Klein, Christian & Sander, 2012*). These ossified tendons would have provided flexibility but were poorly suited to form supportive rods under the neck. The highly stiffened neck of *Tanystropheus* was considered to be poorly adapted for a terrestrial lifestyle, whereas an aquatic lifestyle, in which lateral undulation of the trunk and tail was used for propulsion, seemed more plausible. Additional support for an aquatic lifestyle was provided by the lack of ossification in the tarsus of *Tanystropheus*, which is generally considered an indicator of aquatic habits (*Rieppel, 1989*). The most detailed discussion on the lifestyle of *Tanystropheus* has been presented in *Nosotti (2007)*. Therein, a detailed overview of the literature was provided, and it was concluded that, although both a terrestrial and aquatic lifestyle appear problematic given the overall anatomy of *Tanystropheus*, the latter seems considerably more likely. Specifically, the appendicular skeleton, with proportionally much longer hind limbs than forelimbs would have strongly impeded terrestrial locomotion. Furthermore, the configuration of the tarsus and the metatarsal-like proportions of the first phalanx of digit five indicate that the pes was likely poorly adapted for the propulsive phase of a stride on land. On the other hand, the suggested mode of locomotion by *Tschanz (1986)*, in which *Tanystropheus* used a combination of lateral undulation of the posterior half of the body and strokes of the elongate hind limbs, was considered to be plausible. However, given the poor

hydrodynamic profile of the elongate neck of *Tanystropheus* (*Massare, 1988*; *Troelsen et al., 2019*), and the lack of clear adaptations for aquatic propulsion, *Tanystropheus* was not considered to be a fast swimmer. Nevertheless, stomach remains indicate that *Tanystropheus hydroides* consumed fast-moving fish and cephalopods (*Wild, 1973*). It therefore seems most likely that, rather than actively pursuing its prey, *Tanystropheus hydroides* must have been an ambush predator (*Spiekman et al., 2020*). The presence of a small head on the end of a very elongate neck would have likely allowed *Tanystropheus hydroides* to approach its prey with a lower chance of detection. Additional evidence for an amphibious or semi-aquatic habit was inferred from a bone density analysis of an isolated femur assigned to *Tanystropheus 'conspicuus'* (*Jaquier & Scheyer, 2017*). *Tanystropheus 'conspicuus'* is considered a nomen dubium, but is undoubtedly very closely related, if not synonymous, with *Tanystropheus hydroides* (*Spiekman & Scheyer, 2019*).

An alternative hypothesis was provided by *Renesto (2005)*, in which it was suggested that the neck of *Tanystropheus* might have been lifted considerably above the horizontal plane (contra *Tschanz, 1986*; *Tschanz, 1988*). In this interpretation, the hollow cervical vertebrae would have made the neck sufficiently light to be elevated despite the poorly developed epaxial musculature. As the neck was lifted on land, balance would be maintained by a large muscle mass present at the base of the tail, preventing the animal from falling over forward. *Renesto (2005)* also argued that lateral undulation as a mode of propulsion in water would not have been possible in *Tanystropheus* because the elongate transverse processes/pleurapophyses of the anterior caudal vertebrae and the presence of heterotopic bones directly posterior to the pelvis in approximately 50% of the known specimens did not allow for the required flexibility at the base of the tail. Therefore, *Renesto (2005)* suggested a largely terrestrial lifestyle for *Tanystropheus*, in which it used its long neck to lunge at prey in the water from the shoreline. A recent study (*Renesto & Saller, 2018*) provided an inferred reconstruction of the musculature of the pelvic and proximal tail regions, which suggested that *Tanystropheus* did have a semi-aquatic lifestyle. Providing an updated interpretation of the findings suggested by *Renesto (2005)*, it was found that *Tanystropheus* was able to propel itself through the water by solely using its hind limbs for propulsion, but it was maintained that lateral undulation would have been impossible due to the presence of the heterotopic bones. The presence of slightly transversely expanded neural spines in the trunk and caudal vertebrae was suggested to support this argument, since they were considered to be potentially indicative of the presence of a supraspinous ligament that would have stiffened the trunk (*Renesto & Saller, 2018*). However, no quantitative biomechanical test (e.g. a range of motion analysis of the proximal caudal vertebrae) was provided to assess this hypothesis of a proximally inflexible tail in *Tanystropheus*, and we disagree with some of the conclusions drawn from the morphological observations. Lateral undulation in aquatic reptiles requires oscillating movement of both the trunk and tail, with undulation most predominantly being generated by the pelvic region rather than the base of the tail (*Fish, 1984*; *Massare, 1988*). There is no evidence that the heterotopic bones of *Tanystropheus*, which are also present in the likely aquatic tanystropheid *Tanytrachelos*, nor the presence of long transverse processes/pleurapophyses of the anterior caudal

vertebrae as suggested in *Renesto (2005)*, would have impeded lateral undulation. On the contrary, long transverse processes/pleurapophyses are widespread in the anterior caudal vertebrae of aquatic reptiles exhibiting lateral undulation of the tail (e.g. extant crocodylians, nothosaurs, spinosaurids, *Ibrahim et al., 2020*; *Rieppel, 2000*) and they provide important muscle attachment sites for undulation. Furthermore, the slight expansion of the neural spines of the trunk and tail vertebrae in *Tanystropheus* possibly indicates the presence of supraspinous ligament of some sort, but it does not provide convincing evidence that the trunk and tail were highly stiffened. The horizontal orientation of the articulation between the pre -and postzygapophyses of the dorsal vertebrae has been suggested to indicate some lateral flexibility in the vertebral column (*Nosotti, 2007*; *Tschanz, 1986*) and the accumulation of even limited lateral movement along the dorsal and caudal sections of the vertebral column might have been sufficient to provide undulatory propulsion. We therefore disagree that the available morphological information provides conclusive evidence that *Tanystropheus* was unable to perform lateral undulation. Furthermore, in support of the hypothesis of underwater propulsion being exclusively produced by strokes of the hindlimbs, *Renesto & Saller (2018)* suggested that the pes of *Tanystropheus* would have formed 'an effective rowing device if the feet were webbed'. However, the metatarsals of *Tanystropheus*, like those of other tanystropheids (*Macrocnemus*, *Amotosaurus*, *Tanytrachelos* and *Langobardisaurus*) were tightly bunched in the majority of known specimens (*Ezcurra, 2016*, character 565; *Olsen, 1979*; *Renesto, Dalla Vecchia & Peters, 2002*). This indicates that they were a single functional unit and that the spread of the phalanges was almost certainly limited (*Nosotti, 2007*). Although the pedes possibly did aid in swimming, it seems unlikely that only strokes of the hind limbs alone ('rowing' sensu *Renesto & Saller, 2018*) would have provided sufficient underwater propulsion.

The current study focuses on the cranial morphology of *Tanystropheus hydroides*, and therefore a more detailed investigation into aquatic locomotion in *Tanystropheus* is outside the scope of the present study. However, the morphology of the snout and the dentition *Tanystropheus hydroides* as revealed by PIMUZ T 2790 additionally support an at least semi-aquatic lifestyle for this taxon. The snout is dorsoventrally flattened with the external nares facing entirely dorsally on the top of the snout. This is a clear deviation from the anteroposteriorly directed nares of other archosauromorphs and represents a close convergence to the snout of crown-group crocodylians (Fig. 6B; *Spiekman et al., 2020*). Furthermore, the marginal dentition of PIMUZ T 2790 conform to the 'fish-trap' dentition also present in certain sauropterygians that employed a laterally directed snapping bite (*Rieppel, 2002*). These features were recently discussed in *Spiekman et al. (2020)*. Here, we additionally provide an overview of other aspects of the skull relevant to the reconstructing the palaeobiology of *Tanystropheus hydroides*.

*Endocranial anatomy*
The flocculus is a cerebral lobe that regulates both head and eye stabilization during movement (*Voogd & Wylie, 2004*). Therefore, it is possible that the presence of a large flocculus in *Tanystropheus hydroides* is related to the complex head and eye stabilization

that might have been required because of its extremely elongated neck. However, in the long-necked sauropod dinosaurs a reduction in relative floccular size is observed compared to the plesiomorphic condition (*Bronzati et al., 2017*). It was tentatively suggested that the elongation of the neck in sauropods might have influenced floccular size reduction, since head movement would play a less important role in balancing the trunk with increased neck length. The influence of floccular size on function in vertebrates is currently known. Although relative neck length did not represent a tested variable, a recent statistical investigation could not find any correlation between floccular size and six variables related to locomotion, agility, and feeding type in a total sample of 106 extant mammals and birds (*Ferreira-Cardoso et al., 2017*).

The orientation of the LSC of the endosseous labyrinth has been used as a proxy for head posture, although its reliability has been questioned (*Brown et al., 2019*; *Hullar, 2006*; *Marugán-Lobón, Chiappe & Farke, 2013*; *Neenan & Scheyer, 2012*). Nevertheless, the virtually horizontal orientation of the LSC in PIMUZ T 2790 could suggest a horizontal head posture for *Tanystropheus hydroides*. The elongate cochlea of *Tanystropheus hydroides* indicates advanced auditory capabilities, as the length of the cochlear duct is a relatively reliable proxy for inferring auditory capabilities in reptiles, including birds (*Walsh et al., 2009*). As was also discussed in *Spiekman et al. (2020)*, the geometry of the semicircular canals is correlated to certain aquatic adaptations. Most evidently, the semicircular canals of deep-diving pelagic reptiles are generally short and robust in comparison to terrestrial and nearshore aquatic taxa (*Evers et al., 2019*; *Neenan et al., 2017*; *Neenan & Scheyer, 2012*; *Schwab et al., 2020*). The comparatively elongate and gracile semicircular canals and common crus of PIMUZ T 2790 are clearly in correspondence to the type observed in terrestrial and nearshore aquatic taxa, and therefore it can be excluded that *Tanystropheus hydroides* was a deep-diving pelagic animal. These findings are also supported by a taphonomical analysis of *Tanystropheus* specimens from the Besano Formation of Monte San Giorgio (*Beardmore & Furrer, 2017*).

*Otic joint*
In PIMUZ T 2790, the quadrate facet on the posteroventral surface of the squamosal bears a deep pyramidal excavation which was likely covered by a cartilaginous cap (Fig. 13), whereas the dorsal surface of the dorsal head of the quadrate is anteroposteriorly elongated and terminates posteriorly in a conspicuous hook (Fig. 14). This could have allowed the dorsal head of the quadrate to rotate within the concave facet on the squamosal in a loose articulation, thus forming a permissive kinematic linkage (sensu *Holliday & Witmer, 2008*). A movable otic joint is a prerequisite for streptostyly, which is a form of cranial kinesis in which the quadrate has the ability to swing approximately anteroposteriorly with the rotational axis being formed by the joint between the dorsal head of the quadrate and the squamosal and/or supratemporal (*Metzger, 2002*). In addition to a movable otic joint (=quadratosquamosal joint), streptostyly also requires a sliding contact between the pterygoid flange of the quadrate and the quadrate ramus of the pterygoid, to allow the quadrate to move independently of the pterygoid. This movement could have been facilitated by a concavity present on the distal portion of the pterygoid flange of the

quadrate in PIMUZ T 2790 (Fig. 14). Furthermore, because the infratemporal bar was incomplete, the independent movement of the quadrate would not have been impeded by a connection of the jugal to it or the quadratojugal. However, the presence of a movable joint does not necessarily imply the presence of powered cranial kinesis, and many extant taxa that possess several movable joints do not exhibit kinesis in vivo (*Holliday & Witmer, 2008*). The absence of clear indicators of additional movable joints (e.g. between the frontals and parietals; mesokinesis) makes it uncertain that the skull of *Tanystropheus hydroides* exhibited cranial kinesis in vivo. Nevertheless, although its functional interpretation is currently unclear, the otic joint of *Tanystropheus hydroides* represents a remarkable and noteworthy configuration among archosauromorphs.

*Hyobranchial apparatus*
Suction feeding in vertebrates can be inferred from skeletal adaptations of the hyobranchial apparatus. The contraction of muscles (e.g. the *M. coracohyoideus* and *M. sternohyoideus*) connecting the hyobranchial apparatus to the pectoral girdle retracts this apparatus, which creates a negative pressure within the buccal cavity (*Motani et al., 2013*). The apparatus of a suction feeder generally exhibits several adaptations to handle the considerable stress it experiences during a retraction that is strong enough to create a sufficiently negative pressure. These adaptations typically include the presence of an ossified hyoid corpus and robusticity of the hyobranchial rods that connect to the corpus.

No elements of the hyobranchial apparatus are preserved in PIMUZ T 2790. However, ceratobranchial I is known from *Tanystropheus hydroides* specimens PIMUZ T 2819 and SNSB-BSPG 1953 XV 2. This slightly curved bone is comparatively thin (measurements provided on page 44 of *Wild, 1973*). No other hyobranchial elements are known and therefore there is no evidence of an ossified hyoid corpus in *Tanystropheus hydroides*. Furthermore, the 'fish-trap' dentition seems to preclude the possibility of suction feeding in *Tanystropheus hydroides*, since the elongate fangs would interfere with the prey item entering the buccal cavity.

Hyobranchial elements were previously also identified in *Tanystropheus longobardicus* specimens PIMUZ T 2791, PIMUZ T 2484 (both in *Wild, 1973*), and MSNM BES SC 265 (*Nosotti, 2007*). We were not able to corroborate the identification of these elements due to poor preservation (PIMUZ T 2791 and MSNM BES SC 265) or disarticulation, which does not allow a thin ceratobranchial I to be clearly differentiated from surrounding gastralia and rib fragments (PIMUZ T 2484). However, a pair of hyobranchial rods, possibly ceratobranchial I, are clearly preserved in *Tanystropheus longobardicus* specimen PIMUZ T 3901, directly posterior to the left lower jaw (see figure 2B of *Spiekman & Scheyer, 2019*). These elements have a rod-like shape and are slightly curved. They are relatively gracile (total length: 10.8 mm; maximum width: 0.9 mm). As in *Tanystropheus hydroides*, an ossified hyoid corpus was likely not present in *Tanystropheus longobardicus*. The apparent absence of an ossified hyoid corpus and the lack of robust hyobranchial rods therefore indicate that both species of *Tanystropheus* were not suction feeders.

*Salt glands*

A tetrapod spending large amounts of time in a marine environment needs to excrete excess salt from its body. This is achieved by salt glands. Salt glands are generally located within the nasal or orbital cavities (an overview is provided in *Babonis & Brischoux, 2012*). No evidence for a salt gland can be found in the orbital cavity of *Tanystropheus hydroides*, but the large space of confluent external nares could have facilitated such a gland. This is also the case in Triassic sauropterygians such as *Nothosaurus marchicus* (*Voeten et al., 2018*). However, due to the poor preservation of the nasals, no skeletal correlates for a salt gland have been found in the SRµCT data of PIMUZ T 2790. The generally poor preservation of the nasals in all known *Tanystropheus* specimens, both of *Tanystropheus hydroides* and *Tanystropheus longobardicus*, limits the observation of this trait and therefore nothing can be said unambiguously about the presence of salt glands in *Tanystropheus* currently.

## CONCLUSIONS

The detailed morphological study of the SRµCT data presented here provides much additional information on the skull and anterior cervical region of *Tanystropheus hydroides*, and highlights that the configuration of the skull is entirely unique among archosauromorphs and adapted for a lifestyle as an aquatic ambush predator. The external nares are confluent and positioned on the dorsal surface of the snout. As in *Tanystropheus longobardicus*, but in contrast to other archosauromorphs, the frontals are wide across the interorbital region and form most of the dorsal margin of the orbit. The postorbital region is characterized by dorsally directed supratemporal fenestrae, laterally facing supratemporal fossae of the parietals, a dorsoventrally tall squamosal, and dorsally hooked quadrate. The braincase is characterized by several derived archosaur traits, such as the presence of a laterosphenoid, and the ossification of the lateral wall of the braincase by a connection between the prootics and parabasisphenoid. However, the braincase differs from more derived archosauriforms in the morphology of the ventral ramus of the opisthotic, the horizontal orientation of the parabasisphenoid, and the lack of a clearly defined crista prootica. There is no indication of pneumatization in the braincase of *Tanystropheus hydroides*. The flocculus was pronounced. The cochlear duct of the endosseous labyrinth is well-developed, indicating advanced auditory capabilities in *Tanystropheus hydroides*. The configuration of the palate is distinct from other archosauromorphs, including *Tanystropheus longobardicus*, but possibly resembles in some respects the poorly known palatal region of the marine archosauromorph *Dinocephalosaurus orientalis*. The marginal dentition is also comparable to that of *Dinocephalosaurus orientalis* and forms a clear indication of a piscivorous diet. The skull of *Tanystropheus hydroides* possessed an unusual articulation between the quadrate and squamosal and, together with the sliding contact between the quadrate and pterygoid, provided the osteological potential to retract the quadrate posteriorly. The dentition, as well as the morphology of the hyoid apparatus, indicates that *Tanystropheus hydroides* was clearly not a suction feeder but likely employed a laterally directed snapping bite like certain sauropterygians. The lower jaw of *Tanystropheus hydroides* allows for the

reinterpretation of the mandible of *Tanystropheus longobardicus*, specifically with regards to the splenial, surangular, angular and prearticular.

The SRµCT data of PIMUZ T 2790 previously revealed the taxonomic and ecomorphological distinction between *Tanystropheus hydroides* and *Tanystropheus longobardicus* from the Besano Formation of Monte San Giorgio, thus indicating niche partitioning within the genus (*Spiekman et al., 2020*). Furthermore, the remarkable divergence in cranial morphology between these two *Tanystropheus* species and other tanystropheid taxa such as *Macrocnemus bassanii* as revealed in this study provides a clear indication of the ecomorphological diversity of Tanystropheidae, which was previously only marginally understood.

## INSTITUTIONAL ABBREVIATIONS

**BP**      Evolutionary Studies Institute (previously Bernard Price Institute for Palaeontological Research), University of Witwatersrand, Johannesburg, South Africa

**BSPG**    Bayerische Staatssammlung für Paläontologie und Geologie, Munich, Germany

**IVPP**    Institute of Vertebrate Paleontology and Paleoanthropology, Beijing, China

**MFSN**    Museo Friulano di Scienze Naturali, Udine, Italy

**MSNM**    Museo di Storia Naturale, Milan, Italy

**NMK**     Naturkundemuseum im Ottoneum der Stadt Kassel, Kassel, Germany

**NMS**     National Museums Scotland, Edinburgh, UK

**PIMUZ**   Paläontologisches Institut und Museum der Universität Zürich, Zurich, Switzerland

**SAM**     PK - Iziko South African Museum, Cape Town, South Africa

**ZAR**     Zarzaitine Collection, Muséum National d'Histoire Naturelle, Paris, France

## ACKNOWLEDGEMENTS

Christian Klug allowed for permission to scan PIMUZ T 2790 and access to comparative material from the PIMUZ collections. Additionally, we are grateful for collection access by the following curators: Bernhard Zipfel, Sifelani Jirah, and Jonah Choiniere (BP), Oliver Rauhut (BSPG), Li Chun (IVPP), Cristiano Dal Sasso (MSNM), Cornelia Kurz (NMK), Claire Browning and Roger Smith (SAM-PK), Rainer Schoch (SMNS), and Nour-Eddine Jalil (ZAR). Dylan Bastiaans kindly helped in rendering the images for Figs. 31 and 32 in Mimics Research v19.0. We would like to thank our colleagues Dylan Bastiaans, Feiko Miedema, Christian Klug, Roger Benson, Jonah Choiniere, Roland Sookias, Richard Butler, Fabio Dalla Vecchia, Oliver Rauhut, and Gabriela Sobral for discussions on the morphology of *Tanystropheus*. Reviewers Gabriela Sobral and Felipe Pinheiro provided constructive comments and suggestions that improved the quality of this paper. We are grateful to the European Synchrotron Radiation Facility (ESRF), Grenoble, France, for synchrotron radiation beamtime at beamline BM05.

### Funding

This study is part of the Swiss National Science Foundation project granted to Torsten Scheyer (no. 2′5321-162775). James Neenan was funded by a Leverhulme Trust Early Career Fellowship (ECF-2017-360). The funders had no role in study design, data collection and analysis, decision to publish, or preparation of the manuscript.

### Grant Disclosures

The following grant information was disclosed by the authors:
Swiss National Science Foundation project granted to Torsten Scheyer: 2′5321–162775.
Leverhulme Trust Early Career Fellowship: ECF-2017-360.

### Competing Interests

The authors declare that they have no competing interests.

### Author Contributions

- Stephan N.F. Spiekman conceived and designed the experiments, performed the experiments, analyzed the data, prepared figures and/or tables, authored or reviewed drafts of the paper, and approved the final draft.
- James M. Neenan analyzed the data, authored or reviewed drafts of the paper, and approved the final draft.
- Nicholas C. Fraser conceived and designed the experiments, authored or reviewed drafts of the paper, and approved the final draft.
- Vincent Fernandez performed the experiments, authored or reviewed drafts of the paper, and approved the final draft.
- Olivier Rieppel conceived and designed the experiments, authored or reviewed drafts of the paper, and approved the final draft.
- Stefania Nosotti conceived and designed the experiments, authored or reviewed drafts of the paper, and approved the final draft.
- Torsten M. Scheyer conceived and designed the experiments, analyzed the data, authored or reviewed drafts of the paper, and approved the final draft.

### Data Availability

The studied specimen, PIMUZ T 2790, is housed at the Palaeontological Institute and Museum of the University of Zurich, Switzerland. Another figured specimen of Tanystropheus longobardicus, PIMUZ T 2484, is housed at the same institution.

The raw data of the synchrotron CT scan of PIMUZ T 2790 is available at the ESRF heritage database for palaeontology, evolutionary biology and archaeology: http://paleo.esrf.eu/.

A free account may be required to access the data without any other restriction.

Raw (SRCT) data: http://paleo.esrf.eu/galleries/vertebrate_paleontology/ Archosauromorpha/Tanystropheus_hydroides/org_slices/.

The reconstructed skull model: http://paleo.esrf.eu/galleries/vertebrate_paleontology/ Archosauromorpha/Tanystropheus_hydroides/stl_files/.

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
