# Peer review of "The cranial morphology of Tanystropheus hydroides (Tanystropheidae, Archosauromorpha) as revealed by synchrotron microtomography"

_PeerJ, doi:10.7717/peerj.10299_

## Round 0.1 · original submission · Minor Revisions

The reviewers have suggested some relatively minor revisions, as outlined in the accompanying comments. Please note the annotated PDF provided by Reviewer 1.

·

Basic reporting

In this important contribution, Spiekman et al. describe in exceptional detail the skull and anterior axial morphology of the recently proposed taxon Tanystropheus hydroides. The description is mainly based in 3D modelling of synchrotron tomographic data, and the authors did a superb job in segmenting the individual bones, assembling a reconstructed skull and producing a braincase endocast model. Although the MS has some weaknesses in the present version, the published paper will probably stand out as one of the best anatomical descriptions thus far provided for an early archosauromorph. The manuscript is written in plain and professional English throughout, but some minor language issues detected by me are highlighted in the attached PDF file (but please bear in mind that I’m not a native English speaker myself). The provided background is adequate and well-referenced.

Experimental design

This is an original research. Although a similar study was recently published by the authors (Spiekman et al. 2020), the main goal of the new MS is to provide a detailed anatomical description of T. hydroides, which they do well. The methods are described in sufficient detail and the research is meaningful and relevant, providing valuable information on the anatomy of an enigmatic group of extinct reptiles.

Validity of the findings

Apart from the excellent anatomical description (which is the core of the ms), my main concerns about the work of Spiekman et al. are related to the robustness of their discussion. The anatomical data are employed to corroborate hypotheses that flirt with speculation or that, even if true, lack sound evidence in the context of the present version of the manuscript. This is the case of the putative streptostyly and the argumentation surrounding the aquatic habits of Tanystropheus. Again, I’m not stating that the authors are not correct, but their argumentation may be weak in some points (see below), and would probably benefit from further discussion or different methodological approaches. Most of my suggestions and commentaries are highlighted in the pdf file I’m sending attached to this, but I’ll try to detail some important issues below:

- Aquatic adaptations of Tanystropheus: The authors, albeit not being so explicit, appear to support the hypothesis of an aquatic Tanystropheus (echoing what was already sustained by Spiekman et al. 2020). Their case, however, is mainly based on the confluent dorsally-positioned nares and the ‘fish-trap’ dentition. Although present in several arguably aquatic animals, confluent/dorsal nares are not an aquatic adaptation per se and are also displayed, among early archosauromorphs, by the fully terrestrial rhynchosaurs. It is plain enough that Tanystropheus fed on marine animals, but its dentition, allied to the overspecialized neck and the overall conservative trunk and limb morphologies, appear to support the ‘shoreline fishing’ habit proposed by Renesto (2005), whereas make unlikely a prevalent aquatic habit for this animal. I should also note that, although lacking biomechanical support, the proposed habit of ‘positioning on the seafloor’ while approaching prey (Spiekman et al. 2020) seems unlikely, as the lack of pachyostosis allied to the lightly-built axial skeleton would hinder a neutral buoyancy to Tanystropheus. New evidence provided by Spiekman et al. (the submitted manuscript) also seem to indicate that Tanystropheus was poorly ‘built’ for a mainly aquatic lifestyle. This include the semicircular canals morphology (pag. 35) and the lack of evidence supporting suction feeding (pag. 36) or salt glands (pag. 37). Again, I’m not stating here that Tanystropheus wasn’t aquatic, and I don’t even have a personal opinion on this matter. I need to say, however that, if the authors choose to argue in favor of a mainly aquatic lifestyle, they should provide enough evidence supporting this hypothesis and discuss the wealth of evidence indicating otherwise (most are summarized by Renesto 2005).

- Cranial kinesis in Tanystropheus: Spiekman et al. (pag. 36) propose streptostylic intracranial movements in Tanystropheus hydroides based on two osteological correlates, namely a supposedly synovial joint between the head of the quadrate and the squamosal (an otic joint sensu Holliday and Witmer 2008) and a permissive linkage between the quadrate itself and the pterygoid. Cranial kinesis is an uncommon trait among diapsids, and its proposition for extinct reptiles is conditioned to the detection of several musculoskeletal structures (far more than the two correlates proposed by the authors). I’ll quote Holliday and Witmer (2008): “We recognize minimally four criteria that are necessary for inferences of powered cranial kinesis in fossil (and extant) taxa: (1) a synovial basal (basipterygopterygoid) joint; (2) a synovial otic (quadratosquamosal) joint; (3) protractor musculature; and (4) permissive kinematic linkages. Clearly, intracranial synovial joints such as the otic and basal joints are necessary to allow kinetic movement, and protractor muscles would help drive the system. Additionally, the connections between bones (kinematic linkages) must permit and not obstruct movement” (pag. 1074). It is noteworthy that even living taxa presenting all those correlates may not display cranial movements in vivo and that basal and otic synovial joints are apparently plesiomorphic in Diapsida (Holliday and Witmer 2008). As such, the proposed streptostyly in Tanystropheus is far from convincing and I suggest the authors to verify whether more compelling evidence can be drawn from Tanystropheus skull or to simply abandon that hypothesis.

- I have the feeling that the description, although exceptionally well-written, lacks comparisons with a broader sample of early archosauromorphs/archosauriforms. The comparisons are virtually restricted to timid commentaries about the condition observed in a few taxa, such as Macrocnemus, Prolacerta and Azendohsaurus. I’ve highlighted several passages where further comparisons would be welcome. However, bearing this in mind, the paper would benefit from a thorough revision of the comparative description.

- I have alternative interpretations of some bones (nasals, postorbitals and postfrontals). All of them are explained in the annotated PDF. Of course, the authors are by far the best persons to judge whether I’m right or not, but I suggest to double check the issues I was able to detect.

- The paper is stupendously well illustrated, but I still miss detailed photographs of the specimen. CT data can be misleading in the fine detection of sutures, and some structures (e.g. the dentition) would be better illustrated through photographs than through digital models.

- Please provide a Systematic Paleontology section, including referred material, details on the horizon and locality, as well as an updated diagnosis (I’ve highlighted some potential autapomorphies on the annotated PDF). I know that a Systematic Paleontology was provided by Spiekman et al. (2020), but it is also of great relevance for the present submission.

- As you have all those great 3D models, it would be nice to provide a table of measurements. That could be useful for future researchers! :¬)

My best wishes or the authors and congrats for the great job!

Felipe L. Pinheiro

Additional comments

-

·

Basic reporting

The text is clear and very written, with only very minor issues (eg.: line 54 - Tanystropheidae (the clade) is; line 241: the parietals are; line 663 - cannot BE confidently; line 750 - oriented). Also, in the legend of figure 5, change "dental" to "dentary".

The manuscript is structured in a reasonable manner, in which the data presented support very well the statements made. I would only suggest to move measurements of the skull in lines 90-93 to the description section

The figures are also well done and very helpful to better understand the text. My only suggestion is for authors to label some structures in figure 2, especially the frontals, which are mentioned in line 115. The authors must also be careful to call figures in the order of their appearance.

The authors must not forget to update the status of the Miadema et al. and Spiekman et al. references.

Experimental design

The manuscript presents very high quality data. CT scanning, segmentation, and reconstruction of the material are all above standard. The information yielded is very impressive for a mostly compressed material. The anatomical analysis was carried out carefully.

Validity of the findings

The findings significantly increase our knowledge on the anatomy of the enigmatic tanystropheids. It also provides hitherto unknown information on several structures such as the braincase. Both the reconstruction of the skull and the ecomorphological implications of the anatomy described are sufficiently supported.

Additional comments

Dear authors,

Congratulations on the manuscript detailing the cranial morphology of Tanystropheus. The manuscript is very well written and logically organised, with high quality images. The anatomical analysis is very sound and the statements are sufficiently discussed and supported.

I have only three minor suggestions to make.

Lines 279-290: Could this difference in fossa morphology be due to ontogeny? The fossa changes shape as the animals grow in many mammals, just as the sagittal crest becomes taller. Eg: Holbrook 2006 (https://zslpublications.onlinelibrary.wiley.com/doi/abs/10.1017/S0952836902000250).

Lines 634-635: Please re-phrase. Even though the openings have been incorrectly identified, the fenestra ovalis is still framed by opisthotic and prootic.

Lines 117-1180: The authors could include a comment on the seemingly contradictory findings published for the elongation of sauropod necks and reduction of their floccular fossa - Bronzati et al. 2017 (https://doi.org/10.1038/s41598-017-11737-5).

---

## Round 0.2 · accepted · Accept

Thank you for your close attention to the comments from the reviewers.